# Identification of molecular cluster evaporation rates, cluster formation enthalpies and entropies by Monte Carlo method

Anna Shcherbacheva[1], Tracey Balehowsky[2], Jakub Kubečka[1], Tinja Olenius[3], Tapio Helin[4], Heikki Haario[4,5], Marko Laine[5], Theo Kurtén[6,1], and Hanna Vehkamäki[1]

[1]Institute for Atmospheric and Earth System Research, P.O. Box 64 00014 University of Helsinki, Finland
[2]Department of Mathematics and Statistics Subunit, P.O. Box 64 00014 University of Helsinki, Finland
[3]Department of Environmental Science and Analytical Chemistry & Bolin Centre for Climate Research, Stockholm University, Svante Arrhenius väg 8, SE-11418 Stockholm, Sweden
[4]LUT School of Engineering Science, Lappeenranta-Lahti University of Technology, P.O.Box 20 FI-53851 Lappeenranta, Finland
[5]Finnish Meteorological Institute, P.O. Box 503, FI-00101 Helsinki, Finland Finland
[6]Department of Chemistry, P.O. Box 55 FI-00014 University of Helsinki, Finland

**Correspondence:** Anna Shcherbacheva (anna.shcherbacheva@helsinki.fi)

**Abstract.** We address the problem of identifying the evaporation rates for neutral molecular clusters from synthetic (computer-simulated) cluster concentrations. We applied Bayesian parameter estimation using a Markov chain Monte Carlo (MCMC) algorithm to determine cluster evaporation/fragmentation rates from known cluster distributions, assuming that the cluster collision rates are known. We used the Atmospheric Cluster Dynamic Code (ACDC) with evaporation rates based on quantum chemical calculations to generate cluster distributions for a set of electrically neutral sulphuric acid and ammonia clusters. We then treated these concentrations as synthetic experimental data, and tested two approaches for estimating the evaporation rates. First we have studied a scenario where at one single temperature time-dependent cluster distributions are measured before the system reaches a steady-state. In the second scenario only steady-state cluster distributions are measured, but at several temperatures. This allowed us to use multiple sets of concentrations at different temperatures. Additionally, in the latter case the evaporation rates were represented in terms of cluster formation enthalpies and entropies which were considered to be free parameters. This reparametrization reduced the number of unknown parameters, since several evaporation rates depend on the same cluster formation enthalpy and entropy values. We also estimated the evaporation rates using synthetic steady-state cluster concentration data at one temperature (which has appeared in previous literature) and compared our two study cases to this setting. Both the transient and two-temperature steady-state concentration data estimated the evaporation rates with less variance than the steady-state one temperature case.

We show that in the second setting, even if only two temperatures were used, the temperature-dependent steady-state data outperforms the first setting for parameter estimation. We can thus conclude that for experimentally determining evaporation rates, cluster distribution measurements at several temperatures are recommended over time-dependent measurements at one temperature.

# 1 Introduction

The formation of molecular clusters, and their subsequent growth to aerosol particles, is an important yet poorly understood process in our atmosphere. Clusters and aerosols affect both climate, air chemistry (Yu and Turco, 2000), evapotranspiration in forest environments (Yan et al., 2018), and many other atmospheric processes (Lee et al., 2003).

Recent developments in mass spectrometers have enabled the detection, quantification, and chemical characterization of ionic clusters containing between one and some tens of molecules at atmospherically relevant mixing ratios [1] (Eisele and Hanson, 2000; Junninen et al., 2010; Zhao et al., 2010; Ehn et al., 2014; Almeida et al., 2013; Bianchi et al., 2016). Molecular clusters in atmospheric conditions are predominantly electrically neutral, and must thus be charged prior to mass spectrometric detection. This may affect the measurement results, as only part of the sample molecules or clusters may be charged (Hyttinen et al., 2018), and the charging may also alter cluster compositions. For example, for sulfuric acid base clusters, negative charging tends to lead to loss of base molecules, and positive charging to loss of acid molecules (Ortega et al., 2012). Modelling is thus needed to connect measured ion cluster distributions to the original neutral population.

Even when the atmospheric cluster distribution can be accurately deduced from experimental data, this does not quantify the individual kinetic parameters, such as the cluster collision and evaporation rates (Kupiainen-Määttä, 2016). Collision rates may be computed from kinetic gas theory or classical trajectory simulations with reasonable accuracy (Matsugi, 2018), although recent research has shown that long-range attractive interactions may enhance collision rates (Yang et al., 2018), for example by around a factor of 2-3 for $H_2SO_4 - H_2SO_4$ collisions (Halonen et al., 2019). These relatively minor uncertainties in the collision rates are dwarfed by the error margins of cluster evaporation rates. In computational applications, evaporation rates are usually computed using the detailed balance assumption together with the free energies of cluster formation, which can in turn be computed using quantum chemical (QC) methods, (Kurtén et al., 2007; Ortega et al., 2012; Elm et al., 2013; Elm and Kristensen, 2017; Yu et al., 2018). Unfortunately, the evaporation rates depend exponentially on the free energies variations of several kcal/mol between different QC methods thus translate into orders of magnitude differences in evaporation rates (Kupiainen-Määttä et al., 2013; Nadykto et al., 2014).

Despite uncertainties involved in computational estimates of collision and evaporation rates, cluster population dynamic models based on Becker-Döring equations have been able to predict the sulphuric acid concentration dependence of cluster concentrations (Olenius et al., 2013a), and even absolute particle formation rates (Almeida et al., 2013) in sulphuric acid-ammonia and sulphuric acid-DMA systems, without empirical model calibration or parameter tuning. The Becker-Döring equations are a system of Ordinary Differential Equations (ODE), which account for cluster birth and death processes (which depend on the collision and evaporation rates), as well as external cluster sinks and sources. In both studies (Olenius et al. (2013a) and Almeida et al. (2013)), these equations were implemented through the Atmospheric Cluster Dynamic Code (ACDC) (Mc-Grath et al., 2012), using kinetic gas theory collision rates, and standard quantum chemistry techniques for computing cluster formation free energies (and thus evaporation rates).

---

[1]around or below one part per trillion (ppt)

In mathematical terms, the prediction of cluster concentrations using known collision and evaporation rates is called the forward problem. The associated inverse problem is to use known cluster concentrations to deduce the collision and evaporation rates. The inverse problem can be addressed with Bayesian approaches such as Markov chain Monte Carlo (MCMC) methods.

In a recent paper by Kupiainen-Määttä (2016), Differential Evolution (DE) MCMC (Braak, 2006) was applied to determine evaporation rates for negatively charged sulphuric acid and ammonia clusters (containing up to five of each type of molecules, with the $HSO_4^-$ ion here defined as an "acid"). This study used steady-state cluster concentrations measured in the CLOUD [2] chamber experiment at constant temperature, with varying sulphuric acid and ammonia concentrations (we refer to Almeida et al. (2013) for details relevant to the experimental data). Collision rates were taken from kinetic gas theory. Kupiainen-

Määttä (2016) concluded that these data were insufficient for estimation of all the evaporation rate coefficients. Another recent paper (Kürten, 2019) reported thermodynamic data (cluster formation enthalpies and entropies) for 11 neutral sulphuric acid and ammonia clusters. In the CLOUD experiment, these were deduced from new particle formation (NPF) rates measured at 5 different temperatures, over a wide range of sulphuric acid and ammonia concentrations. Most of the thermodynamic parameters could not be narrowly constrained, as the ranges of cluster formation enthalpies and entropies that reproduced the

measured NPF rates were quite wide. However, for each cluster only one monomer evaporation rate was taken into account (either acid or base). Furthermore, the NPF rates obtained using the fitted parameters were systematically lower than the measured ones for warmer temperatures ($\geq$ 248 K).

In this study, we test which combinations of experimental data and fitted parameters lead to the best identification of the evaporation rates. As experiments are expensive and time-consuming to perform, we use synthetic cluster concentration data

created from ACDC simulations to test if the use of time-dependent cluster distribution data would significantly improve the accuracy of the evaporation rates. Use of synthetic data also allows us to know for sure if our inverse modelling actually produces the correct kinetic parameters or not, which would not be possible with experimental concentration data. As in the Kupiainen-Määttä (2016) study, we compute collision rates from kinetic gas theory, while the evaporation rates used to generate our synthetic data are calculated from Gibbs free energies published by Ortega et al. (2012). Note that the conclusions of this

study are not sensitive to the accuracy of the quantum chemical data, as our focus is on the inverse problem of how to determine evaporation rates from known concentrations rather than the forward problem.

For simplicity, we consider the case of neutral sulphuric acid-ammonia clusters containing up to five of each type of molecules. Studying neutral clusters has the advantage that we can restrict ourselves to a smaller set of kinetic parameters, and ignore uncertainties related to charging and neutralization processes. In situations where a large fraction of the clusters are

80 charged, accurate modelling would require at least three times as many parameters, as both the negative, positive and neutral cluster populations interact with each other. The downside of this simplification is that we lose the direct connection to potential real-life experiments, as neutral atmospheric clusters cannot currently be measured without first charging them.

We investigate two different scenarios for estimating evaporation rates. First, we test the use of time-dependent cluster concentrations measured before the system has attained a steady state. This is motivated by the fact that this transient data

should provide additional information about the speed of the processes, which is missing from the steady-state data. Second,

---

[2]Cosmics Leaving OUtdoor Droplets

we apply the approach of Kürten (2019), and express the evaporation rates as parameterized functions of the temperature, with the cluster formation enthalpies and entropies (assumed here to be temperature-independent) as the unknown parameters. This reparametrization is useful for two reasons. First, since the formation enthalpies and entropies of the monomers can be set to zero, and since several evaporation rates depend on the same enthalpy and entropy values, the dimension of the unknown parameter space for our problem is actually reduced, despite the apparent doubling of the number of parameters. Second, utilizing the temperature dependence allows us to produce and use arbitrarily many synthetic data sets at various temperatures, which mathematically has a regularizing effect on the problem. Note that unlike in Kürten (2019), all possible evaporation processes, including cluster fissions into two daughter clusters, are taken into consideration.

## 2 SIMULATION METHODS

In this section we describe the methods used to create synthetic cluster concentration data sets. We also explain the Monte Carlo type algorithms used to estimate the cluster evaporation rates from the data sets.

### 2.1 Generation of synthetic data

The 16 cluster types included in our study are summarized in Table 1. To save computational time, we have excluded clusters where the number of acid and base molecules differs significantly from each other. Irrespective of the level of theory, quantum chemical data predict that these clusters will have very high evaporation rates, leading to negligibly small concentrations. This is also supported by mass spectrometric measurements showing that the clusters with highest concentrations have roughly the same number of acid and base molecules (Kirkby et al., 2011; Schobesberger et al., 2015; Elm and Kristensen, 2017; Yu et al., 2018). The ammonia monomer mixing ratio is assumed to remain constant in each individual simulation, and varied between 5 and 200 ppt. (These correspond to concentrations of $1.3 \times 10^8$ and $5.0 \times 10^9$ molecules per $cm^3$ for the temperature ranges studied here, respectively). The sulfuric acid monomer source rate is kept constant at $Q = 6.3 \times 10^4$ $cm^{-3}s^{-1}$ in all simulations. See Table 2 for the summary of ammonia mixing ratio and the source of sulphuric acid monomer used for the ACDC simulations.

Synthetic concentration data for such neutral clusters were generated by the following method.

First, we computed the collision rates using the Eq. A3 from kinetic gas theory. Then, we used these values for the collision rates along with Eq. A4 and the Gibbs free energies computed from Eq. A5 to obtain the evaporation rates. Note that to compute the Gibbs free energies, we substituted the values for cluster formation enthalpies and entropies given by Olenius et al. (2013b) into Eq. A5. Additionally, we consider the losses on the CLOUD chamber walls which depend on the cluster size computed with Eq. A2 (Kürten et al., 2015) and a dilution loss of $S = 9.6 \times 10^{-5}$ $s^{-1}$. These values for the rates and losses were substituted into the ACDC algorithm (McGrath et al., 2012), which simulates the time evolution of molecular cluster concentrations. The ACDC code computes the first-order non-linear, ordinary differential system of cluster concentrations as given by Eq. A1. We then integrate the system produced by ACDC using the Fortran ordinary differential equation solver VODE (N. Brown et al., 1989). A detailed description of this strategy for solving the forward-problem of finding the cluster concentration rates from

Eq. A1 was published in McGrath et al. (2012). To reproduce the experimental conditions as realistically as possible, each simulation was initialized with non-zero concentration of ammonia monomer and no sulphuric acid. The source of sulphuric acid monomer was supplied at a constant rate as it was previously mentioned.

The above method we used for producing synthetic concentration rates is similar to the one described in Kupiainen-Määttä (2016). We note that unlike Kupiainen-Määttä (2016), in this paper, our particle system is considered at various temperatures.

Using the above algorithm, model configuration and parameters, we generated two data sets. First, time evolution of the concentrations $Y_i(t)$ is computed for time values less than the time at which the system has attained the steady state. The maximum time we run is 60 minutes from beginning of the simulation, in the above model configurations. In this case, it is assumed that the concentrations for all the clusters are measured under constant temperature with time resolution comprising 1.5 minutes, which comprises overall 41 time-dependent concentration data for each of the cluster types $i$ measured from beginning to the end of each simulation, before the system has attained a steady state.

Secondly, we solve for time-independent steady-state concentrations for all the cluster types for two temperatures comprising 278 K and 292 K. In both data configurations, the steady-state cluster concentrations are calculated as the average of the concentrations determined for time instances $t_1 := 50$ min and $t_2 := 60$ min. The measure of how close the system has reached to the steady state is monitored by a convergence parameter, which is the ratio of the concentrations at times $t_2$ and $t_1$, taken in each case for the cluster for which this ratio deviated most from unity, (Kupiainen-Määttä, 2016).

In both data settings, the simulation outputs are amended with the measurement errors sampled from a multivariate, non-correlated, Gaussian distribution, where the variance of the distribution depends on cluster type $i$, temperature $T$ and time instance $t$. While a simplification of noise characteristics of the real data obtained from a mass spectrometer, we impose that the standard deviation of the noise comprises 0.001% of the original concentration.

Note that apart from generation of synthetic data, we apply the ACDC as a kinetic model of cluster population in the MCMC simulations. The ACDC outputs are compared to the synthetic measurements and explained in Section 2.2.

**Table 1.** Neutral molecular clusters included into model system. The first column indicates the number of sulphuric acid molecules, the second column stands for the number of ammonia in the cluster.

| Number of $H_2SO_4$ molecules | Number of $NH_3$ molecules | Number of clusters |
| --- | --- | --- |
| 0 | 1 | 1 |
| 1 | 0-1 | 2 |
| 2 | 0-2 | 3 |
| 3 | 1-3 | 3 |
| 4 | 2-5 | 4 |
| 5 | 3-5 | 3 |

**Table 2.** Monomer concentrations used in simulations

| [H$_2$SO$_4$] monomer source | [NH$_3$] concentration |
|---|---|
| $6.3 \times 10^4$ cm$^{-3}$s$^{-1}$ | 5 ppt |
| $6.3 \times 10^4$ cm$^{-3}$s$^{-1}$ | 35 ppt |
| $6.3 \times 10^4$ cm$^{-3}$s$^{-1}$ | 100 ppt |
| $6.3 \times 10^4$ cm$^{-3}$s$^{-1}$ | 200 ppt |

## 2.2 Markov chain Monte-Carlo simulations

The evaporation rate coefficients $\gamma_{i+j \to i,j}$ appearing in the ACDC simulation of Equation A1 are treated as unknown parameters. Now we describe how we estimate the evaporation rates from the noisy synthetic data sets obtained by the method described in Section 2.1. We first give a general overview of the basic Metropolis algorithm (Metropolis et al., 1953), then describe a modification of the algorithm we implemented in this study, and finally, in Section 2.2.3 we apply this general framework to each of our study cases. Our purpose is to determine all the parameter sets that reproduce the synthetic data within their noise level (which is known). We do this using Markov Chain Monte Carlo (MCMC) sampling.

The objective of MCMC in parameter estimation is to identify possible parameter values which yield the best fit with the experimental data. Unlike optimization algorithms that produce one best combination of parameter values, in the MCMC procedure all the most-probable combinations of parameter values are estimated given the data. To obtain these combinations, the values of parameters are generated and stored into the MCMC "chain". The MCMC chain will converge to the distribution containing all the most-likely combinations of parameter values as a number of sampled parameter sets (i.e., the chain length) increases. The distribution formed from the chain approximates a posterior probability density function which gives the likelihood of observing each of the parameters given the concentration data.

### 2.2.1 The Metropolis algorithm

First, a prior distribution for the parameter values $\boldsymbol{\theta}$ (represented in array form) is chosen and set to be the proposed "true" distribution from which possible parameters are sampled. The prior is typically selected based on the previous knowledge of the parameter values. Then an initial guess for parameter values (denoted as $\theta_0$ or $\theta_{\text{old}}$) is selected from the prior distribution.

Starting from the initial guess, the algorithm samples candidate parameter values (denoted as $\theta_{\text{new}}$) from a proposal distribution centred at the previous point (denoted as $q(\theta_{\text{old}}, \theta_{\text{new}})$). The proposal density $q(\theta_{\text{old}}, \theta_{\text{new}})$ is symmetric, which means that the probability of step taken from the 'old' $\theta_{\text{old}}$ to the 'new' point $\theta_{\text{new}}$ is same as the probability of the reverse step ($q(\theta_{\text{old}}, \theta_{\text{new}}) = q(\theta_{\text{new}}, \theta_{\text{old}})$).

Then the candidate point $\boldsymbol{\theta}_{\text{new}}$ is either accepted or rejected, according to the least-squares fit of the output to the data, which measures the difference between the modelled $\mathbf{Y}_{\text{mod}}$ and measured $\mathbf{Y}_{\text{exp}}$ cluster concentrations:

$$F(\boldsymbol{\theta}_{\text{new}}) = \sum_{i=1}^{N} \frac{(Y_{\text{exp,i}} - Y_{\text{mod,i}}(\boldsymbol{\theta}_{\text{new}}))^2}{\sigma_i^2}, \tag{1}$$

where $N$ stands for the number of measurements in synthetic data. We consider two sets of synthetic cluster concentrations: time-dependent, measured at $T = 278$ K and steady-state, measured for two temperatures (at $T = 278$ K and $T = 292$ K), as explained in Section 2.1. For the time-dependent synthetic data $N = N_C \times N_t$, where $N_C = 16$ stands for the number of cluster types included into simulations, while $N_t = 41$ stands for the number of time-step measurements available for each of the cluster types. For the second data set, $N = N_C \times N_T$, where $N_T = 2$ denotes the number of experiments conducted at different temperatures. In the formula above we scale the squared residuals by the measurement error variance $\sigma_i^2$ to avoid overfitting to the larger concentration values. The error variance $\sigma_i^2$ is matched depending on cluster type, time instance and temperature. See A2 for more details.

At each iteration of the Metropolis algorithm, the value $F(\boldsymbol{\theta}_{\text{new}})$ is compared to the least-square sum from the previous step $F(\boldsymbol{\theta}_{\text{old}})$. If the new value is lower (i.e., the candidate parameters fit the data at least as good as the the old values), then the step is accepted. In the opposite case, when $F(\boldsymbol{\theta}_{\text{new}}) > F(\boldsymbol{\theta}_{\text{old}})$, the point will be accepted with the probability

$$\alpha_{\text{acc}} = \exp\left[-\frac{1}{2}(F(\boldsymbol{\theta}_{\text{new}}) - F(\boldsymbol{\theta}_{\text{old}}))\right]. \tag{2}$$

If the candidate point is accepted, the parameter combination $\boldsymbol{\theta}_{\text{new}}$ is added to the chain, in the opposite case the old value is replicated in the chain. Finally, the value $F(\boldsymbol{\theta}_{\text{old}})$ is replaced with $F(\boldsymbol{\theta}_{\text{new}})$ and saved for the next iteration.

In this paper we employ a variant of the Metropolis algorithm which is more efficient at parameter sampling when the parameter space is large (Haario et al., 2006). This variant is called the Delayed Rejection Adaptive Metropolis (DRAM), introduced in Haario et al. (2006). We briefly explain our approach below.

### 2.2.2 The DRAM algorithm

Similar to the basic Metroplois algorithm, the DRAM is initialized with a chosen prior distribution and initial guess for parameter values.

We make our initial guess $\boldsymbol{\theta} = \boldsymbol{\theta}_{old}$, where $\boldsymbol{\theta}_{old}$ is the flat distribution which obeys the estimates in Tabs. 3-4. The limits are explained in Section 2.2.3. We also assume that the conditional probability distributions for the parameters given the concentration data are of Gaussian type.

Once initialized, the following iterative steps take place. From the likelihood probability distribution for $\boldsymbol{\theta}_{old}$, a new candidate for the unknown parameter values, $\boldsymbol{\theta}_{new}$, is sampled using the proposed Gaussian likelihood distribution. We then use the algorithm in Section 2.1 to obtain concentration outputs from the evaporation rates $\boldsymbol{\theta}_{new}$. In the first stage of DRAM, we chose to accept the new proposed values $\boldsymbol{\theta}_{new}$ with probability

$$p_{\text{acc}}(\boldsymbol{\theta}_{\text{old}}, \boldsymbol{\theta}_{\text{new}}) = \min\left\{1, \frac{p(\mathbf{Y}_{\text{exp}}|\boldsymbol{\theta}_{\text{new}})}{p(\mathbf{Y}_{\text{exp}}|\boldsymbol{\theta}_{\text{old}})}\right\}, \tag{3}$$

where $\mathbf{Y}_{\exp}$ is the array of synthetic cluster concentration data, and $\mathrm{p}(\mathbf{Y}_{\exp}|\boldsymbol{\theta}_{\mathrm{old}})$, $p(\mathbf{Y}_{exp}|\boldsymbol{\theta}_{new})$ denote the likelihood (conditional) probabilities for the old and new parameter values, respectively. These likelihood probabilities quantify how closely the kinetic model with parameters $\boldsymbol{\theta}$ reproduce the data, as they depend on the sum of squared residuals (see Eqs. A6 and 1) between the given data and the concentrations obtained from the ACDC and VODE simulations with parameters $\boldsymbol{\theta}_{old}$ and $\boldsymbol{\theta}_{new}$, respectively. This relationship is explained further in Appendix A1.

In DRAM we allow for partial modification of the proposed parameters (the "delayed rejection" component of DRAM). This second stage of sampling improves the computational time needed to obtain an estimate for $\boldsymbol{\theta}$; it is performed as follows. If the proposed $\boldsymbol{\theta}_{\mathrm{new}}$ is rejected, a nearby proposal is created, $\boldsymbol{\theta}_{\mathrm{new2}}$. We accept this second proposal keeping in mind the rejection probability of the first, according to

$$\mathrm{p}_{\mathrm{acc2}} = \min\left\{1, \frac{\mathrm{p}(\mathbf{Y}_{\exp}|\boldsymbol{\theta}_{\mathrm{new}})\mathrm{p}(\mathbf{Y}_{\exp}|\boldsymbol{\theta}_{\mathrm{new}},\boldsymbol{\theta}_{\mathrm{new2}})[1-\mathrm{p}_{\mathrm{acc}}(\boldsymbol{\theta}_{\mathrm{new}},\boldsymbol{\theta}_{\mathrm{new2}})]}{\mathrm{p}(\mathbf{Y}_{\exp}|\boldsymbol{\theta}_{\mathrm{old}})\mathrm{p}(\mathbf{Y}_{\exp}|\boldsymbol{\theta}_{\mathrm{old}},\boldsymbol{\theta}_{\mathrm{new}})[1-\mathrm{p}_{\mathrm{acc}}(\boldsymbol{\theta}_{\mathrm{old}},\boldsymbol{\theta}_{\mathrm{new}})]}\right\}. \tag{4}$$

At the start of the MCMC simulations, the proposal covariances for both stages are initialized using arbitrary diagonal matrices with equal variances. It is assumed that the proposals of the form $\mathrm{p}(\mathbf{Y}_{\exp}|\cdot)$ and $\mathrm{p}(\mathbf{Y}_{\exp}|\cdot,\cdot)$ are Gaussian. They are updated at each successive iteration of the MCMC algorithm to improve the mixing of the chains.

The first-stage proposal covariance is recomputed via the Adaptive Metropolis (AM) procedure (Haario et al., 2001). Let $\mathrm{d}$ be the dimension of the parameter space, and $\{\mathbf{X_0},\ldots,\mathbf{X_n}\} \subset \mathbb{R}^d$ be a set of d-dimensional vectors containing the sampled values of free parameters. Then the first-stage proposal is centred at the current position of the Markov chain $\mathbf{X_n}$, whereas the corresponding proposal covariance $\mathbf{C}_{\mathrm{n}}^1$ is updated using the path of the previously sampled MCMC chain:

$$\mathbf{C}_{\mathrm{n}}^1 = \begin{cases} \mathbf{C}_0, & n \le n_0 \\ s_{\mathrm{d}}\mathrm{Cov}(\mathbf{X_0},\ldots,\mathbf{X_{n-1}}), & \mathrm{n} > \mathrm{n}_0, \end{cases} \tag{5}$$

where $\mathrm{C}_0$ is the initial covariance assigned at the beginning of the MCMC runs, $n_0$ stands for the length of the initial non-adaptation period, $s_d = 2.4/d$ is the scaling parameter, and $\mathrm{Cov}(\mathbf{X_0},\ldots,\mathbf{X_{n-1}})$ is the empirical covariance matrix for the vectors $\mathbf{X_0},\ldots,\mathbf{X_{n-1}}$:

$$\mathrm{Cov}(\mathbf{X_0},\ldots,\mathbf{X_{n-1}}) = \frac{1}{\mathrm{n}-1}\left(\sum_{\mathrm{i}=0}^{\mathrm{n}-1}\mathbf{X}_{\mathrm{i}}\mathbf{X}_{\mathrm{i}}^{\mathrm{T}} - \mathrm{n}\overline{\mathbf{X}}_{\mathrm{n}-1}\overline{\mathbf{X}}_{\mathrm{n}-1}^{\mathrm{T}},\right), \tag{6}$$

where $\overline{\mathbf{X}}_{n-1}^{T} = \frac{1}{n}\sum_{i=0}^{n-1}\mathbf{X}_i$ and $\mathbf{X}_i \in \mathbb{R}^d$ are column vectors. In our study and all runs therein, we set $n_0$ to be 100 iterations.

Simultaneously, the second-stage proposal covariance is computed as a scaled version of the first-stage proposal covariance:

$$\mathbf{C}_{\mathrm{n}}^2 = \gamma\mathbf{C}_{\mathrm{n}}^1, \tag{7}$$

with the scaling factor $\gamma = 5$ borrowed from Haario et al. (2006). This value was chosen to increase the acceptance at the second stage.

Then, if both $\boldsymbol{\theta}_{old}$ and $\boldsymbol{\theta}_{new}$ are rejected at this stage, a new parameter candidate is sampled and the process is repeated. If the parameter candidate is accepted, the Markov chain is advanced one step and sampling as above is repeated. The process stops once the chain length is exhausted.

Parameter estimation is conducted using the **'mcmcstat'** toolbox implemented for FORTRAN (Haario et al., 2001, 2006). See the description and the examples of usage on the web page helios.fmi.fi/~lainema/.

### 2.2.3 Overview of the MCMC runs

In our implementation of the DRAM algorithm, we impose upper and lower limits for the parameter values. We add such domain restrictions to exclude unphysical estimates for our parameters. These restrictions are encoded in our prior distribution, which we set to be a combination of so-called "flat priors", which are distributions that are proportional to a constant, (see Tabs. 3-4).

We emphasize that there are currently *no theoretical principles or experimental results which indicate possible restrictions for even the order of magnitude of the evaporation rates.* However, we assume that the evaporation rates with orders of magnitude less than $10^{-10}\text{s}^{-1}$ are irrelevant in practise, since such an evaporation event is highly improbable, and it is very likely that instead the cluster will grow further by collisions. Similarly, when the evaporation rate is of the order of magnitude more than $10^{+10}\text{s}^{-1}$, it is reasonable to expect that the cluster will most certainly evaporate before it has a chance to grow further. With these assumptions, the prior distribution of the evaporation rates spans over several orders of magnitude, and the base 10 logarithm of evaporation rates was sampled from the range of -12 to 12.

Next, we justify the limits selected for data setting 2, where we sample thermodynamic parameters. For the formation enthalpies an upper limit of 0 kcal/mol is chosen by the fact that a positive $\Delta$H would mean an absence of attractive interactions in the molecular cluster, which is physically incorrect for polar, H-bonding molecules such as $H_2SO_4$ and $NH_3$. For the lower limit (-400 kcal/mol) we mean that on average each $H_2SO_4$ is bound substantially stronger than in the $HSO_4^- * H_2SO_4$ cluster, for which the most recent computational studies indicate a binding enthalpy roughly around -40 kcal/mol, (Elm et al., 2013; Elm and Kristensen, 2017). Another motivation for the prior distribution selected for the cluster formation enthalpies comes from the fact that the largest cluster included into the system has 5 $H_2SO_4$ and 5 $NH_3$, so 10 molecules, and -400 kcal/mol would give an enthalpy of -40 kcal/mol per molecule, which 1) corresponds to the strongest known cluster in the system and 2) which implies that the evaporation rate is zero for all purposes of measurement (Kurtén, 2007).

Next, we set the upper limit for the formation entropies to 0 cal/K/mol, since molecule clustering must have a negative $\Delta$S, as the number of gas molecules is reduced (and translational and rotational degrees of freedom are converted into much more constrained vibrational degrees of freedom). For the lower limit of -400 cal/K/mol, we state that the typical per-molecule $\Delta$S for clustering is around -30 cal/K/mol, with a typical variation of up to +-10 cal/K/mol (Kürten, 2019). So for the largest clusters the upper limit corresponds to a per-molecule $\Delta$S of -40 cal/K/mol. In this situation, all the new vibrational degrees of freedom formed in the product clusters are quite rigid, i.e. have very low entropy (Kurtén, 2007).

An outline of the sampling procedure is illustrated in Figure 1 below.

**Table 3.** Domain limitations for two data settings under consideration imposed to exclude non-physical parameters in parameter estimation procedure.

| Data settings | Estimated parameters | Minimal value | Maximal value |
|---|---|---|---|
| Data setting 1 | Base 10 logarithms of evaporation rates (in $s^{-1}$) | -12 | 12 |
| Data setting 2 | Cluster formation enthalpies (kcal $mol^{-1}$) and | -400 | 0 |
| | entropies (cal $K^{-1}$ $mol^{-1}$) | -400 | 0 |

**Table 4.** Additional domain limitations for the data setting 2 from Table 3 (estimation of thermodynamic data), where the cluster formation enthalpy of the $i$-th cluster is denoted by $\Delta H_i$ and the symbols $A$ and $N$ stand for ammonia and sulphuric acid, respectively.

| | |
|---|---|
| $\Delta H_{2A} > \Delta H_{2A1N}$ | $\Delta H_{3A2N} > \Delta H_{4A2N}$ |
| $\Delta H_{1A1N} > \Delta H_{2A1N}$ | $\Delta H_{4A2N} > \Delta H_{4A3N}$ |
| $\Delta H_{2A1N} > \Delta H_{3A1N}$ | $\Delta H_{4A3N} > \Delta H_{4A4N}$ |
| $\Delta H_{2A2N} > \Delta H_{3A2N}$ | $\Delta H_{4A4N} > \Delta H_{5A5N}$ |
| $\Delta H_{3A1N} > \Delta H_{3A2N}$ | $\Delta H_{4A4N} > \Delta H_{4A5N}$ |

We next explicitly describe what synthetic data ($\mathbf{Y}_{exp}$) and parameters ($\boldsymbol{\theta}$) which give the acceptance probability in Equation 3 represent in the two study cases.

In the first study, the free parameters $\boldsymbol{\theta}$ represent the evaporation rates. The data $\mathbf{Y}_{exp}$ is either the time-independent steady-state or transient cluster concentrations measured at temperature 278 K.

In the second study, we use Eq. A4 and A5 to express the evaporation rates as functions of thermodynamic data, parametrized by temperature:

$$\gamma_{i+j \to i,j} = f(T, \{\Delta H_k, \Delta S_k\}_{k \in \{i+j,i,j\}}). \tag{8}$$

In Eq. 8, we set $T = 278$ K or $T = 292$ K. We emphasize that the rates $\gamma_{i+j \to i,j}$ now depend on temperature and six parameters: the cluster formation enthalpy $\Delta H_{i+j}$ and entropy $\Delta S_{i+j}$ of the evaporating cluster $i+j$, and the formation enthalpies $\Delta H_i, \Delta H_j$ and entropies $\Delta S_i, \Delta S_j$ of the clusters $i$ and $j$ respectively. In this setting $\boldsymbol{\theta}$ represents the array of quantities $\Delta H_{i+j}, \Delta S_{i+j}, \Delta H_i, \Delta H_j, \Delta S_i, \Delta S_j$ with $i + j \in \{1, 2, \ldots, 16\}$.

At either temperature $T = 278$ K or $T = 292$ K, the smaller clusters for certain combinations of ammonia and sulphuric acid may arise from the evaporation of several larger clusters. This implies that several of the pairs $\Delta H_i, \Delta S_i$ appear in expression 8 for the evaporation rates of different cluster types. Additionally, the Gibbs formation free energies of monomers are fixed to be zero, and their associated enthalpies and entropies do not vary in our simulations. This imposes additional constraints

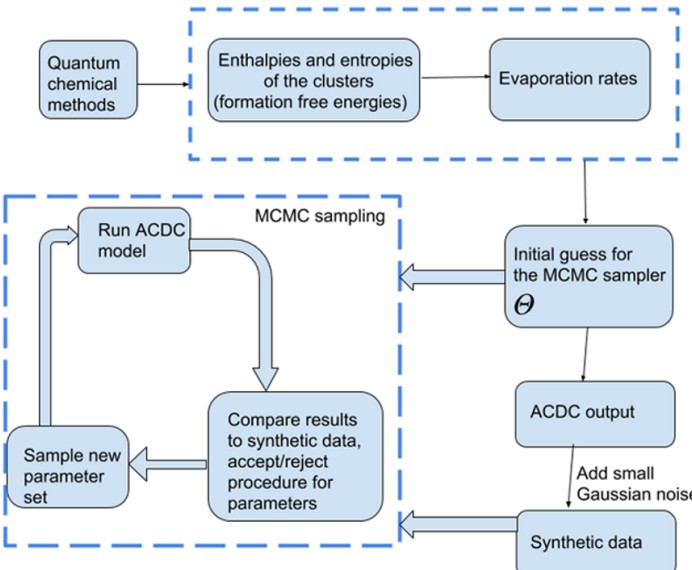

**Figure 1.** Schematic representation of the study methods.

on possible parameter values. One can calculate that of the 39 evaporations that are involved in the dynamics of the neutral

cluster system under consideration, only 28 distinct entropy and enthalpy values appear. Consequently, in this case the number

of free parameters has been reduced from 39 to 28. This information is summarized in Table 3. Moreover, from this table one

can see that the entropy and enthalpy values lie within two orders of magnitude. This feature of the cluster formation entropies

and enthalpies has the effect of reducing the *stiffness* of the differential system in  Equation A1 (computed via ACDC) which

allows for easier integration via VODE.

For the setting above, the data $\mathbf{Y}_{exp}$ are the time-independent steady-state cluster concentrations measured at temperature

K or 292 K. We note that several experiments conducted at different temperatures are needed to obtain state information

concerning the specific evaporation rate associated with each temperature level (Soncini, 2014). In this work we consider two

temperatures, which is one such minimal configuration that contains information sufficient for determination of thermodynamic

data. Similar approaches were applied for the inverse problem of chemical kinetics modelled by the Arrhenius equation, where

chemical reaction rates are temperature dependent (Vahteristo et al., 2008).

    Note that to create a reliable sample from the underlying parameter distribution, the length of the MCMC chain must be

"large enough" in an appropriate sense (Haario et al., 1999, 2001), that is, many different parameter combinations must be

tested. We remark here that in both our studies, the MCMC chain length typically comprised of 3 million samples. The MCMC

acceptance probabilities (defined below) in each of the cases were about 88.0%, which is a typical level of acceptance since

the "forward" ACDC model (in which the evaporation and collision rates are known) is deterministic.

In all simulations of the algorithm given in the previous section, the sets of parameters which produce cluster concentrations within the allotted noise level of the data are kept in the chain. Specifically, the sampled parameters of the posterior distribution represent the model evaluations which produce values within the noise level of 0.001% of the data concentrations for each of the respective cluster types.

## 3 RESULTS AND DISCUSSION

### 3.1 Identification of the evaporation rate coefficients from steady-state data

First, we generate synthetic steady-state data by the method in Section 2.1, for varying initial ammonia monomer concentrations, previously summarized in Table 2; the sulphuric acid monomer is supplied to the system at a constant rate comprising $6.3 \times 10^4$ s$^{-1}$ at the temperature T $= 278$ K. As an output, we obtain the concentrations for all cluster types considered (listed earlier in Table 1), measured when the system has attained the steady-state. A graphical representation of the data set is given above in Figure 2.

Next, from the steady-state data we determine the base 10 logarithms of the evaporation rate coefficients. Since the noise added to cluster concentrations results in a random bias towards an increase (or decrease) from the original values produced from the ACDC, the estimates of parameters derived from synthetic data are likely to be biased. In order to average the effects attributed to the random bias, we generated 3 sets of synthetic data by adding random increments to original concentration measurements. Utilizing these data sets, three independent MCMC runs were conducted, each run containing 3 million parameter samples. An example of one of the sampled chains is depicted in Figs. B1-B2. We omit the initial one million samples and plot the stationary[3] parts of the chains. As we observe from the plots in Figs. B1-B2, all the parameter chains for the evaporation rates have values bounded above by an upper limit which differs for different evaporation rates. However, only 15 out of 39 evaporation rates are limited from below (see subfigures labelled 1-5, 7, 10, 12, 16, 18, 22, 27, 31, 33 and 35 in Figs. B1-B2). This subset of evaporation parameters is comprised of the evaporation rates of monomers, with the exception of monomer evaporation rates for: $H_2SO_4$ from $(H_2SO_4)_5 (NH_3)_4$ and $(H_2SO_4)_5 (NH_3)_5$, and the evaporation rate of $NH_3$ from $(H_2SO_4)_5 (NH_3)_5$. These excluded parameters correspond to the evaporations of monomers from the largest and most stable clusters. Note that the estimated lower limits of monomer evaporations from all the clusters except for the most stable ones are far above the $10^{-10}$ s$^{-1}$ as defined for complete growth.

For each evaporation parameter, we calculate the one dimensional (that is, depending only on the evaporation rate) marginal posterior distribution as the position-wise average of the stationary parts of the three sampled chains. This procedure is needed to average the bias originating from random noise. The resulting distributions are given in Figs. 3-4. We use the maximum (also called the mode in the statistics literature) of the posterior marginal distribution function as our parameter estimate in the case when the marginal posterior distributions have precisely one maximum value. In the cases where we have multiple estimators, we provide a range for the evaporation rate values.

---

[3]Here stationary means that the probability of transitioning from the current state at position $j$ to the new state at position $j + 1$ is independent of $j$.

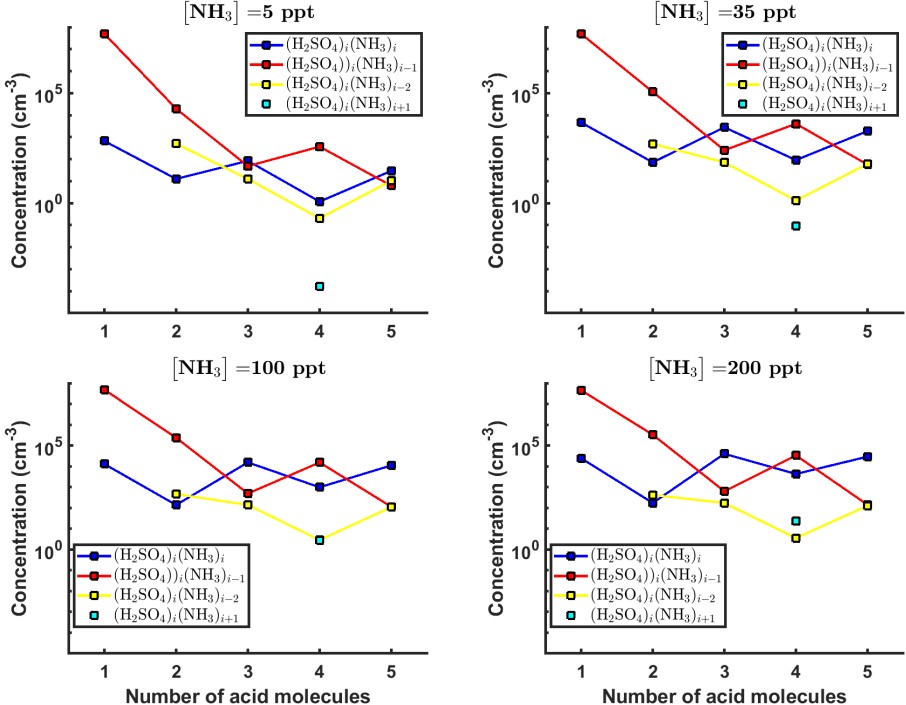

**Figure 2.** Steady-state cluster concentrations for the clusters containing sulphuric acid and a varying number of ammonia molecules as a function of the number of acid molecules for $[NH_3]$ concentrations comprising (a) 5 ppt, (b) 35 ppt, (c) 100 ppt and (d) 200 ppt at temperature T=278 K. The concentrations have been amended with multivariate non-correlated Gaussian noise with standard deviation comprising 0.001% of the original cluster concentration. The source of sulphuric acid monomers is $[H_2SO_4] = 6.3 \times 10^4$ s$^{-1}$ in each of the simulations.

All the evaporation rates larger than $10^{-3}$ s$^{-1}$ are well-identified (see subfigures labelled 1, 2, 4, 5, 7, 10, 12, 16, 18, 22, 27, 31 and 35 in Figs. 3- 4), in the sense that their estimated variances are well within our accepted error range of less then one order of magnitude. The estimates for the remaining evaporation rates can take values within ranges spanning several orders of magnitude and are thus uncertain. Also, notice that most of the marginal posterior distributions are non-uniform, except for the evaporation rate of $(H_2SO_4)_2(NH_3)_2$ from $(H_2SO_4)_5(NH_3)_5$. In five cases (refer to subfigures labelled 6, 21, 28, 32 and 36 in Figs. 3- 4), the estimated parameter values are not unique; that is the marginal posterior distributions feature multiple modes. The results of our parameter estimation are summarized in Tabs. C1- C2 and in subfigures labelled (a) and (b) in Figure 5.

The pairwise marginal posterior distributions for the estimated evaporation rates are illustrated in Figs. B3-B6. From these plots one can see that the majority of parameters are not correlated. However, the evaporation of monomers from $(H_2SO_4)_5NH_3$, $(H_2SO_4)_3(NH_3)_2$ and $(H_2SO_4)_5(NH_3)_4$ display non-linear inverse correlations. This implies that either $H_2SO_4$ rarely evaporates (at the rate less then $10^{-4}$ s$^{-1}$) and that $NH_3$ evaporates often, or the evaporation rates of $H_2SO_4$

and $NH_3$ are of comparable magnitude in these cases. Additionally, it can be seen from the pairwise posteriors that most of the estimated parameters are highly uncertain. Therefore, we conclude that in the situation where we determine parameters from the synthetic steady-state data, parameter estimation is not unique.

From a mathematical perspective, the existence of multiple distinct parameter estimates indicates that the problem of recovering evaporation rates from the synthetic steady-state concentration data is ill-posed. In these situations, one seeks to regularize the problem; that is, add more data or information to the model to reduce the number of possible estimates.

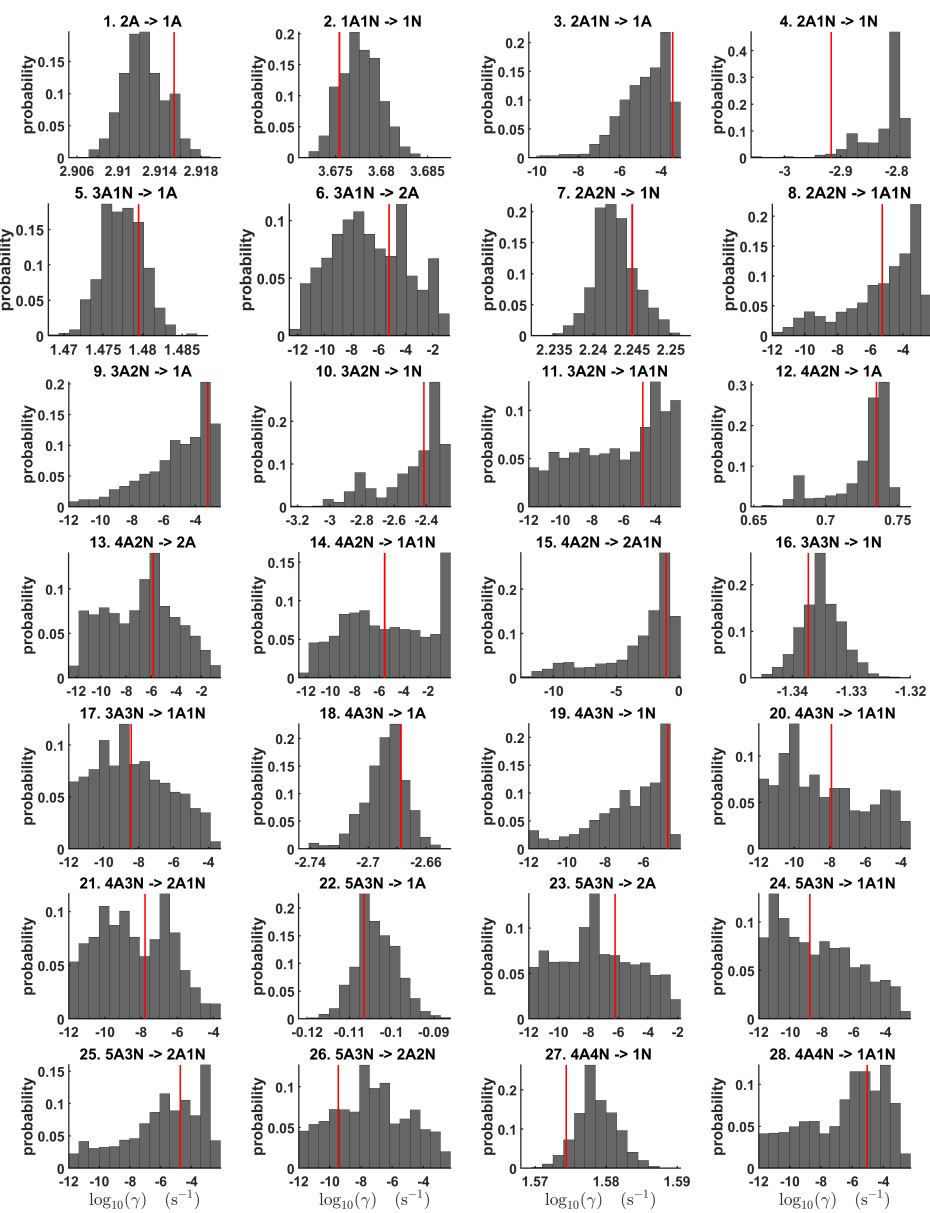

**Figure 3.** One-dimensional marginal posterior distributions (for parameter indexes ranging from 1 to 28) of the base 10 logarithm of the evaporation rates (units given in $s^{-1}$) determined from steady-state cluster concentration measurements at the temperature 278 K. Red lines denote the baseline values from Ortega et al. (2012) used to generate the synthetic data. In reactions "A" stands for $H_2SO_4$ and "N" for $NH_3$.

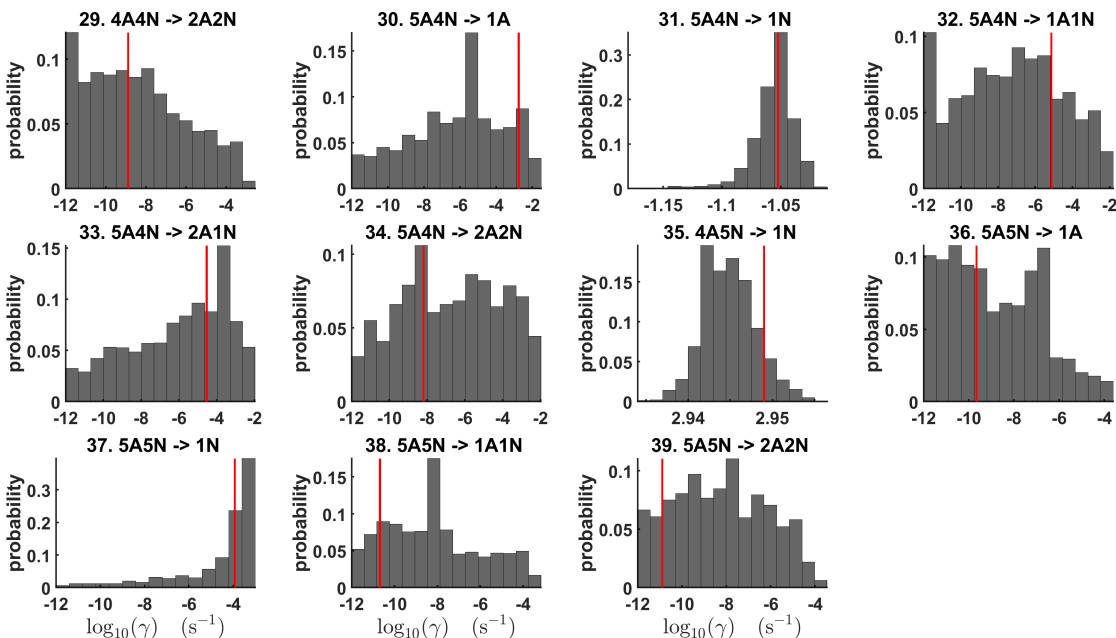

**Figure 4.** One-dimensional marginal posterior distributions (for parameter indexes ranging from 29 to 39) of the base 10 logarithm of the evaporation rates (units given in $s^{-1}$) determined from steady-state cluster concentration measurements at the temperature 278 K. Red lines denote the baseline values from Ortega et al. (2012) used to generate the synthetic data. In reactions "A" stands for $H_2SO_4$ and "N" for $NH_3$.

### 3.2 Identification of the evaporation rate coefficients from transient data

First, we extend the synthetic measurement data from steady state concentrations to transient concentrations. The data set for transient cluster concentrations at one temperature is larger than the data set for steady-state cluster concentrations at one temperature, as the transient data contains the concentration values at multiple times instances. Also the transient data contain information about the slope of the concentrations changing with time (see C1.), which contributes to quantification of the molecular-scale processes (such as collisions and evaporations). We thus expect that this larger data set will reduce the dimension of the solution space for the evaporation rates. Indeed, we will show that this is the case. We generate a synthetic transient cluster concentration data set using the method in Section 2.1. The time resolution of our new synthetic data set is 1.5 minutes, which results in 656 total concentration measurements for all the cluster type measured for four different ammonia concentrations. These data sets are illustrated in C1.

From this transient cluster concentration data set, we then conduct analogous MCMC runs (as described in Section 2.2). As in the steady-state setting, we conduct three independent MCMC runs to determine the base 10 logarithms of the evaporation rates. One of these runs is presented in Figs. C2-C3. Again, we omit the first one million samples, which are the samples before the chains have obtained their stationary distributions.

It is shown in Figs. C2-C3, that all the chains have the upper limits. Most of the chains are bounded from below, with five exceptions. Specifically, the evaporation rates of $(H_2SO_4)_2(NH_3)_2$ from $(H_2SO_4)_4(NH_3)_4$ and $(H_2SO_4)_5(NH_3)_3$, the evaporation rates of $H_2SO_4$, $H_2SO_4NH_3$ and $(H_2SO_4)_2(NH_3)_2$ from $(H_2SO_4)_5(NH_3)_5$ have arbitrarily large magnitude.

We examine the one-dimensional marginal posterior distributions for the estimated parameters in Figs. 6-7. From these plots, one sees that most of the estimates are close to the baseline values used for generation of the synthetic data. However, the estimated evaporation parameters still feature substantial uncertainties, as their marginal posterior distributions span several orders of magnitude (see subfigures 6, 8, 9, 11, 13, 14, 17, 21, 23-26, 30, 32-34, 37-39 in Figs. 6-7). Three parameters (subfigures 20, 29 and 36 in Figs. 6-7) have multimodal marginal posterior distributions. We also note that the evaporation rate of $(H_2SO_4)_2(NH_3)_2$ from $(H_2SO_4)_5(NH_3)_3$ (which corresponds to subfigure 26) has a uniform posterior distribution. Further, we can only specify that the upper limits for the evaporation rates depicted in subfigures 20 and 36 are less than $1.96 \times 10^{-5}$ s$^{-1}$. However, given the reliable upper estimates, the evaporation processes $(H_2SO_4)_4(NH_3)_3 \rightarrow (H_2SO_4)_4(NH_3)_2 + NH_3$ and $(H_2SO_4)_5(NH_3)_5 \rightarrow (H_2SO_4)_4(NH_3)_5 + H_2SO_4$ can be neglected, as they are relatively slow when compared with the other competing processes.

Pairwise marginal posterior distributions for the evaporation rates are plotted in Figs. C4-C8. Notice that the evaporation rates of monomers for the cluster $(H_2SO_4)_2NH_3$ display strong inverse linear relationship, which is indicated by the pairwise marginal posterior distribution of the coefficients $(H_2SO_4)_2NH_3 \rightarrow (H_2SO_4)_2 + NH_3$ and $(H_2SO_4)_2NH_3 \rightarrow H_2SO_4NH_3 + H_2SO_4$, (see Figure C4). Also, the estimated rate coefficients $(H_2SO_4)_2 \rightarrow H_2SO_4 + H_2SO_4$ and $H_2SO_4NH_3 \rightarrow H_2SO_4 + NH_3$ exhibit linear correlation. Additionally, the uncertainties in all the correlated parameters are relatively small (less then an order of magnitude). We also remark that from these plots one can see that most of the evaporation rates do not display any substantial correlations.

In Tabs. C1-C2 we summarize the results of parameter estimation for the above-discussed two data settings. Note that the estimated upper limits for some of the small evaporation rates (less than $10^{-5}$ s$^{-1}$) determined from the steady-state data can be as large as $1.55 \times 10^{-2}$ s$^{-1}$. This is a poor estimate, since the uncertainties in the synthetic data are small. For example, see the results for parameters shown in subfigures 32 and 34 of Figure 7. In these cases the identification has improved when we extended the data set with time-dependent measurements. Overall one observes that the transient data enabled us to determine the lower bounds for most of the parameters, with the exception of those parameters shown in subfigures numbered 26 and 29. Moreover, the additional time dependent data enabled us to reduce the uncertainties in the estimates of parameters in subfigures 15, 19 and 37. As a result, with the aid of time-dependent data we have improved the estimates of minimal and maximal values for the evaporation rate parameters (see comparison of the 95 % confidence intervals plotted in Figure 5).

In the case of the steady-state cluster concentrations we include only one value for each of the 16 cluster types considered in the study, which were taken when the system has attained a steady state (at the end of the ACDC simulation). The transient data contain the steady-state data as subset. Specifically, in this case we consider the concentrations measured when the system has attained the steady state together with the time-step concentration data measured from the starting point to the end of the ACDC simulation.

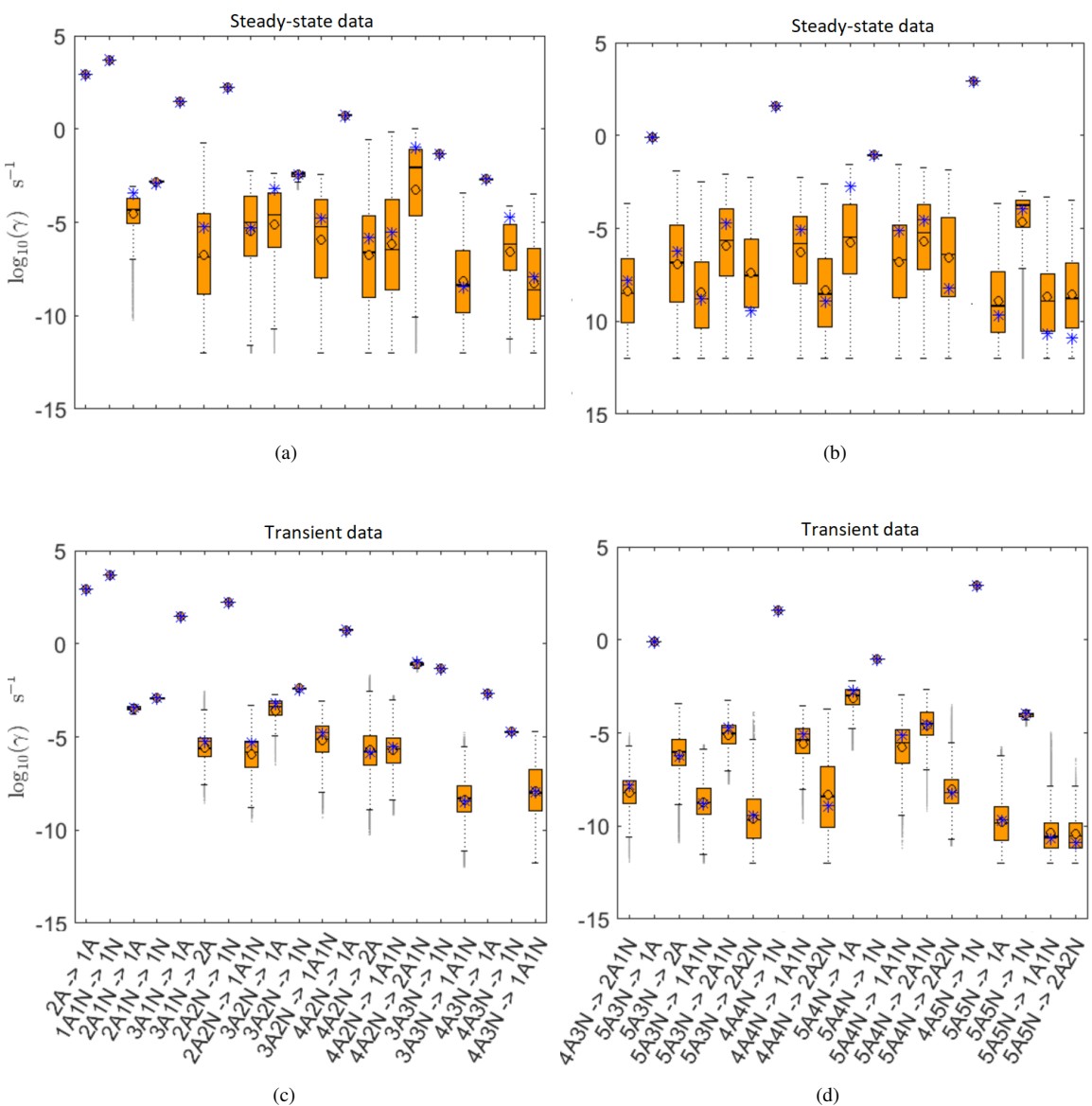

**Figure 5.** Comparison of 95 % confidence intervals (orange box plots) of base 10 logarithms of the evaporation rates determined from (a)-(b) steady-state and (c)-(d) time-dependent synthetic data measured at temperature 278 K. In reactions "A" stands for $H_2SO_4$ and "N" for $NH_3$. Here blue asterisks denote the baseline values used for creating the synthetic data (borrowed from Ortega et al. (2012)). Black circle and horizontal line markers indicate the mode and the mean value of the distribution, respectively.

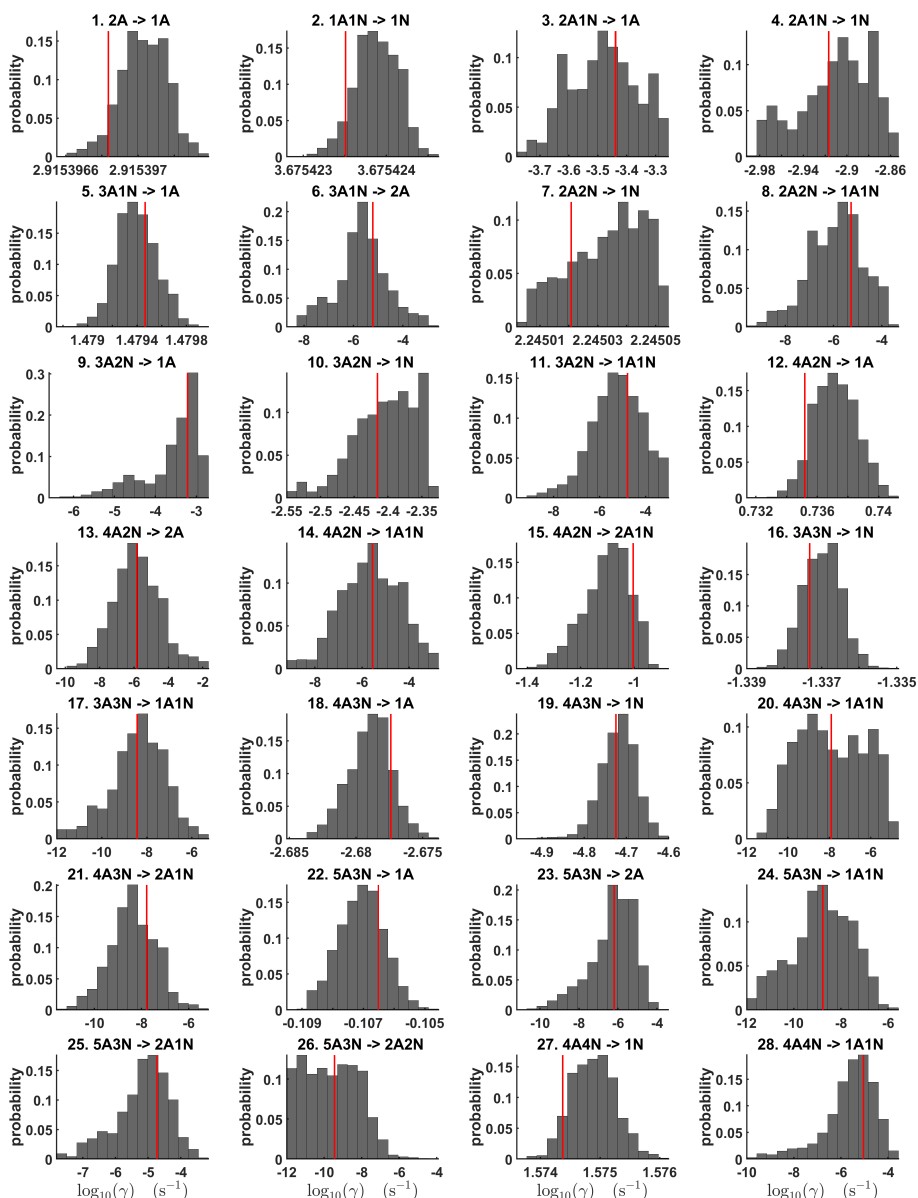

**Figure 6.** One-dimensional marginal posterior distributions (for parameter indexes ranging from 1 to 28) of the base 10 logarithm of the evaporation rates (units given in $s^{-1}$) determined from transient measurements of the cluster concentrations with time resolution comprising 1.5 minutes at the temperature 278 K. Red lines denote the baseline values from Ortega et al. (2012) used to generate the synthetic data. In reactions "A" stands for $H_2SO_4$ and "N" for $NH_3$.

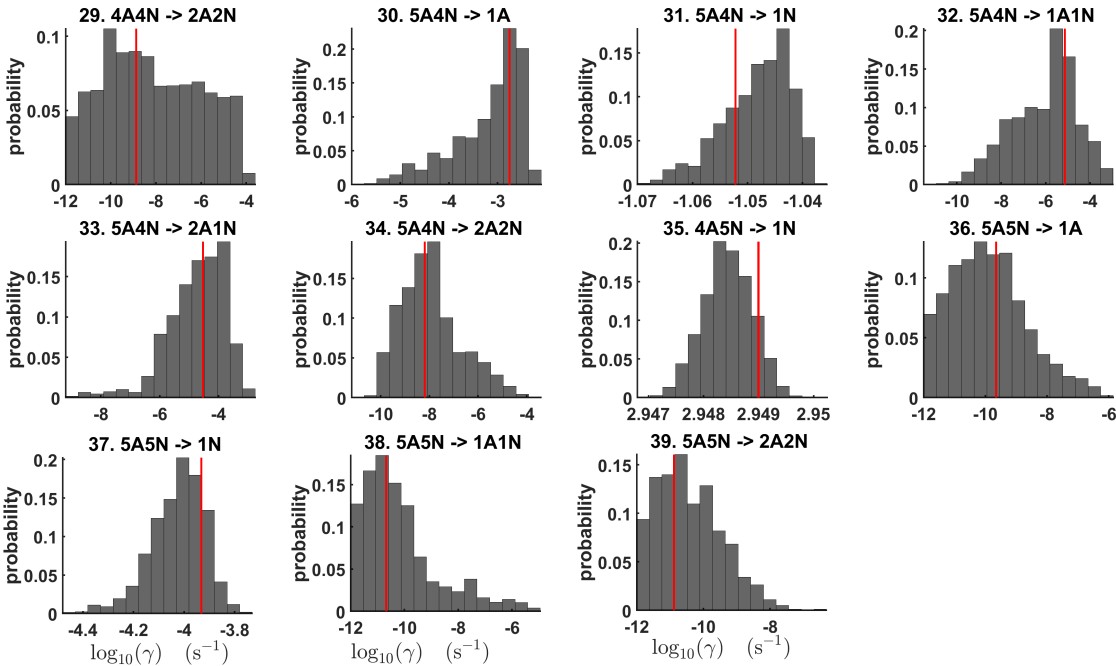

**Figure 7.** One-dimensional marginal posterior distributions (for parameter indexes ranging from 29 to 39) of the base 10 logarithm of the evaporation rates (units given in $s^{-1}$) determined from transient measurements of the cluster concentrations with time resolution comprising 1.5 minutes at the temperature 278 K. Red lines denote the baseline values from Ortega et al. (2012) used to generate the synthetic data. In reactions "A" stands for $H_2SO_4$ and "N" for $NH_3$.

## 3.3 Estimating thermodynamic data from steady-state concentration measurements

In this section we describe another method for regularizing our problem of estimating evaporation rates from steady-state concentration data. We will determine the cluster formation enthalpies and entropies from two sets of synthetic, steady-state cluster concentrations, now measured at two temperatures: 278 and 292 K. This data set is plotted in Figs. 2 and D1 for 278 K and 292 K, respectively.

We will demonstrate that reparameterization (in terms of thermodynamic data) plus the extended data set transforms our parameter estimation problem from an ill-posed problem to a well-posed one. We use synthetic steady-state cluster concentrations generated for two temperatures to recover the thermodynamic parameters. This is done to improve the identification by using the temperature dependence of the Gibbs free energies (and the evaporation rates).

For each temperature choice, we use the methods described in Section 2 to obtain synthetic steady-state cluster concentration data. We summarize this data in Table 2; the data sets are plotted in Figure 2 for 278 K and D1 for 292 K. Three MCMC runs were conducted to average the bias attributed to random noise added to the data, as discussed in the previous section. An example of one of the sampled chains is illustrated in Figure D2. It can be seen that all the chains are bounded, with the

exception of the formation enthalpy and entropy of the biggest cluster ($(H_2SO_4)_5(NH_3)_5$).

Next we consider the one-dimensional (depending on the particular cluster formation entropy or enthalpy parameters) marginal posterior distributions of free parameters built from the stationary parts of the three sampled chains merged together, see Figure 9. It can be seen that for all the clusters except $(H_2SO_4)_5(NH_3)_5$ the variance for the estimated formation enthalpies are less than $0.46\,\mathrm{kcal\,mol^{-1}}$, while the estimated formation entropies vary at most by $5.4\,\mathrm{cal\,K^{-1}mol^{-1}}$. The estimated free parameters together with the baseline quantum chemistry-based values from Ortega et al. (2012) used for generation of the synthetic data are summarized in Table D1.

Although the posterior distributions of sampled thermodynamic parameters for $(H_2SO_4)_5(NH_3)_5$ feature higher uncertainties in comparison to the corresponding posterior distributions identified for the smaller clusters, the evaporation rates for evaporations from $(H_2SO_4)_5(NH_3)_5$, as calculated from the aforementioned posterior distributions, have low variances, see Table D3.

Notice that the evaporation rates for all the molecular clusters calculated from a posterior distribution of sampled thermodynamic parameters for the temperature 278 K are close to the baseline values from Ortega et al. (2012) used for generation of the synthetic data and their variances are less than one order of magnitude, see Figs. D6-D7.

Additionally, strong correlations are observed between formation enthalpies (entropies) of the clusters containing same number of ammonia molecules larger then 2, except the case of $(H_2SO_4)_5(NH_3)_5$. Since our parameters are strongly correlated, we may alternatively consider just cluster formation enthalpies or the ratios of cluster formation entropies and enthalpies as our free parameters.

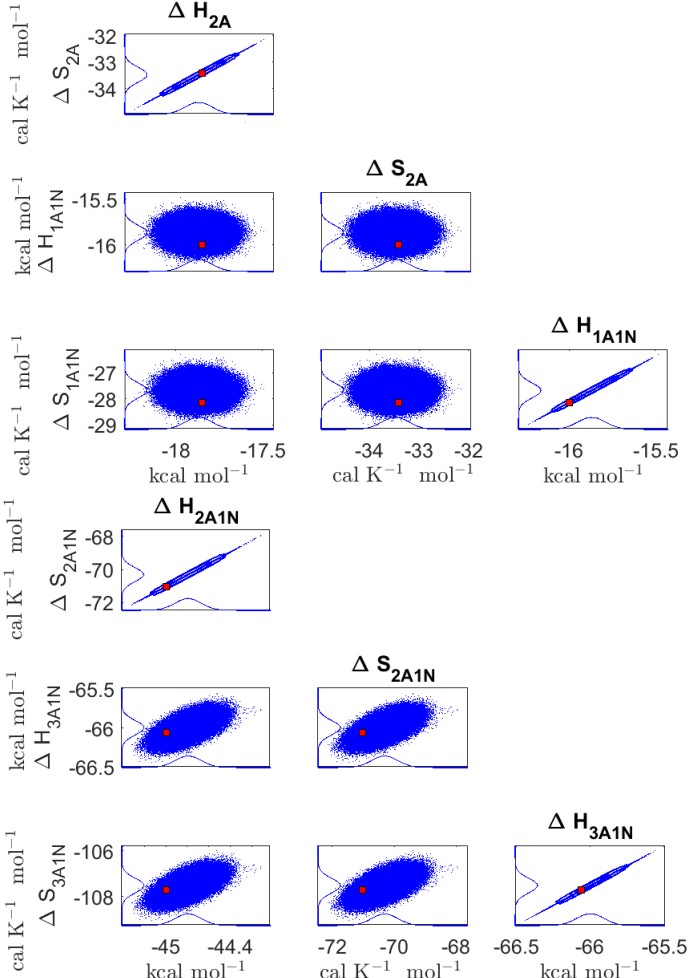

**Figure 8.** Pairwise marginal posterior distributions (for parameter indexes ranging from 1 to 8) of the cluster formation enthalpies and entropies determined from steady-state cluster concentration measurements at two temperatures T=278 K and T = 292 K. Red rectangles denote the baseline values from Ortega et al. (2012) used to generate the synthetic data. Here the symbols $\Delta H$ and $\Delta S$ stand for cluster formation enthalpies and entropies, respectively. Symbols "A", "N" denote $H_2SO_4$ and "$NH_3$", correspondingly.

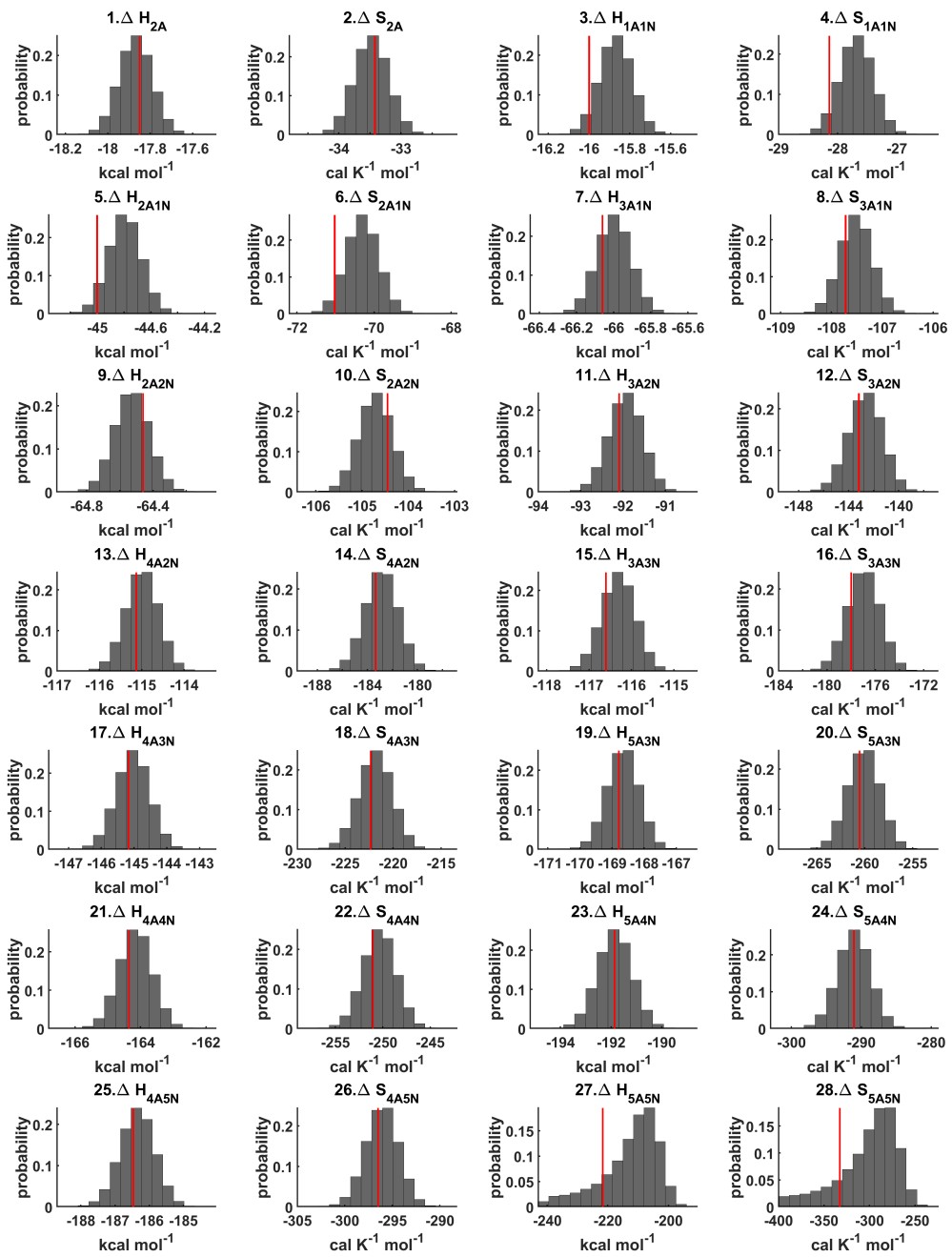

**Figure 9.** One-dimensional marginal posterior distributions of the cluster formation enthalpies (units given in kcal/mol) and entropies (units given in $\mathrm{cal\ K^{-1}\ mol^{-1}}$)) determined from steady-state cluster concentration measurements at two temperatures T=278 K and T = 292 K. Red lines denote the baseline values from Ortega et al. (2012) used to generate the synthetic data. Here the symbols $\Delta$H and $\Delta$S stand for cluster formation enthalpies and entropies, respectively. Symbols "A", "N" denote $H_2SO_4$ and "$NH_3$", correspondingly.

### 3.4 Comparison to previous evaporation rate determinations

The evaporation rates can be obtained either experimentally or computationally, when applying the Quantum Chemical (QC) methods, (Kürten, 2019). Experimental detection was conducted from the measurements in a flow tube (Hanson and Eisele, 2002; Jen et al., 2016; Hanson et al., 2017) and in the CLOUD chamber (Kurtén et al., 2007; Nadykto and Yu, 2007; Ortega et al., 2012; Elm and Kristensen, 2017; Yu et al., 2018). The summary of thermodynamic parameters obtained from different methods has previously been published in Kürten (2019). These parameters can be employed to calculated the evaporation rates at different temperatures.

In this study we determine the evaporation rates and thermodynamic data from measurements of cluster concentrations. Supplementary to the methodology presented in Kupiainen-Määttä (2016), our first method enables to determine parameters from the time-dependent cluster concentrations measured before the system has attained the steady state. The transient data improved the estimates for all the evaporation rates.

In the second method we identify thermodynamic parameters from the steady-state cluster concentrations measured at two different temperatures. This approach is similar to Kürten (2019), but our model takes into account all the possible evaporation processes. In Kürten (2019) the thermodynamic parameters had been determined from the New Particle Formation Rates (NPFs) measured at different temperatures. Instead of the NPFs, we employ the measurements of cluster concentrations. By so doing, we find the combination of data and fitted parameters which enables to determine the evaporation rates with the variances comprising less that one order of magnitude.

Although the transient data have improved the estimates, the temperature-dependent data have been demonstrated to yield the most accurate estimates of the evaporation rates, when we treat cluster formation enthalpies and entropies as free parameters.

### 3.5 Discussion and future work

The MCMC results are not specific for the simulation box considered in the present study, but rather general. This is supported by the fact that although the size of the system (the number of clusters included into simulations) has impact on the particle formation rates at high temperatures ($> 278$ K), the particle formation rates and cluster concentrations produced using different simulation boxes are qualitatively similar. Thus the changes of the ACDC outputs due to the difference in the simulation box does not change for MCMC parameter estimation results. In Besel et al. (2020) is was shown that the 5x5 simulation box (which is used for generation of the synthetic data) produces reasonable results with a good agreement with the measurements obtained from the CLOUD chamber experiment. Additionally, the boundary conditions for the outgrowing clusters (the choice of the clusters that are considered as formed particles) has only minor influence on the simulation results, given that the simulated system of clusters is defined in a reasonable way (Besel et al., 2020).

In general, the accuracy of the MCMC results increases when we include additional data. In particular, including more concentration data measured at different ammonia concentrations will yield better estimates for the evaporation rates. The sensitivity of the estimates to the number of ammonia concentrations will be considered in the future work. In the present study

we rather focus on the question which combination of estimated parameters and concentration data will produce an accurate estimates for the evaporation rate.

The data of steady-state concentration with two temperatures allowed us to apply two general principles of inverse problems/Bayesian estimation to the problem of estimating evaporation rates. First, the two temperature data set enabled us to reformulate the problem in a numerically effective way (in terms of enthalpy and entropy) that reduced the number of unknown parameters we sought to estimate. Second, the reformulated differential equation describing the time evolution of the concentrations was more numerically stable than the original expression (the stiffness of the equation was reduced in the reformulated form). This made our estimates for the rates less sensitive to small perturbations/errors.

In addition, the fact that the formation entropies and enthalpies were strongly correlated made them an effective parametrization. The strong inverse correlations have a physical explanation. Firstly, both formation enthalpy and entropy follow from the partition function of the molecular complex, and their functional forms are partly similar (Kurtén, 2007). Practically, if a cluster has really strong bonds between the molecules, then that means the formation enthalpy is very negative, and also the intermolecular vibrational frequencies corresponding in a broad sense to vibrations involving those bonds (note that these frequencies dominate the "variable part" of the formation entropy, as the entropy effect from the loss of translational and rotational degrees of freedom is almost a constant factor) are fairly high, meaning that the entropy loss in forming the cluster is large. So if the formation enthalpy is very negative so is also the formation entropy. Conversely, if the cluster is only quite weakly bound, the formation enthalpy is only slightly negative, and the intermolecular frequencies can be very low, leading to a less negative (though still negative of course) formation entropy (Kurtén, 2007).

Note that experimental data can differ from the synthetic data in the sense that they contain noise which originate from measurement instruments and uncertainties associated with experimental conditions (e.g., in CLOUD chamber experiments). Treating the noise inherent for experimental data will be the topic of our future studies.

## 4   Conclusions

We applied a Bayesian parameter estimation using a Markov chain Monte Carlo (MCMC) algorithm to identify cluster evaporation/fragmentation rates from known cluster distribution data and known cluster collision rates. We used Atmospheric Cluster Dynamic Code (ACDC) with quantum chemistry based evaporation rates to generate synthetic data for the purpose of validating the parameter estimation.

First, we sought to determine the cluster evaporation rates from both steady-state and time-dependent cluster concentration data at one temperature. In this first scenario, we sought to determine the cluster evaporation rates from both steady-state and time-dependent cluster concentration data. Due to the mathematical stiffness of the ordinary differential equations describing the time evolution of the cluster concentrations, we were only able to identify a subset of the free parameters (evaporation rates) from the available data. This stiffness originates from the vastly different timescales of some of the key evaporation rates.

In the second scenario, we used only steady-state concentration data but for two different temperatures. We introduced a reparametrization expressing the evaporation rates in terms of cluster formation enthalpies and entropies, and temperature. This reduced the number of parameters we sought to identify. It also lessened the stiffness of the system, as the cluster formation enthalpies and entropies for our system have comparable orders of magnitude. We demonstrated that steady-state concentration data at two different temperatures could be used to determine all the unknown formation enthalpies and entropies, and thus the evaporation rates, to within acceptable accuracy.

The approach presented here can also be applied to infer evaporation rates from mass spectrometric measurements of molecular cluster concentrations. This naturally requires accounting for the process of charging neutral clusters, with its associated uncertainties. A clear conclusion of our proof-of-concept study is that steady-state data at different temperatures is more useful for determining evaporation rates than time-dependent data at a single temperature. Determining very low (below $10^{-5}$ s$^{-1}$) evaporation rates may also require additional measurements at low vapor concentrations, which naturally require longer timescales to reach a steady state.

*Code availability.*   The code is available via GitHub repository: http://doi.org/10.5281/zenodo.3766925

## Appendix A: Supplementary mathematical material

### A1   Cluster kinematics

The kinetics of cluster formation is described by Becker-Döring equations (Ball et al., 1986; Hingant and Yvinec, 2017), which model cluster birth and death which arises from collisions of the smaller clusters into larger ones and evaporations from the bigger clusters into smaller ones. Precisely, labelling the clusters by $i \in \{1, 2, \ldots, N\}$, the time derivative of the $i$th cluster concentration $Y_i$ is governed by

$$\frac{dY_i}{dt} = \frac{1}{2} \sum_{j<i} \beta_{i,(i-j)} Y_i Y_{i-j} + \sum_j \gamma_{i+j \to i,j} Y_{i+j} - \sum_j \beta_{i,j} Y_i Y_j - \frac{1}{2} \sum_{j<i} \gamma_{i \to j, i-j} Y_i + Q_i - S_i, \tag{A1}$$

where $\beta_{i,j}$ is the collision coefficient of clusters $i$ with $j$, and $\gamma_{i+j \to i,j}$ is the evaporation coefficient of cluster $i+j$ into clusters $i$ and $j$, $Q_i$ is an external source term of $i$, and $S_i$ represents the total possible types of losses for the cluster of type $i$. These last two terms, which stand for external supply and destruction mechanisms, depend on the system under consideration.

We now specify the quantity and type of sinks and sources included in our studies. We assume that the concentration of ammonia monomers is constant, while sulphuric acid monomers are supplied to the system at a constant rate comprising $Q = 6.3 \times 10^4$ cm$^{-3}$s$^{-1}$. This settings are selected to imitate the conditions inside of the CLOUD chamber, (Kirkby et al., 2011; Kürten et al., 2015). Further, we include wall losses arising from clusters sticking on the walls of the experimental chamber, (Kürten et al., 2015). These wall losses are parametrized by the size of the cluster

$$S_{\mathrm{wall},i} = 10^{-12}/(2r_i + 0.3 \times 10^{-9}) \quad \mathrm{s}^{-1}, \tag{A2}$$

where $r_i$ is the mass radius of the cluster (in cm). From Eq. A2, wall loss rates decrease with cluster size; in practise it also varies with respect to cluster position in the chamber and time. We neglect any uncertainties attributed to the wall losses. However, we do account for dilution losses, with size-independent value comprising $S_{\mathrm{dil},i} = 9.6 \times 10^{-5}$s$^{-1}$, which had previously been determined in the CLOUD chamber, (Kirkby et al., 2011; Kürten et al., 2015).

Let T denote the temperature of the system of molecular clusters. Using classical kinetic gas theory, the collision rates $\beta_{i,j}$ in Eq. A1 obey

$$\beta_{i,j} = \sqrt{T} \left( \frac{3}{4\pi} \right)^{1/6} \left[ 6k_B \left( \frac{1}{m_i} + \frac{1}{m_j} \right) \right]^{1/2} \left( V_i^{1/3} + V_j^{1/3} \right)^2, \tag{A3}$$

where $m_i$ and $V_i$ are respectively the mass and volume of cluster $i$, and $k_B$ is Boltzmann's constant. In this paper, we assume that the masses and volumes are temperature-independent.

The cluster evaporation rates $\gamma_{i+j \to i,j}$ in Eq. A1 are given by the expression

$$\gamma_{i+j \to i,j} = \beta_{i,j} \frac{P_{\mathrm{ref}}}{k_B T} \exp \left( \frac{\Delta G_{i+j} - \Delta G_i - \Delta G_j}{k_B T} \right), \tag{A4}$$

where $P_{\mathrm{ref}}$ is the reference pressure and $\Delta G_i$ is the Gibbs free energy of formation for cluster $i$. We may further describe the $i$th Gibbs free energy in terms of the cluster formation enthalpy $\Delta H_i$ and entropy $\Delta S_i$:

$$\Delta G_i = \Delta H_i - T \Delta S_i. \tag{A5}$$

We neglect here the weak temperature dependence of real cluster formation enthalpies and entropies.

## A2 Likelihood, data and cost function

The likelihood of observing the data $\mathbf{Y}_{exp}$ given the parameter values $\boldsymbol{\theta}$ is

$$p(\mathbf{Y}_{\text{exp}}|\boldsymbol{\theta}) = \frac{1}{(2\pi)^{n_{\text{out}}/2}} \exp(-\frac{1}{2}F(\boldsymbol{\theta})), \tag{A6}$$

where $n_{\text{out}}$ is the number of measurements and $F(\boldsymbol{\theta})$ is the cost function. We elucidate the cost function below. In our first study in which simulations are conducted with time-dependent data, the number of measurements is $n_{\text{out}} = 4*(N_c*N_t+1)$, where $N_c = 16$ is the number of cluster types whose concentrations are measured and $N_t = 41$ is the number of time-step measurements available for each of the cluster types. As explained in Section 2.1, after each VODE integration, a convergence coefficient is computed from the steady-state cluster concentrations to ensure that the system has attained the steady-state.

In our first study, the parameter fit to the data was evaluated by the sum of squared residuals of the model outputs $\mathbf{Y}_{mod}$ and the measurements, $\mathbf{Y}_{exp}$. The *cost function* (sum of squared residuals) measures how far our model outputs are from the "true" experimental outputs. Precisely,

$$F(\boldsymbol{\theta}) = \sum_{i=1}^{N_c} \sum_{j=1}^{N_t} \frac{(Y_{\text{exp},i}(t_j) - Y_{\text{mod},i}(\boldsymbol{\theta}, t_j))^2}{\sigma_{ji}^2}. \tag{A7}$$

Since concentrations of molecular clusters span a large range (from $10^{-5}$ to $10^9$ particles per $\text{cm}^3$), we normalize the residuals by the measurement error variance $\sigma_{ji}^2$. Normalization in this way avoids overfitting to the larger concentration values. Note also that the error variance $\sigma_{ji}^2$ is matched separately for each cluster type and every time instance. We assume that the instrument is capable of detecting all the cluster types represented in the system at arbitrary small levels of concentration. This simplification was considered in order to illustrate the proposed approach.

When parameter estimation is conducted with steady-state cluster concentrations (as is considered in our second study), we use the following cost function:

$$F(\boldsymbol{\theta}) = \sum_{i=1}^{N_c} \sum_{j=1}^{N_T} \frac{(Y_{\text{exp},i}(T_j) - Y_{\text{mod},i}(\boldsymbol{\theta}, T_j))^2}{\sigma_{ji}^2}. \tag{A8}$$

Now $N_T = 2$ denotes the number of steady state configurations at different *temperatures* (not times!) and $T_j$ stands for the measured *temperature*. In this study, the number of measurements for the likelihood given by Eq.. ?? is $n_{\text{out}} = 4*(N_c*N_T+1)$ (again $N_c = 16$ cluster types).

**Appendix B:  Estimation of the evaporation rates from steady-state data**

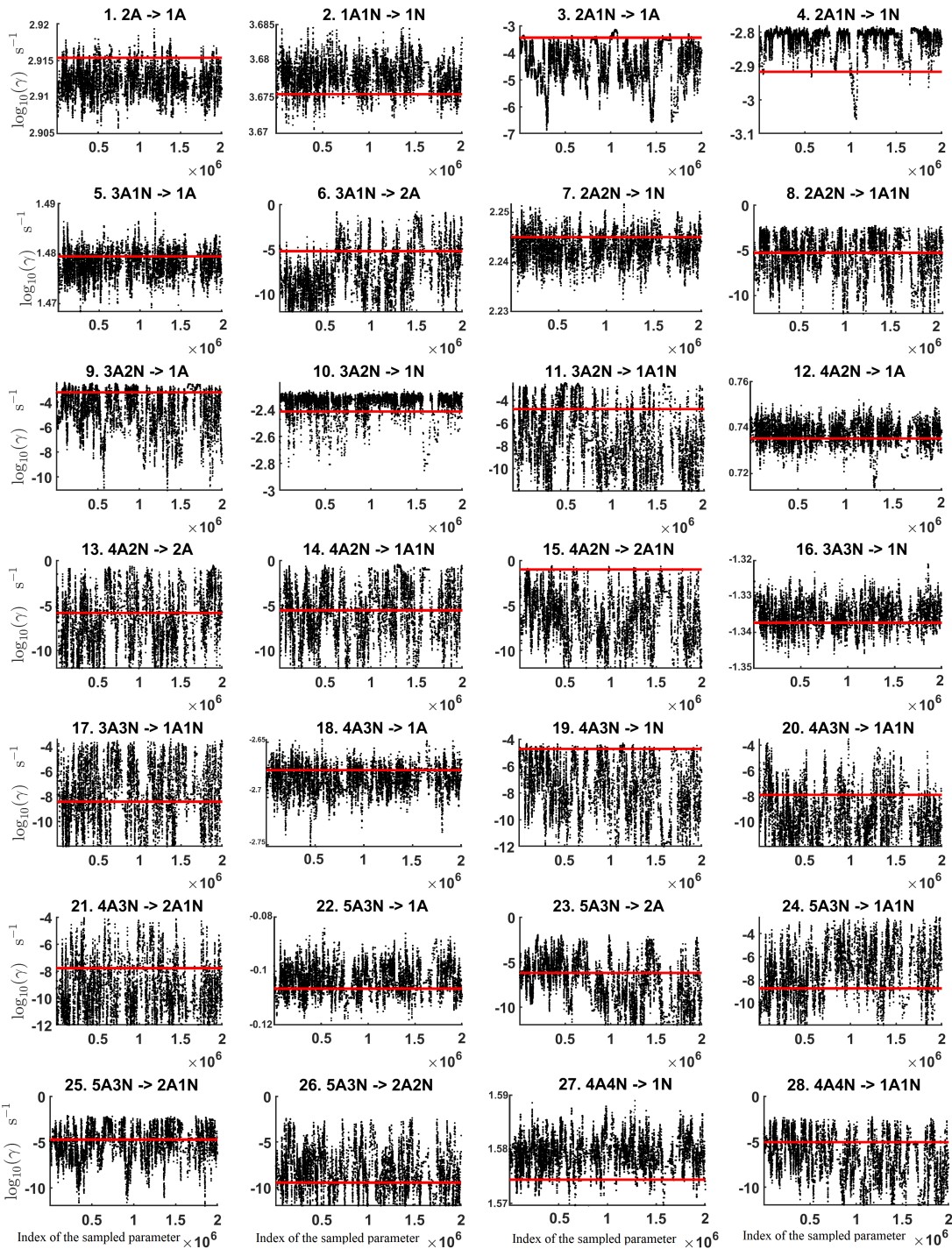

**Figure B1.** Parameter chains (for parameter indexes ranging from 1 to 28) of the base 10 logarithm of the evaporation rates (units given in $s^{-1}$) determined from steady-state cluster concentration measurements at the temperature 278 K. Red lines denote the baseline values from Ortega et al. (2012) used to generate the synthetic data. In reactions "A" stands for $H_2SO_4$ and "N" for $NH_3$.

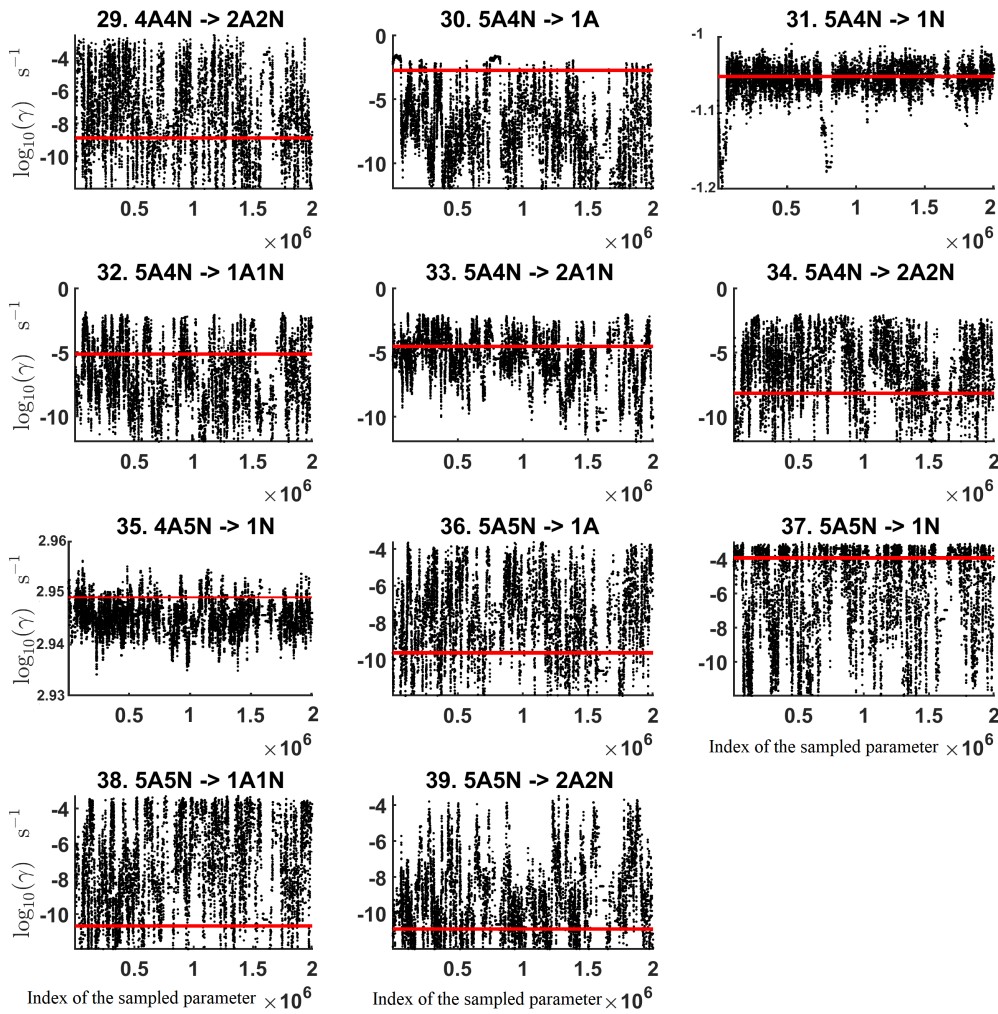

**Figure B2.** Parameter chains (for parameter indexes ranging from 29 to 39) of the base 10 logarithm of the evaporation rates (units given in $s^{-1}$) determined from steady-state cluster concentration measurements at the temperature 278 K. Red lines denote the baseline values from Ortega et al. (2012) used to generate the synthetic data. In reactions "A" stands for $H_2SO_4$ and "N" for $NH_3$.

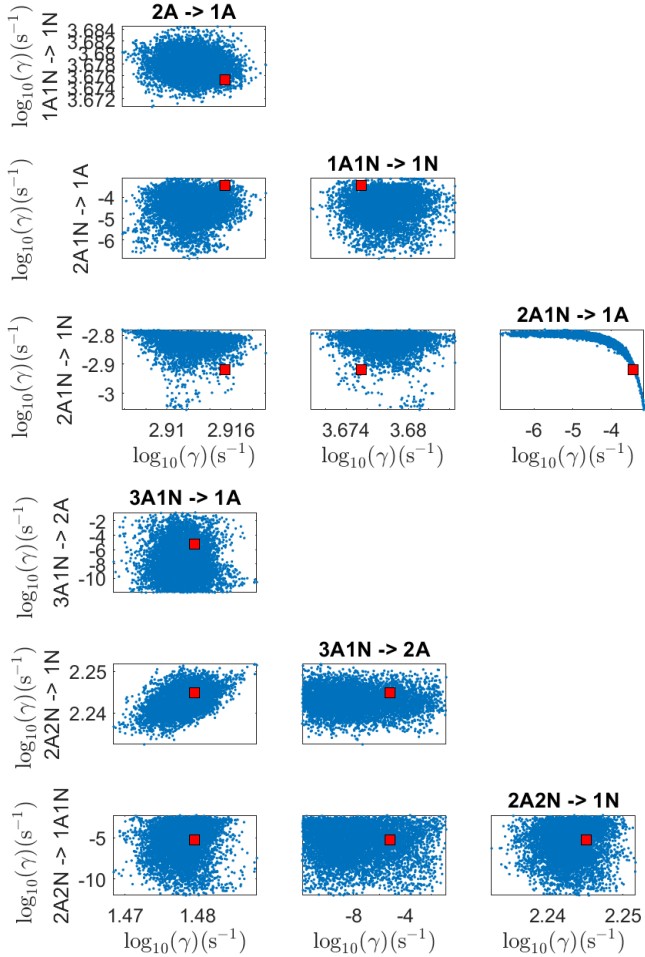

**Figure B3.** Pairwise marginal posterior distributions (for parameter indexes ranging from 1 to 8) of the base 10 logarithm of the evaporation rates (units given in $s^{-1}$) determined from steady-state cluster concentration measurements at the temperature 278 K. Red rectangles denote the baseline values from Ortega et al. (2012) used to generate the synthetic data. In reactions "A" stands for $H_2SO_4$ and "N" for $NH_3$.

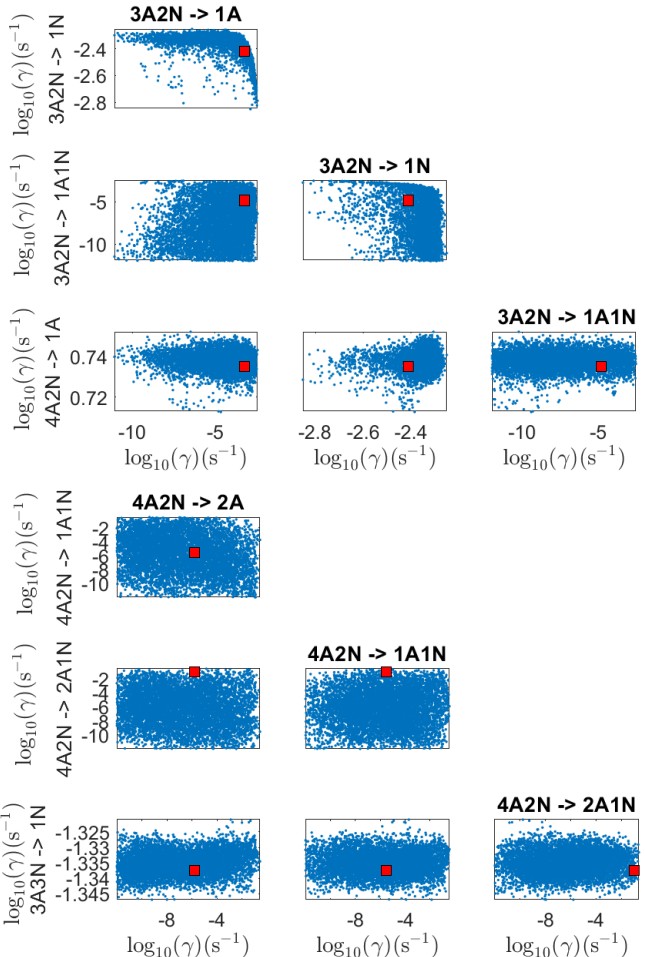

**Figure B4.** Pairwise marginal posterior distributions (for parameter indexes ranging from 9 to 16) of the base 10 logarithm of the evaporation rates (units given in $s^{-1}$) determined from steady-state cluster concentration measurements at the temperature 278 K. Red rectangles denote the baseline values from Ortega et al. (2012) used to generate the synthetic data. In reactions "A" stands for $H_2SO_4$ and "N" for $NH_3$.

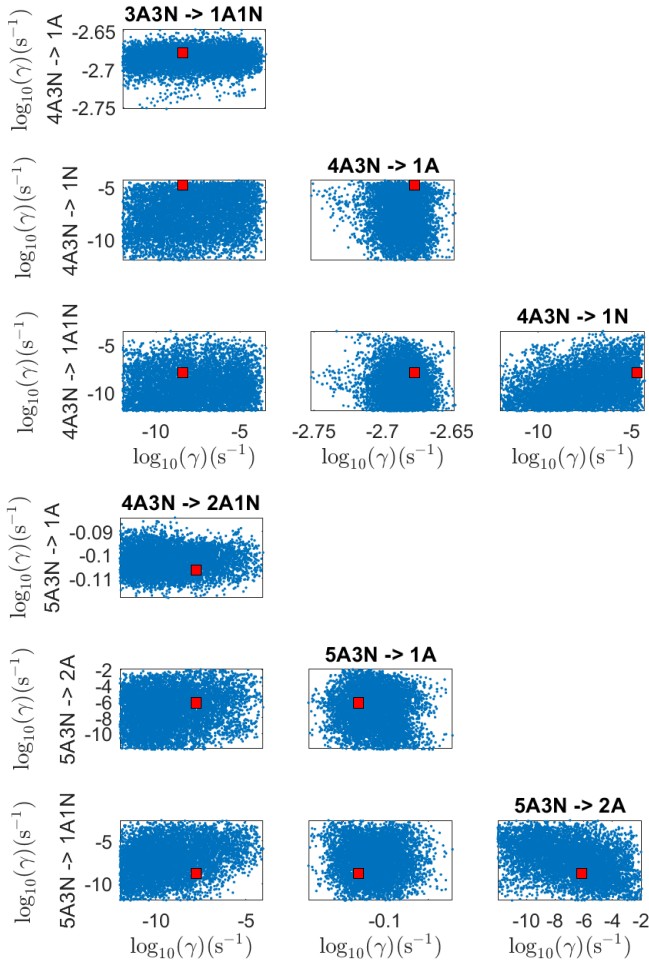

**Figure B5.** Pairwise marginal posterior distributions (for parameter indexes ranging from 17 to 24) of the base 10 logarithm of the evaporation rates (units given in $\mathrm{s}^{-1}$) determined from steady-state cluster concentration measurements at the temperature 278 K. Red rectangles denote the baseline values from Ortega et al. (2012) used to generate the synthetic data. In reactions "A" stands for $H_2SO_4$ and "N" for $NH_3$.

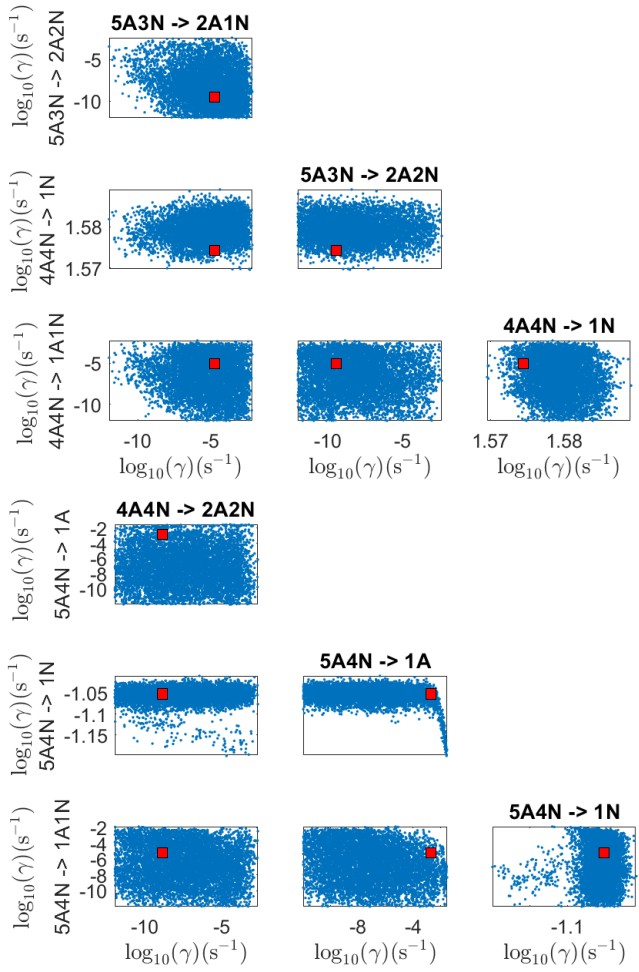

**Figure B6.** Pairwise marginal posterior distributions (for parameter indexes ranging from 25 to 32) of the base 10 logarithm of the evaporation rates (units given in $\mathrm{s}^{-1}$) determined from steady-state cluster concentration measurements at the temperature 278 K. Red rectangles denote the baseline values from Ortega et al. (2012) used to generate the synthetic data. In reactions "A" stands for $H_2SO_4$ and "N" for $NH_3$.

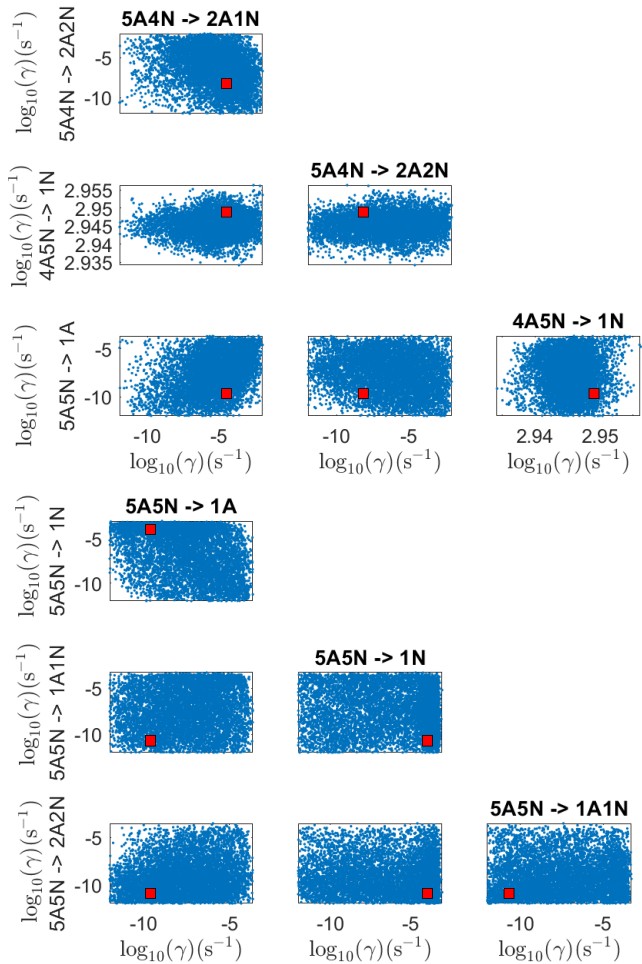

**Figure B7.** Pairwise marginal posterior distributions (for parameter indexes ranging from 33 to 39) of the base 10 logarithm of the evaporation rates (units given in $s^{-1}$) determined from steady-state cluster concentration measurements at the temperature 278 K. Red rectangles denote the baseline values from Ortega et al. (2012) used to generate the synthetic data. In reactions "A" stands for $H_2SO_4$ and "N" for $NH_3$.

**Appendix C:  Estimation of the evaporation rates from transient data**

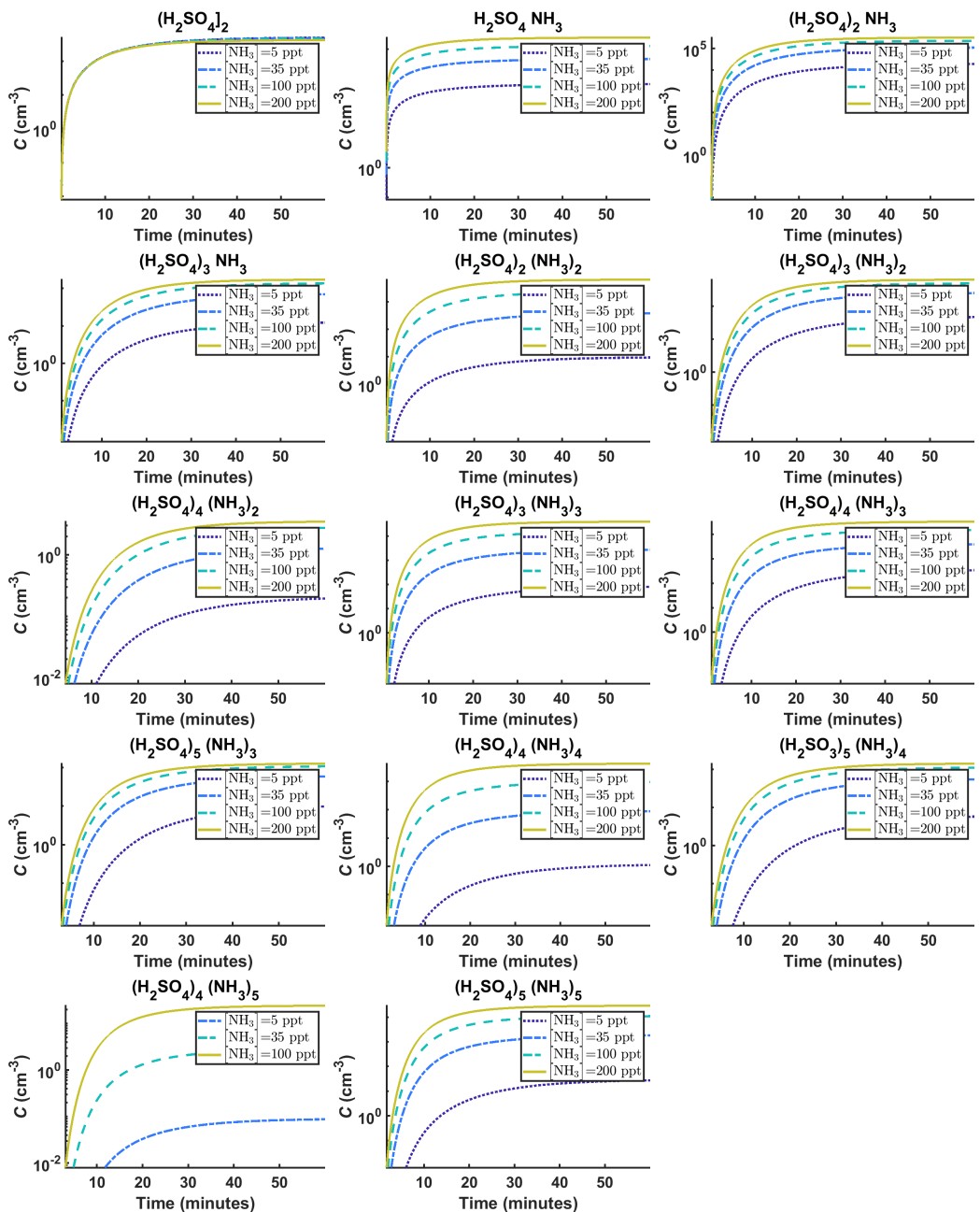

**Figure C1.** Time-dependent cluster concentrations. Simulated time evolution of concentrations for different cluster types at temperature T=278 K for varying $[\mathrm{NH_3}]$ concentration: 5 ppt, 35 ppt, 100 ppt and 200 ppt (see the legend). All the model outputs are amended with multivariate non-correlated Gaussian noise with standard deviation comprising 0.001% of the original cluster concentration. Time resolution comprises 1.5 minutes. The source of sulphuric acid monomer is $[\mathrm{H_2SO_4}] = 6.3 \times 10^4 \ s^{-1}$ in all simulations. In reactions "A" stands for $\mathrm{H_2SO_4}$ and "N" for $\mathrm{NH_3}$.

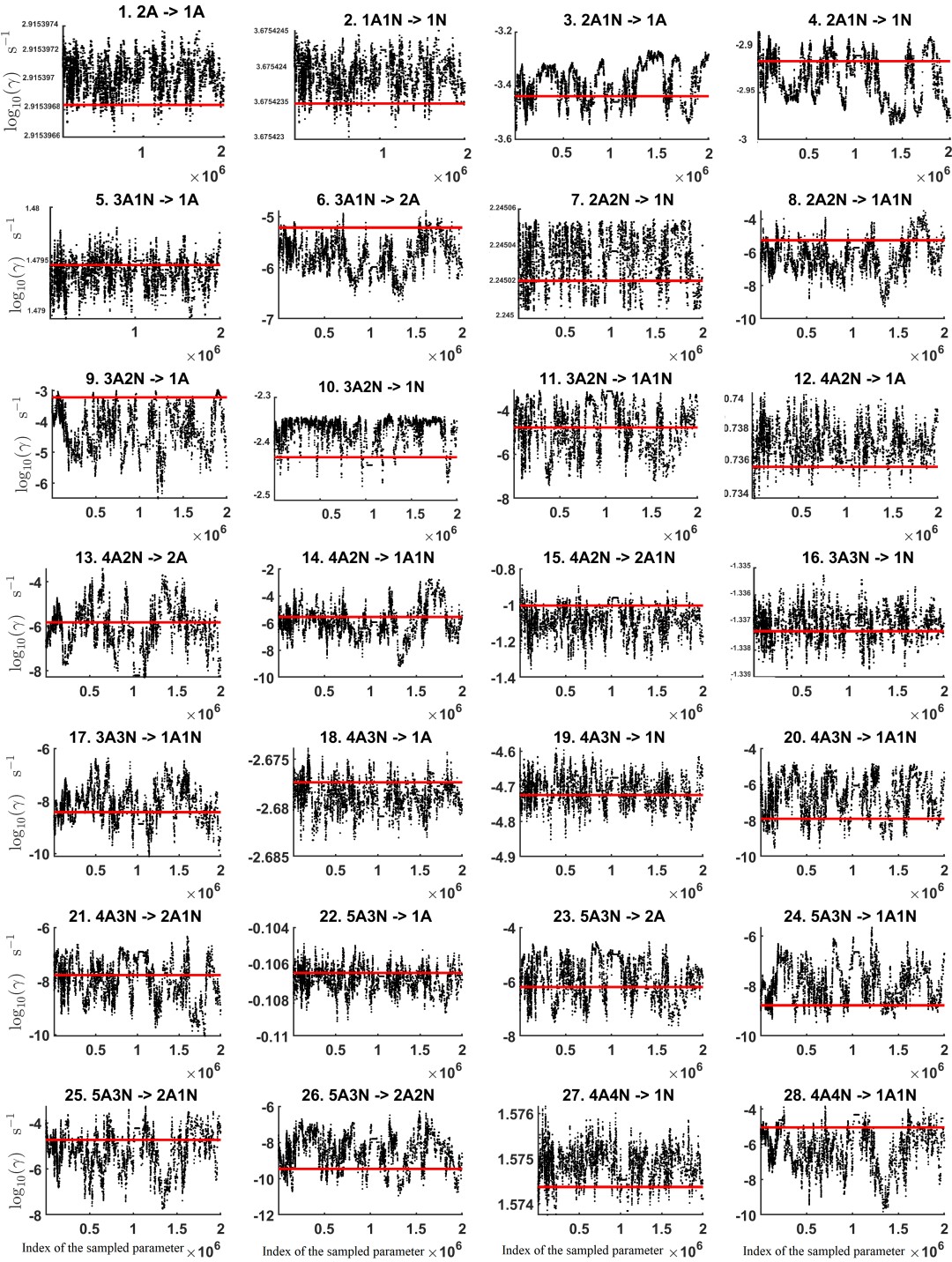

**Figure C2.** Parameter chains (for parameter indexes ranging from 1 to 28) of the base 10 logarithm of the evaporation rates (units given in $s^{-1}$) determined from transient measurements of the cluster concentrations with time resolution comprising 1.5 minutes at the temperature 278 K. Red lines denote the baseline values from Ortega et al. (2012) used to generate the synthetic data.

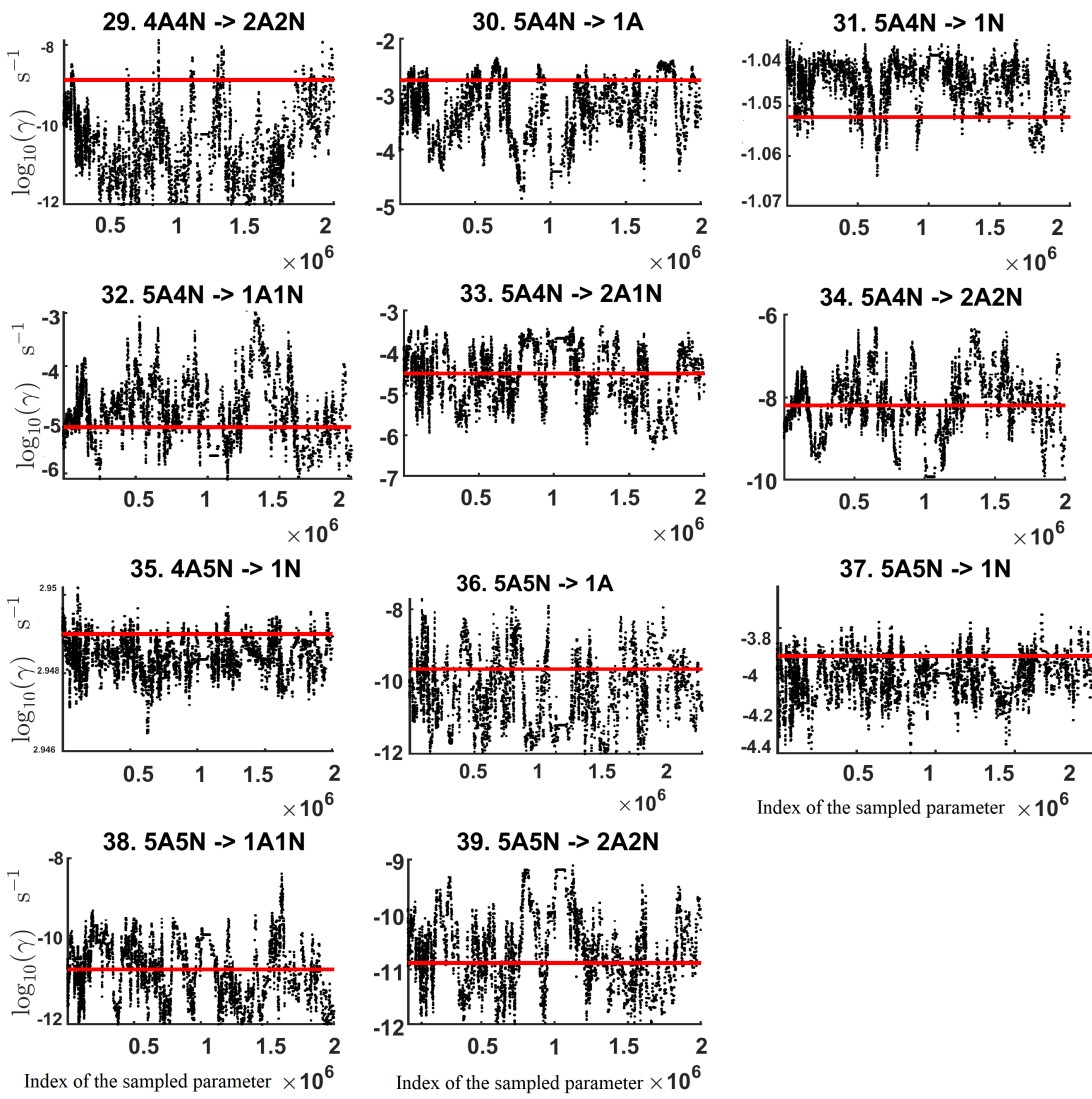

**Figure C3.** Parameter chains (for parameter indexes ranging from 29 to 39) of the base 10 logarithm of the evaporation rates (units given in $s^{-1}$) determined from transient measurements of the cluster concentrations with time resolution comprising 1.5 minutes at the temperature 278 K. Red lines denote the baseline values from Ortega et al. (2012) used to generate the synthetic data. In reactions "A" stands for $H_2SO_4$ and "N" for $NH_3$.

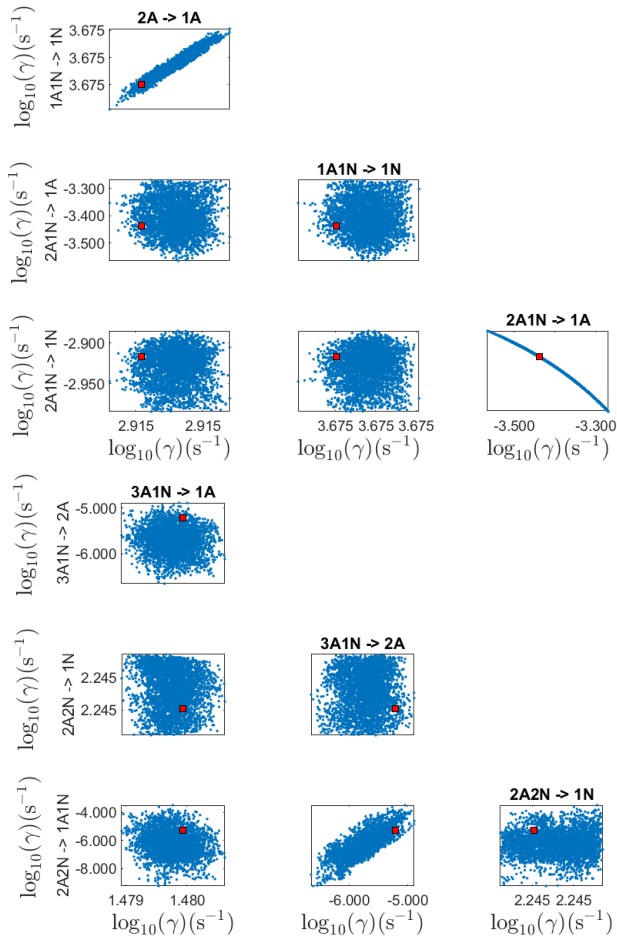

**Figure C4.** Pairwise marginal posterior distributions (for parameter indexes ranging from 1 to 8) of the base 10 logarithm of the evaporation rates (units given in $\mathrm{s}^{-1}$) determined from transient measurements of the cluster concentrations with time resolution comprising 1.5 minutes at the temperature 278 K. Red rectangles denote the baseline values from Ortega et al. (2012) used to generate the synthetic data. In reactions "A" stands for $H_2SO_4$ and "N" for $NH_3$.

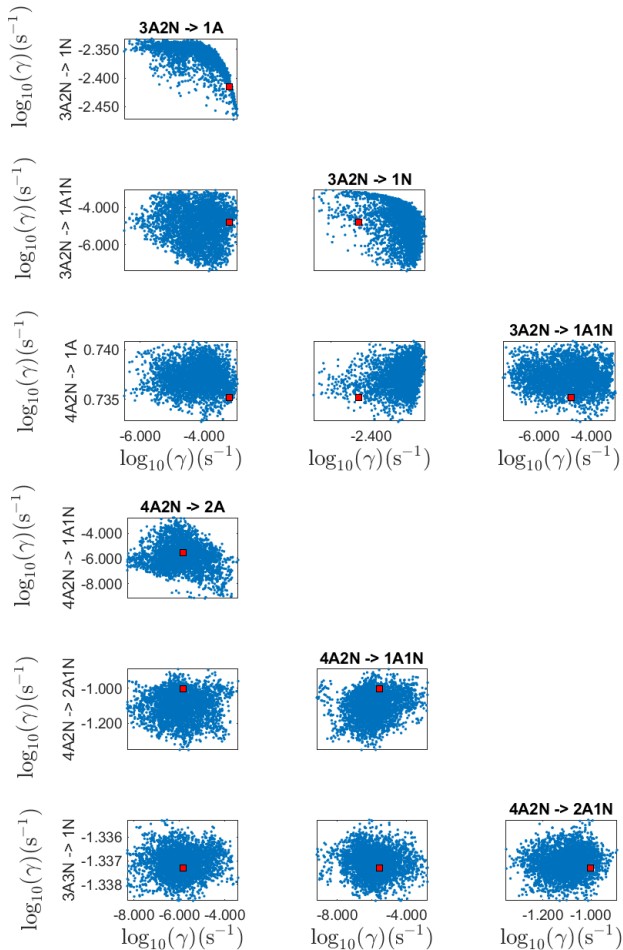

**Figure C5.** Pairwise marginal posterior distributions (for parameter indexes ranging from 9 to 16) of the base 10 logarithm of the evaporation rates (units given in $s^{-1}$) determined from transient measurements of the cluster concentrations with time resolution comprising 1.5 minutes at the temperature 278 K. Red rectangles denote the baseline values from Ortega et al. (2012) used to generate the synthetic data. In reactions "A" stands for $H_2SO_4$ and "N" for $NH_3$.

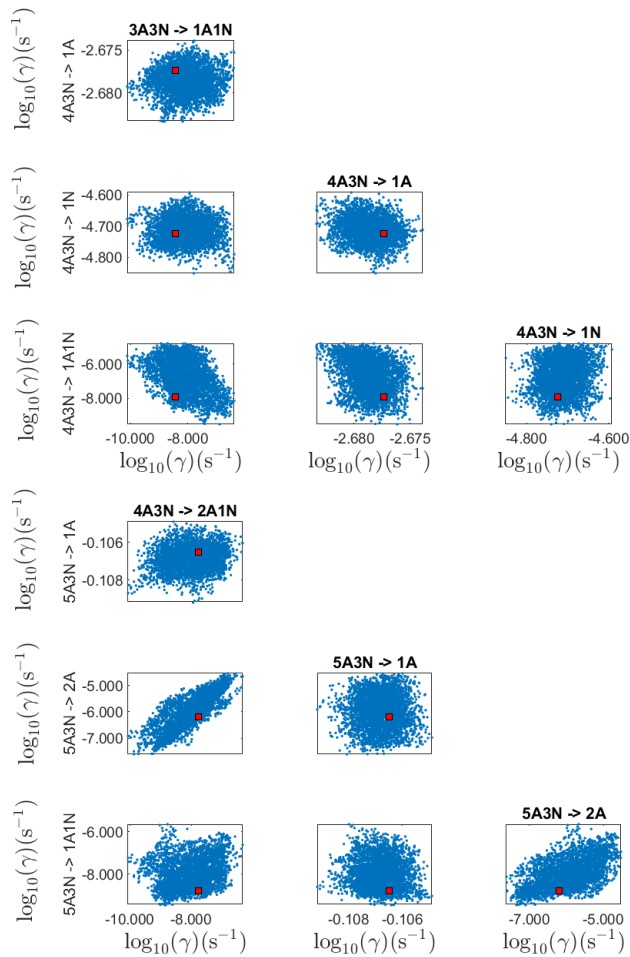

**Figure C6.** Pairwise marginal posterior distributions (for parameter indexes ranging from 17 to 24) of the base 10 logarithm of the evaporation rates (units given in $\text{s}^{-1}$) from transient measurements of the cluster concentrations with time resolution comprising 1.5 minutes at the temperature 278 K. Red rectangles denote the baseline values from Ortega et al. (2012) used to generate the synthetic data. In reactions "A" stands for $H_2SO_4$ and "N" for $NH_3$.

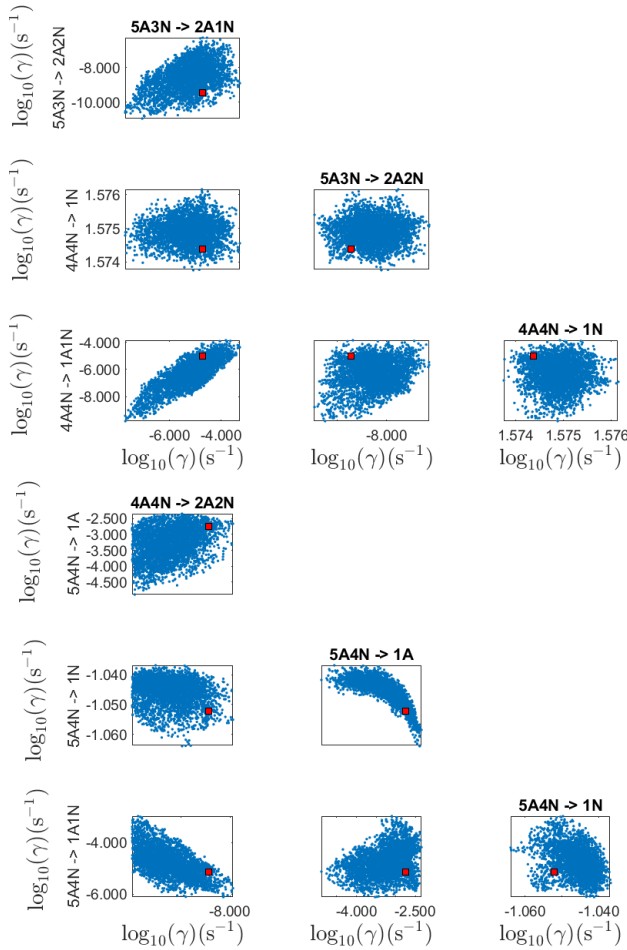

**Figure C7.** Pairwise marginal posterior distributions (for parameter indexes ranging from 25 to 32) of the base 10 logarithm of the evaporation rates (units given in $s^{-1}$) from transient measurements of the cluster concentrations with time resolution comprising 1.5 minutes at the temperature 278 K. Red rectangles denote the baseline values from Ortega et al. (2012) used to generate the synthetic data. In reactions "A" stands for $H_2SO_4$ and "N" for $NH_3$.

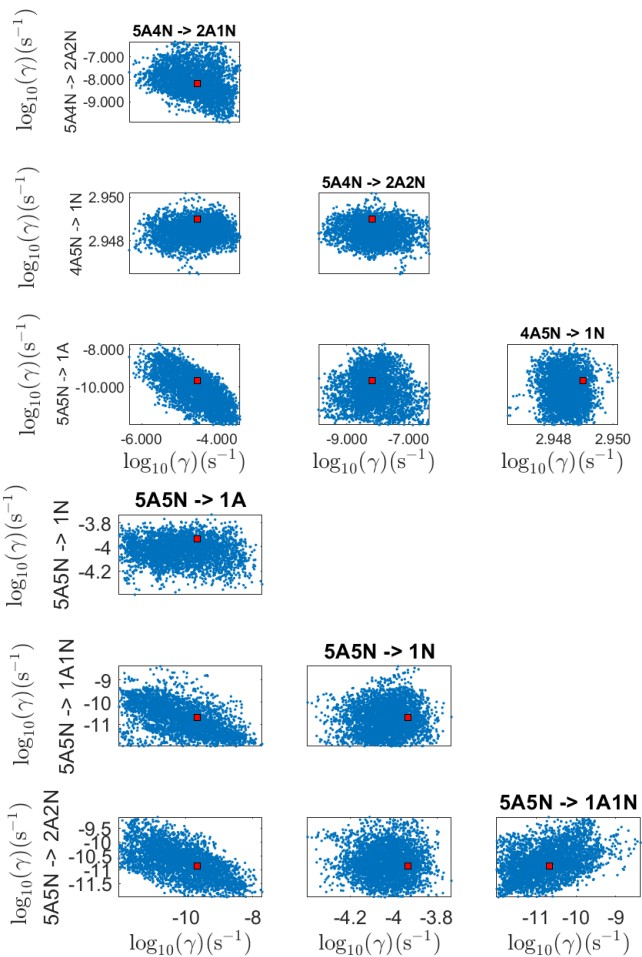

**Figure C8.** Pairwise marginal posterior distributions (for parameter indexes ranging from 33 to 39) of the base 10 logarithm of the evaporation rates (units given in $s^{-1}$) from transient measurements of the cluster concentrations with time resolution comprising 1.5 minutes at the temperature 278 K. Red rectangles denote the baseline values from Ortega et al. (2012) used to generate the synthetic data. In reactions "A" stands for $H_2SO_4$ and "N" for $NH_3$.

| Symbol | Steady-state data ($s^{-1}$) | Transient data ($s^{-1}$) | QC ($s^{-1}$) |
|---|---|---|---|
| 1: $2A \rightarrow 1A$ | $\mathbf{8.16 \times 10^2}$ $(8.05 \times 10^2, 8.31 \times 10^2)$ | $\mathbf{8.23 \times 10^2}$ | $8.23 \times 10^2$ |
| 2: $1A1N \rightarrow 1N$ | $\mathbf{4.75 \times 10^3}$ $(4.69 \times 10^3, 4.87 \times 10^3)$ | $\mathbf{4.74 \times 10^3}$ | $4.74 \times 10^3$ |
| 3: $2A1N \rightarrow 1A$ | $\mathbf{4.22 \times 10^{-4}}$ $(5.92 \times 10^{-11}, 7.27 \times 10^{-4})$ | $\mathbf{3.30 \times 10^{-4}}$ $(1.75 \times 10^{-4}, 5.37 \times 10^{-4})$ | $3.64 \times 10^{-4}$ |
| 4: $2A1N \rightarrow 1N$ | $\mathbf{1.56 \times 10^{-3}}$ $(8.78 \times 10^{-4}, 1.67 \times 10^{-3})$ | $\mathbf{1.33 \times 10^{-3}}$ $(1.04 \times 10^{-3}, 1.4 \times 10^{-3})$ | $1.21 \times 10^{-3}$ |
| 5: $3A1N \rightarrow 1A$ | $\mathbf{2.99 \times 10^1}$ $(2.94 \times 10^1, 3.08 \times 10^1)$ | $\mathbf{3.02 \times 10^1}$ $(3.01 \times 10^1, 3.02 \times 10^1)$ | $3.02 \times 10^1$ |
| 6: $3A1N \rightarrow 2A$ | $-$ $1.50 \times 10^{-1}$ | $\mathbf{2.81 \times 10^{-6}}$ $(2.86 \times 10^{-9}, 2.76 \times 10^{-3})$ | $6.09 \times 10^{-6}$ |
| 7: $2A2N \rightarrow 1N$ | $\mathbf{1.74 \times 10^2}$ $(1.71 \times 10^2, 1.79 \times 10^2)$ | $\mathbf{1.76 \times 10^2}$ | $1.76 \times 10^2$ |
| 8: $2A2N \rightarrow 1A1N$ | $\mathbf{5.52 \times 10^{-4}}$ $< 5.16 \times 10^{-3}$ | $\mathbf{2.11 \times 10^{-6}}$ $(2.95 \times 10^{-10}, 3.59 \times 10^{-4})$ | $5.33 \times 10^{-6}$ |
| 9: $3A2N \rightarrow 1A$ | $\mathbf{3.30 \times 10^{-4}}$ $< 2.91 \times 10^{-3}$ | $\mathbf{7.51 \times 10^{-4}}$ $(3.18 \times 10^{-7}, 1.78 \times 10^{-3})$ | $6.07 \times 10^{-4}$ |
| 10: $3A2N \rightarrow 1N$ | $\mathbf{4.47 \times 10^{-3}}$ $(5.85 \times 10^{-4}, 5.60 \times 10^{-3})$ | $\mathbf{4.16 \times 10^{-3}}$ $(2.86 \times 10^{-3}, 4.66 \times 10^{-3})$ | $3.84 \times 10^{-3}$ |
| 11: $3A2N \rightarrow 1A1N$ | $\mathbf{9.79 \times 10^{-5}}$ $< 3.88 \times 10^{-3}$ | $\mathbf{1.00 \times 10^{-5}}$ $(4.68 \times 10^{-10}, 7.22 \times 10^{-4})$ | $1.64 \times 10^{-5}$ |
| 12: $4A2N \rightarrow 1A$ | $\mathbf{5.50 \times 10^0}$ $(4.50 \times 10^0, 5.72 \times 10^0)$ | $\mathbf{5.46 \times 10^0}$ $(5.39 \times 10^0, 5.51 \times 10^0)$ | $5.43 \times 10^0$ |
| 13: $4A2N \rightarrow 2A$ | $\mathbf{5.24 \times 10^{-7}}$ $< 2.74 \times 10^{-1}$ | $\mathbf{1.03 \times 10^{-6}}$ $(5.66 \times 10^{-11}, 1.88 \times 10^{-2})$ | $1.48 \times 10^{-6}$ |
| 14: $4A2N \rightarrow 1A1N$ | $\mathbf{2.79 \times 10^{-1}}$ $< 6.92 \times 10^{-1}$ | $\mathbf{2.78 \times 10^{-6}}$ $(6.50 \times 10^{-10}, 1.66 \times 10^{-3})$ | $2.80 \times 10^{-6}$ |
| 15: $4A2N \rightarrow 2A1N$ | $\mathbf{6.49 \times 10^{-2}}$ $< 1.02 \times 10^0$ | $\mathbf{9.04 \times 10^{-2}}$ $(3.66 \times 10^{-2}, 1.33 \times 10^{-1})$ | $9.94 \times 10^{-2}$ |
| 16: $3A3N \rightarrow 1N$ | $\mathbf{4.62 \times 10^{-2}}$ $(4.50 \times 10^{-2}, 4.78 \times 10^{-2})$ | $\mathbf{4.61 \times 10^{-2}}$ $(4.58 \times 10^{-2}, 4.62 \times 10^{-2})$ | $4.60 \times 10^{-2}$ |
| 17: $3A3N \rightarrow 1A1N$ | $\mathbf{1.37 \times 10^{-9}}$ $< 3.58 \times 10^{-4}$ | $\mathbf{6.32 \times 10^{-9}}$ $(1.05 \times 10^{-12}, 4.91 \times 10^{-6})$ | $3.74 \times 10^{-9}$ |
| 18: $4A3N \rightarrow 1A$ | $\mathbf{2.08 \times 10^{-3}}$ $(1.79 \times 10^{-3}, 2.27 \times 10^{-3})$ | $\mathbf{2.10 \times 10^{-3}}$ $(2.07 \times 10^{-3}, 2.12 \times 10^{-3})$ | $2.10 \times 10^{-3}$ |
| 19: $4A3N \rightarrow 1N$ | $\mathbf{1.19 \times 10^{-5}}$ $< 7.29 \times 10^{-5}$ | $\mathbf{1.96 \times 10^{-5}}$ $(1.11 \times 10^{-5}, 2.50 \times 10^{-5})$ | $1.88 \times 10^{-5}$ |
| 20: $4A3N \rightarrow 1A1N$ | $\mathbf{9.29 \times 10^{-11}}$ $< 2.65 \times 10^{-4}$ | $-$ $(1.81 \times 10^{-12}, 1.96 \times 10^{-5})$ | $1.23 \times 10^{-8}$ |

**Table C1.** Part 1. Evaporation rates (units given in $s^{-1}$) determined from the steady-state and the transient data presented in Figure 5-6 and Figs. 16-17, respectively. For parameters that have a posterior distribution with the clear peak and practically zero probability density elsewhere, the mode of the distribution (bold face) is given together with the range of possible values in the parenthesis. In some of the cases only the limits can be determined. The last column presents the baseline values from Ortega et al. (2012) used to generate the synthetic data. In reactions "A" stands for $H_2SO_4$ and "N" for $NH_3$.

| Symbol | Steady-state data ($s^{-1}$) | Transient data ($s^{-1}$) | QC ($s^{-1}$) |
|---|---|---|---|
| 21: $4A3N \rightarrow 2A1N$ | — <br> $< 2.14 \times 10^{-4}$ | $\mathbf{4.83 \times 10^{-9}}$ <br> $(3.36 \times 10^{-12}, 6.93 \times 10^{-6})$ | $1.66 \times 10^{-8}$ |
| 22: $5A3N \rightarrow 1A$ | $\mathbf{7.88 \times 10^{-1}}$ <br> $(7.56 \times 10^{-1}, 8.20 \times 10^{-1})$ | $\mathbf{7.81 \times 10^{-1}}$ <br> $(7.77 \times 10^{-1}, 7.86 \times 10^{-1})$ | $7.83 \times 10^{-1}$ |
| 23: $5A3N \rightarrow 2A$ | $\mathbf{2.35 \times 10^{-8}}$ <br> $(\quad < 1.21 \times 10^{-2})$ | $\mathbf{6.34 \times 10^{-7}}$ <br> $(1.26 \times 10^{-11}, 3.35 \times 10^{-4})$ | $6.37 \times 10^{-7}$ |
| 24: $5A3N \rightarrow 1A1N$ | $\mathbf{9.12 \times 10^{-12}}$ <br> $< 3.39 \times 10^{-3}$ | $\mathbf{1.50 \times 10^{-9}}$ <br> $(1.02 \times 10^{-12}, 2.22 \times 10^{-6})$ | $1.70 \times 10^{-9}$ |
| 25: $5A3N \rightarrow 2A1N$ | $\mathbf{7.22 \times 10^{-4}}$ <br> $< 6.95 \times 10^{-3}$ | $\mathbf{1.24 \times 10^{-5}}$ <br> $(1.86 \times 10^{-8}, 5.33 \times 10^{-4})$ | $1.85 \times 10^{-5}$ |
| 26: $5A3N \rightarrow 2A2N$ | $\mathbf{1.52 \times 10^{-8}}$ <br> $< 4.49 \times 10^{-3}$ | — <br> $< 1.25 \times 10^{-4}$ | $3.52 \times 10^{-10}$ |
| 27: $4A4N \rightarrow 1N$ | $\mathbf{3.79 \times 10^{1}}$ <br> $(3.70 \times 10^{1}, 3.88 \times 10^{1})$ | $\mathbf{3.76 \times 10^{1}}$ <br> $(3.75 \times 10^{1}, 3.77 \times 10^{1})$ | $3.75 \times 10^{1}$ |
| 28: $4A4N \rightarrow 1A1N$ | — <br> $< 5.38 \times 10^{-3}$ | $\mathbf{9.05 \times 10^{-6}}$ <br> $(1.52 \times 10^{-10}, 2.57 \times 10^{-4})$ | $9.06 \times 10^{-6}$ |
| 29: $4A4N \rightarrow 2A2N$ | $\mathbf{2.07 \times 10^{-12}}$ <br> $< 2.43 \times 10^{-3}$ | $\mathbf{8.55 \times 10^{-11}}$ <br> $< 1.90 \times 10^{-4}$ | $1.33 \times 10^{-9}$ |
| 30: $5A4N \rightarrow 1A$ | $\mathbf{3.87 \times 10^{-6}}$ <br> $< 2.52 \times 10^{-2}$ | $\mathbf{2.51 \times 10^{-3}}$ <br> $(1.20 \times 10^{-6}, 5.86 \times 10^{-3})$ | $1.77 \times 10^{-3}$ |
| 31: $5A4N \rightarrow 1N$ | $\mathbf{8.92 \times 10^{-2}}$ <br> $(6.68 \times 10^{-2}, 9.74 \times 10^{-2})$ | $\mathbf{9.03 \times 10^{-2}}$ <br> $(8.52 \times 10^{-2}, 9.19 \times 10^{-2})$ | $8.87 \times 10^{-2}$ |
| 32: $5A4N \rightarrow 1A1N$ | — <br> $< 1.55 \times 10^{-2}$ | $\mathbf{3.60 \times 10^{-6}}$ <br> $(6.48 \times 10^{-12}, 1.04 \times 10^{-3})$ | $7.33 \times 10^{-6}$ |
| 33: $5A4N \rightarrow 2A1N$ | $\mathbf{2.28 \times 10^{-4}}$ <br> $< 1.06 \times 10^{-2}$ | $\mathbf{1.32 \times 10^{-4}}$ <br> $(6.46 \times 10^{-10}, 1.53 \times 10^{-3})$ | $2.97 \times 10^{-5}$ |
| 34: $5A4N \rightarrow 2A2N$ | — <br> $< 1.08 \times 10^{-2}$ | $\mathbf{7.30 \times 10^{-9}}$ <br> $(1.51 \times 10^{-11}, 3.17 \times 10^{-4})$ | $6.42 \times 10^{-9}$ |
| 35: $4A5N \rightarrow 1N$ | $\mathbf{8.75 \times 10^{2}}$ <br> $(8.59 \times 10^{2}, 9.03 \times 10^{2})$ | $\mathbf{8.88 \times 10^{2}}$ <br> $(8.85 \times 10^{2}, 8.92 \times 10^{2})$ | $8.89 \times 10^{2}$ |
| 36: $5A5N \rightarrow 1A$ | — <br> $< 2.32 \times 10^{-4}$ | — <br> $< 1.14 \times 10^{-6}$ | $2.23 \times 10^{-10}$ |
| 37: $5A5N \rightarrow 1N$ | $\mathbf{4.96 \times 10^{-4}}$ <br> $< 9.89 \times 10^{-4}$ | $\mathbf{1.00 \times 10^{-4}}$ <br> $(3.48 \times 10^{-5}, 1.85 \times 10^{-4})$ | $1.17 \times 10^{-4}$ |
| 38: $5A5N \rightarrow 1A1N$ | $\mathbf{5.93 \times 10^{-9}}$ <br> $< 5.06 \times 10^{-4}$ | $\mathbf{1.48 \times 10^{-11}}$ <br> $< 1.06 \times 10^{-5}$ | $2.11 \times 10^{-11}$ |
| 39: $5A5N \rightarrow 2A2N$ | — <br> $< 3.09 \times 10^{-4}$ | $\mathbf{2.06 \times 10^{-11}}$ <br> $< 4.11 \times 10^{-7}$ | $1.31 \times 10^{-11}$ |

**Table C2.** Part 2. Evaporation rates (units given in $s^{-1}$) determined from the steady-state and the transient data presented in Figure 5-6 and Figs. 16-17, respectively. For parameters that have a posterior distribution with the clear peak and practically zero probability density elsewhere, the mode of the distribution (bold face) is given together with the range of possible values in the parenthesis. In some of the cases only the limits can be determined. The last column presents the baseline values from Ortega et al. (2012) used to generate the synthetic data. In reactions "A" stands for $H_2SO_4$ and "N" for $NH_3$.

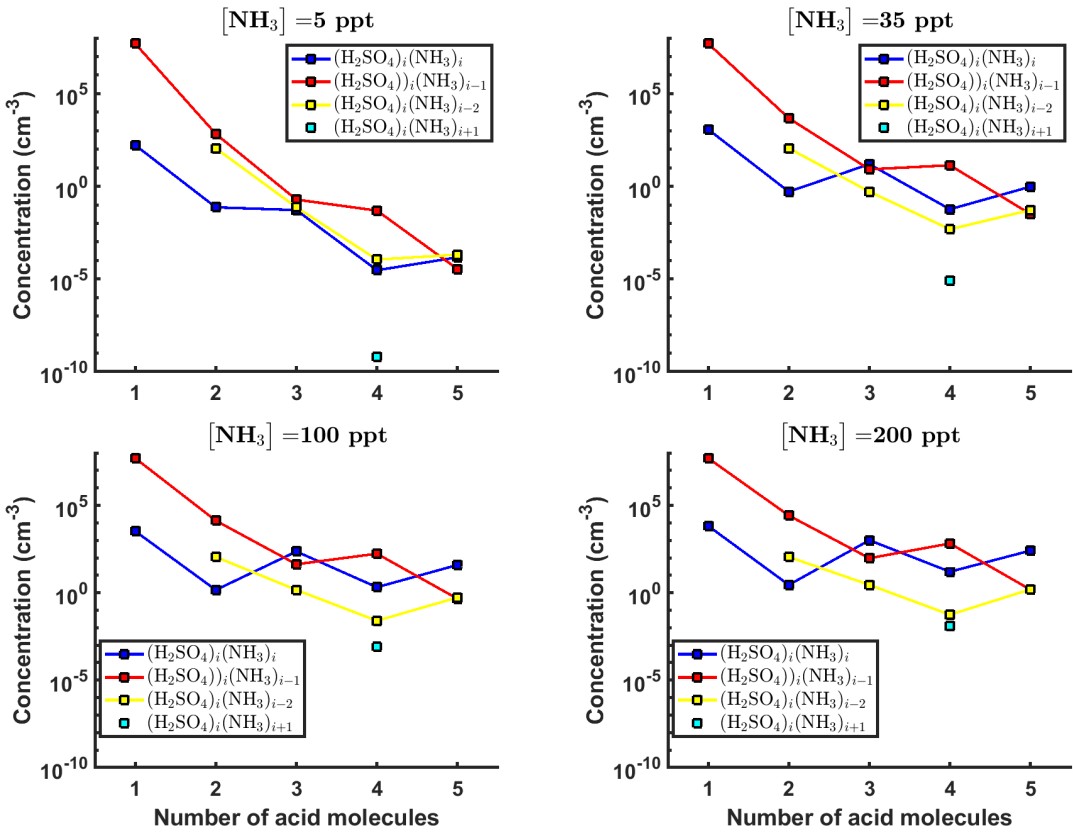

**Figure D1.** Steady-state cluster concentrations for the clusters containing sulphuric acid and a varying number of ammonia molecules as a function of the number of acid molecules for $[NH_3]$ concentrations comprising (a) 5 ppt, (b) 35 ppt, (c) 100 ppt and (d) 200 ppt at temperature T=292 K amended with multivariate non-correlated Gaussian noise with standard deviation comprising 0.001% of the original cluster concentration. The source of sulphuric acid monomer comprises $[H_2SO_4] = 6.3 \times 10^4$ s$^{-1}$in all the simulations. Here the symbols $\Delta H$ and $\Delta S$ stand for cluster formation enthalpies and entropies, respectively. Symbols "A", "N" denote $H_2SO_4$ and "NH$_3$", correspondingly.

**Appendix D: Estimation of the cluster formation enthalpies and entropies from steady-state concentration measurements**

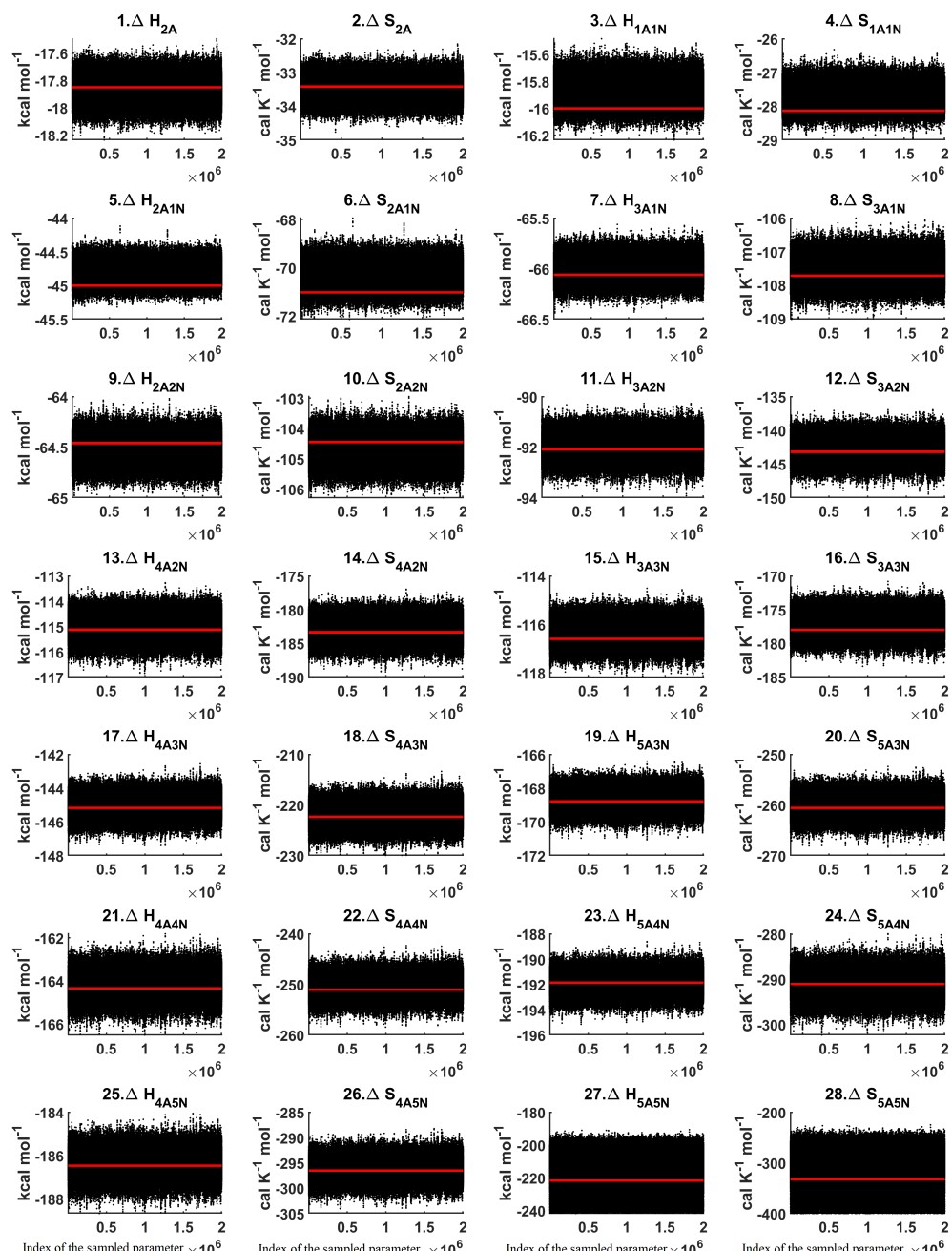

**Figure D2.** Parameter chains of the cluster formation enthalpies (units given in kcal/mol) and entropies (units given in cal $K^{-1}$ $mol^{-1}$) determined from steady-state cluster concentration measurements at two temperatures T=278 K and T = 292 K. Red lines denote the baseline values from Ortega et al. (2012) used to generate the synthetic data. Here the symbols $\Delta H$ and $\Delta S$ stand for cluster formation enthalpies and entropies, respectively. Symbols "A", "N" denote $H_2SO_4$ and "$NH_3$", correspondingly.

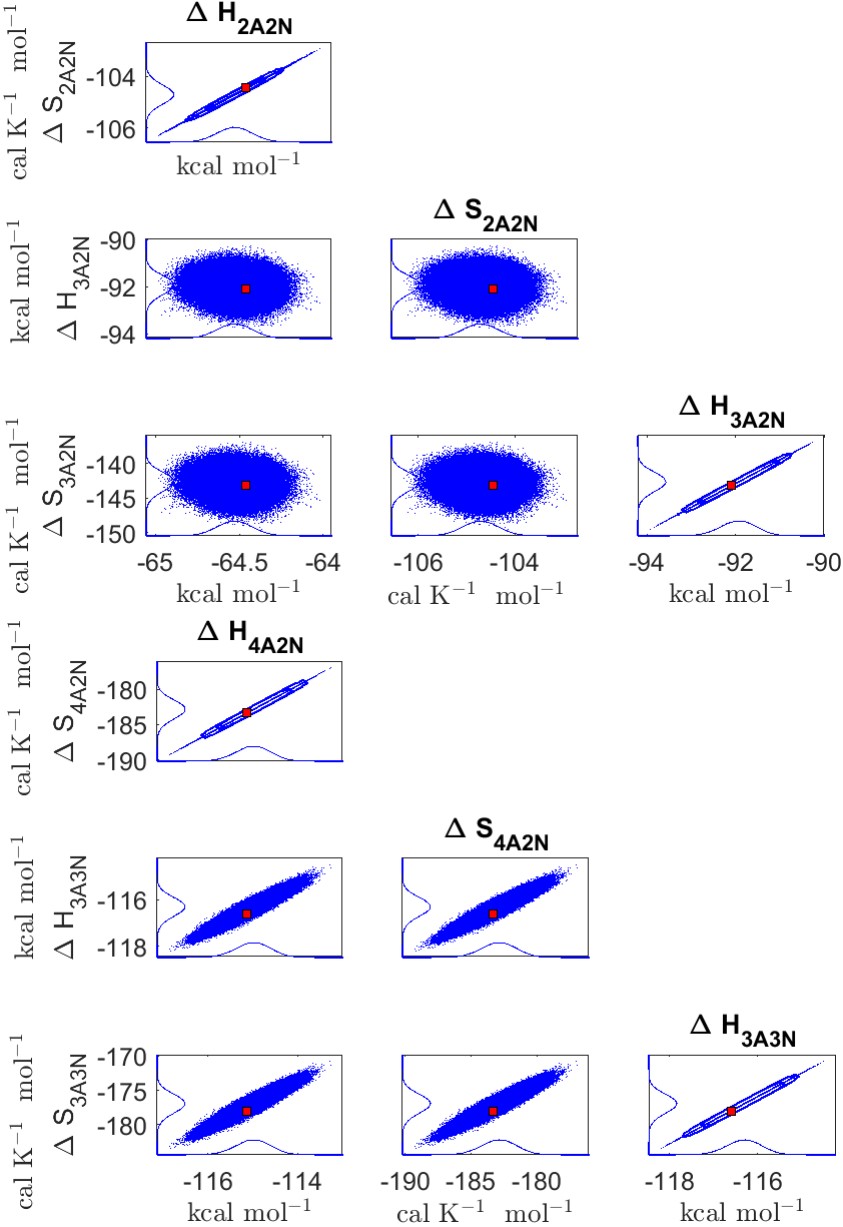

**Figure D3.** Pairwise marginal posterior distributions (for parameter indexes ranging from 9 to 16) of the cluster formation enthalpies and entropies determined from steady-state cluster concentration measurements at two temperatures T=278 K and T = 292 K. Red rectangles denote the baseline values from Ortega et al. (2012) used to generate the synthetic data. Here the symbols $\Delta H$ and $\Delta S$ stand for cluster formation enthalpies and entropies, respectively. Symbols "A", "N" denote $H_2SO_4$ and "$NH_3$", correspondingly.

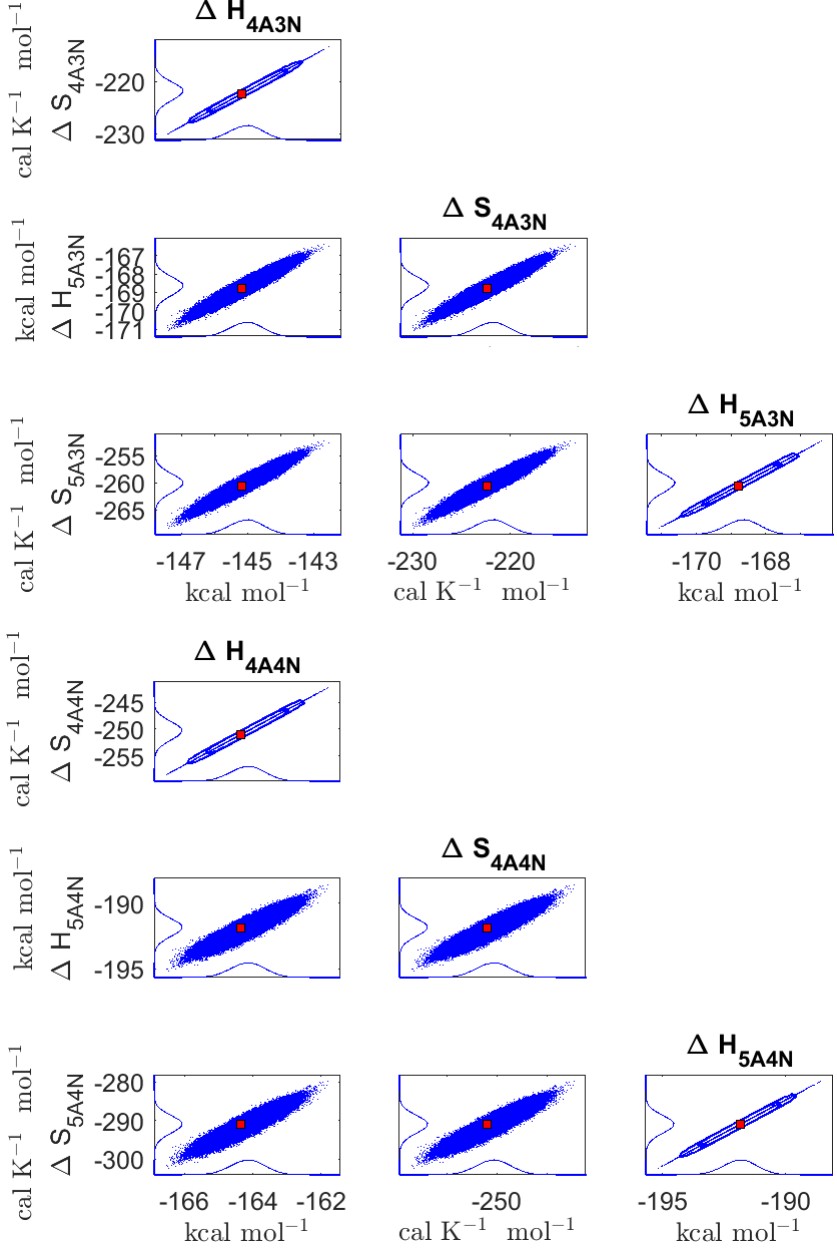

**Figure D4.** Pairwise marginal posterior distributions (for parameter indexes ranging from 17 to 24) of the cluster formation enthalpies and entropies determined from steady-state cluster concentration measurements at two temperatures T=278 K and T = 292 K. Red rectangles denote the baseline values from Ortega et al. (2012) used to generate the synthetic data. Here the symbols $\Delta H$ and $\Delta S$ stand for cluster formation enthalpies and entropies, respectively. Symbols "A", "N" denote $H_2SO_4$ and "$NH_3$", correspondingly.

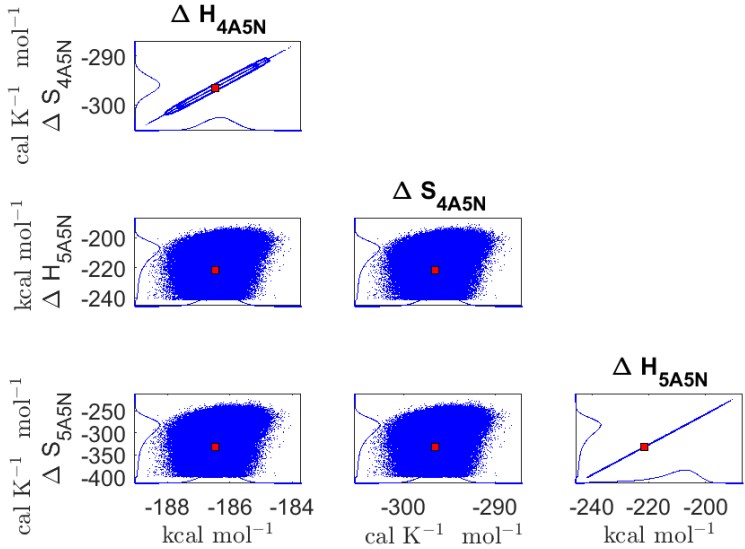

**Figure D5.** Pairwise marginal posterior distributions (for parameter indexes ranging from 25 to 28) of the cluster formation enthalpies and entropies determined from steady-state cluster concentration measurements at two temperatures T=278 K and T = 292 K. Red rectangles denote the baseline values from Ortega et al. (2012) used to generate the synthetic data. Here the symbols $\Delta H$ and $\Delta S$ stand for cluster formation enthalpies and entropies, respectively. Symbols "A", "N" denote $H_2SO_4$ and "$NH_3$", correspondingly.

| Symbol | Mode value | 95% confidence interval | QC | Units |
|---|---|---|---|---|
| 1: $\Delta H_{2A}$ | -17.8891 | (-18.1913,-17.4941) | -17.85 | kcal mol$^{-1}$ |
| 2: $\Delta S_{2A}$ | -33.5475 | (-34.6104,-32.1575) | -33.42 | cal K$^{-1}$ mol$^{-1}$ |
| 3: $\Delta H_{1A1N}$ | -15.8751 | (-16.2344,-15.5158) | -16 | kcal mol$^{-1}$ |
| 4: $\Delta S_{1A1N}$ | -27.6984 | (-28.9594,-26.4374) | -28.14 | cal K$^{-1}$ mol$^{-1}$ |
| 5: $\Delta H_{2A1N}$ | -44.8076 | (-45.2922,-44.174) | -45 | kcal mol$^{-1}$ |
| 6: $\Delta S_{2A1N}$ | -70.3501 | (-72.029,-68.1545) | -71.02 | cal K$^{-1}$ mol$^{-1}$ |
| 7: $\Delta H_{3A1N}$ | -66.0006 | (-66.428,-65.5732) | -66.06 | kcal mol$^{-1}$ |
| 8: $\Delta S_{3A1N}$ | -107.5233 | (-109.0059,-106.0407) | -107.72 | cal K$^{-1}$ mol$^{-1}$ |
| 9: $\Delta H_{2A2N}$ | -64.5005 | (-64.9799,-64.021) | -64.46 | kcal mol$^{-1}$ |
| 10: $\Delta S_{2A2N}$ | -104.6181 | (-106.2857,-102.9505) | -104.45 | cal K$^{-1}$ mol$^{-1}$ |
| 11: $\Delta H_{3A2N}$ | -91.8512 | (-93.9174,-90.2712) | -92.09 | kcal mol$^{-1}$ |
| 12: $\Delta S_{3A2N}$ | -142.3625 | (-149.4438,-136.9474) | -143.18 | cal K$^{-1}$ mol$^{-1}$ |
| 13: $\Delta H_{4A2N}$ | -115.0105 | (-116.7515,-113.2696) | -115.13 | kcal mol$^{-1}$ |
| 14: $\Delta S_{4A2N}$ | -182.938 | (-188.9067,-176.9693) | -183.34 | cal K$^{-1}$ mol$^{-1}$ |
| 15: $\Delta H_{3A3N}$ | -116.3273 | (-118.1437,-114.5108) | -116.6 | kcal mol$^{-1}$ |
| 16: $\Delta S_{3A3N}$ | -177.0462 | (-183.2768,-170.8156) | -177.99 | cal K$^{-1}$ mol$^{-1}$ |
| 17: $\Delta H_{4A3N}$ | -144.9757 | (-147.3975,-142.554) | -145.17 | kcal mol$^{-1}$ |
| 18: $\Delta S_{4A3N}$ | -221.6575 | (-229.9554,-213.3595) | -222.33 | cal K$^{-1}$ mol$^{-1}$ |
| 19: $\Delta H_{5A3N}$ | -168.7305 | (-171.0579,-166.4031) | -168.79 | kcal mol$^{-1}$ |
| 20: $\Delta S_{5A3N}$ | -260.3509 | (-268.3225,-252.3794) | -260.55 | cal K$^{-1}$ mol$^{-1}$ |
| 21: $\Delta H_{4A4N}$ | -164.1272 | (-166.4394,-161.815) | -164.35 | kcal mol$^{-1}$ |
| 22: $\Delta S_{4A4N}$ | -250.2634 | (-258.1819,-242.3449) | -251.03 | cal K$^{-1}$ mol$^{-1}$ |
| 23: $\Delta H_{5A4N}$ | -191.7779 | (-194.9426,-188.6133) | -191.86 | kcal mol$^{-1}$ |
| 24: $\Delta S_{5A4N}$ | -290.7782 | (-301.6196,-279.9369) | -291.05 | cal K$^{-1}$ mol$^{-1}$ |
| 25: $\Delta H_{4A5N}$ | -186.3473 | (-188.639,-184.0557) | -186.47 | kcal mol$^{-1}$ |
| 26: $\Delta S_{4A5N}$ | -296.0839 | (-303.9359,-288.2319) | -296.51 | cal K$^{-1}$ mol$^{-1}$ |
| 27: $\Delta H_{5A5N}$ | -205.943 | (-241.6193,-190.6532) | -221.65 | kcal mol$^{-1}$ |
| 28: $\Delta S_{5A5N}$ | -277.4 | (-,-224.8575) | -332.49 | cal K$^{-1}$ mol$^{-1}$ |

**Table D1.** Thermodynamic parameters identified from steady-state data measured at two temperatures (278 and 292 K). The last column presents the quantum-chemistry based values from (Ortega et al., 2012) used to generate the synthetic data. Here the symbols $\Delta H$ and $\Delta S$ stand for cluster formation enthalpies and entropies, respectively. Symbols "A", "N" denote $H_2SO_4$ and "$NH_3$", correspondingly.

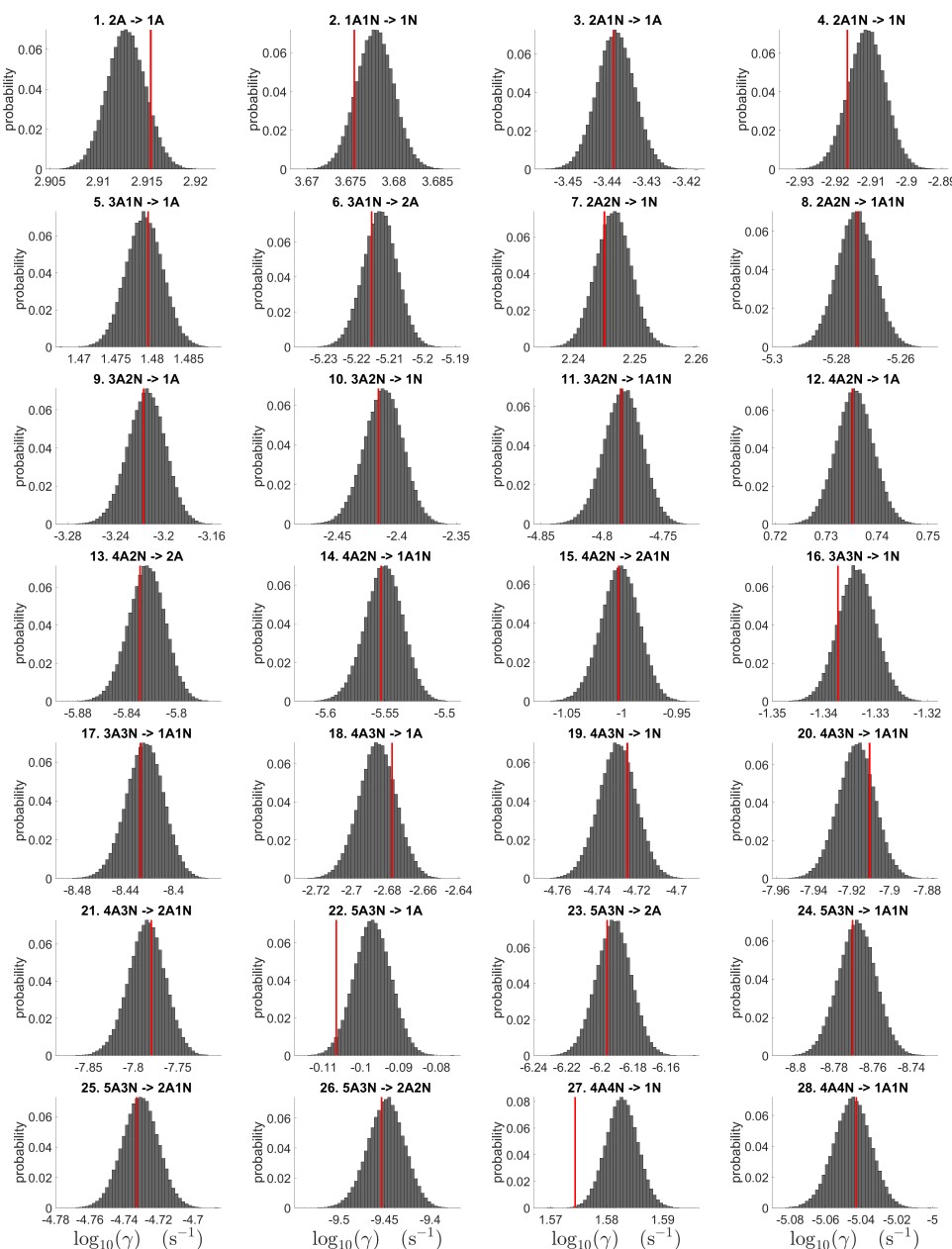

**Figure D6.** One-dimensional marginal distributions (for parameter indexes ranging from 1 to 28) of the base 10 logarithm of the evaporation rates (units given in $\mathrm{s}^{-1}$) at temperature 278 K obtained from a posterior distribution of thermodynamic parameters (cluster formation enthalpies and entropies) determined from steady-state cluster concentration measured at temperatures 278 K and 292 K. Red lines denote the baseline values from Ortega et al. (2012) used to generate the synthetic data. In reactions "A" stands for $\mathrm{H_2SO_4}$ and "N" for $\mathrm{NH_3}$.

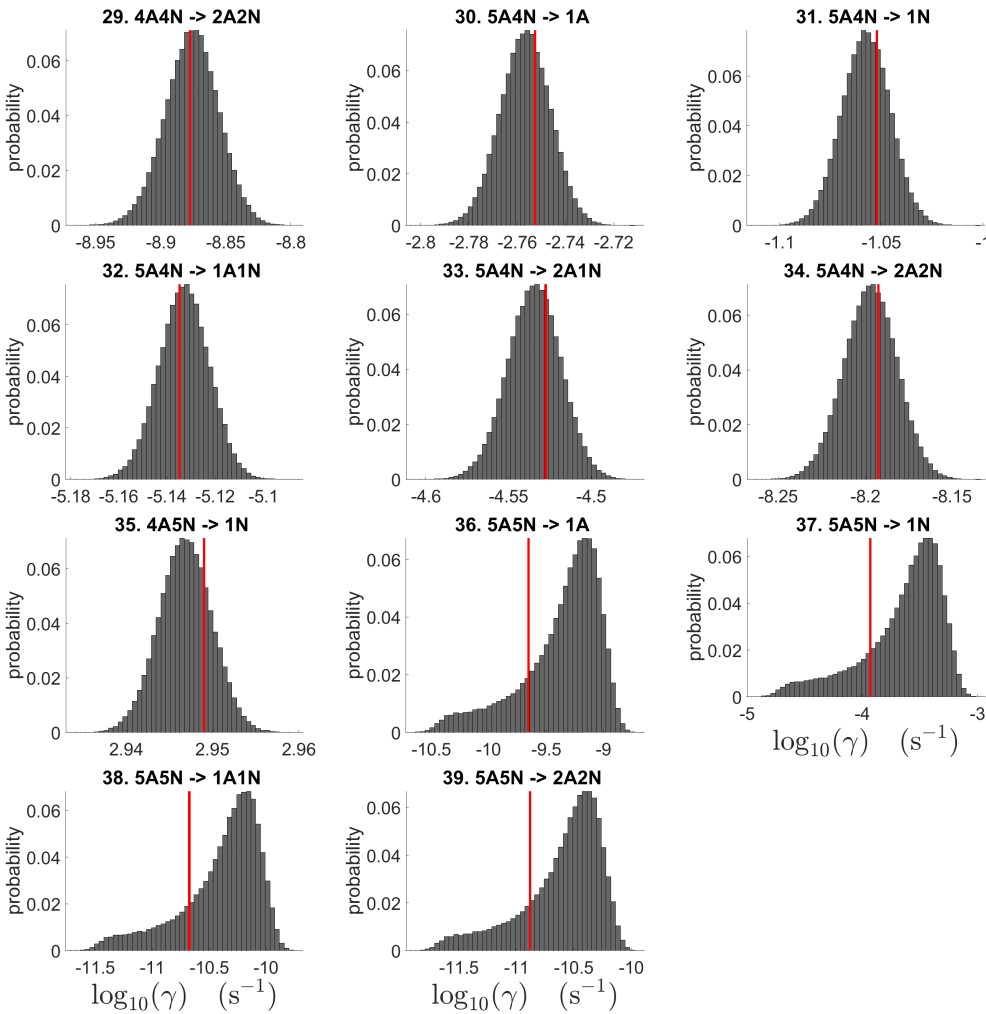

**Figure D7.** One-dimensional marginal distributions (for parameter indexes ranging from 29 to 39) of the base 10 logarithm of the evaporation rates (units given in $s^{-1}$) at temperature 278 K obtained from a posterior distribution of thermodynamic parameters (cluster formation enthalpies and entropies) determined from steady-state cluster concentration measured at temperatures 278 K and 292 K. Red lines denote the baseline values from Ortega et al. (2012) used to generate the synthetic data. In reactions "A" stands for $H_2SO_4$ and "N" for $NH_3$.

| Symbol | Steady-state data for 278 K and 292 K ($s^{-1}$) | QC ($s^{-1}$) |
|---|---|---|
| 1: $2A \rightarrow 1A$ | $\mathbf{8.17 \times 10^2}$ ($8.03 \times 10^2, 8.36 \times 10^2$) | $8.23 \times 10^2$ |
| 2: $1A1N \rightarrow 1N$ | $\mathbf{4.76 \times 10^3}$ ($4.66 \times 10^3, 4.87 \times 10^3$) | $4.74 \times 10^3$ |
| 3: $2A1N \rightarrow 1A$ | $\mathbf{3.64 \times 10^{-4}}$ ($3.48 \times 10^{-4}, 3.84 \times 10^{-4}$) | $3.64 \times 10^{-4}$ |
| 4: $2A1N \rightarrow 1N$ | $\mathbf{1.23 \times 10^{-3}}$ ($1.16 \times 10^{-3}, 1.29 \times 10^{-3}$) | $1.21 \times 10^{-3}$ |
| 5: $3A1N \rightarrow 1A$ | $\mathbf{3.01 \times 10^1}$ ($2.93 \times 10^1, 3.09 \times 10^1$) | $3.02 \times 10^1$ |
| 6: $3A1N \rightarrow 2A$ | $\mathbf{6.12 \times 10^{-6}}$ ($5.77 \times 10^{-6}, 6.47 \times 10^{-6}$) | $6.09 \times 10^{-6}$ |
| 7: $2A2N \rightarrow 1N$ | $\mathbf{1.77 \times 10^2}$ ($1.71 \times 10^2, 1.82 \times 10^2$) | $1.76 \times 10^2$ |
| 8: $2A2N \rightarrow 1A1N$ | $\mathbf{5.33 \times 10^{-6}}$ ($5.02 \times 10^{-6}, 5.64 \times 10^{-6}$) | $5.33 \times 10^{-6}$ |
| 9: $3A2N \rightarrow 1A$ | $\mathbf{6.09 \times 10^{-4}}$ ($5.14 \times 10^{-4}, 7.05 \times 10^{-4}$) | $6.07 \times 10^{-4}$ |
| 10: $3A2N \rightarrow 1N$ | $\mathbf{3.89 \times 10^{-3}}$ ($3.27 \times 10^{-3}, 4.50 \times 10^{-3}$) | $3.84 \times 10^{-3}$ |
| 11: $3A2N \rightarrow 1A1N$ | $\mathbf{1.65 \times 10^{-5}}$ ($1.40 \times 10^{-5}, 1.90 \times 10^{-5}$) | $1.64 \times 10^{-5}$ |
| 12: $4A2N \rightarrow 1A$ | $\mathbf{5.45 \times 10^0}$ ($5.25 \times 10^0, 5.65 \times 10^0$) | $5.43 \times 10^0$ |
| 13: $4A2N \rightarrow 2A$ | $\mathbf{1.49 \times 10^{-6}}$ ($1.27 \times 10^{-6}, 1.72 \times 10^{-6}$) | $1.48 \times 10^{-6}$ |
| 14: $4A2N \rightarrow 1A1N$ | $\mathbf{2.82 \times 10^{-6}}$ ($2.37 \times 10^{-6}, 3.26 \times 10^{-6}$) | $2.80 \times 10^{-6}$ |
| 15: $4A2N \rightarrow 2A1N$ | $\mathbf{1.01 \times 10^{-1}}$ ($8.35 \times 10^{-2}, 1.18 \times 10^{-1}$) | $9.94 \times 10^{-2}$ |
| 16: $3A3N \rightarrow 1N$ | $\mathbf{4.64 \times 10^{-2}}$ ($4.47 \times 10^{-2}, 4.81 \times 10^{-2}$) | $4.60 \times 10^{-2}$ |
| 17: $3A3N \rightarrow 1A1N$ | $\mathbf{3.77 \times 10^{-9}}$ ($3.19 \times 10^{-9}, 4.36 \times 10^{-9}$) | $3.74 \times 10^{-9}$ |
| 18: $4A3N \rightarrow 1A$ | $\mathbf{2.08 \times 10^{-3}}$ ($1.86 \times 10^{-3}, 2.29 \times 10^{-3}$) | $2.10 \times 10^{-3}$ |
| 19: $4A3N \rightarrow 1N$ | $\mathbf{1.87 \times 10^{-5}}$ ($1.69 \times 10^{-5}, 2.05 \times 10^{-5}$) | $1.88 \times 10^{-5}$ |
| 20: $4A3N \rightarrow 1A1N$ | $\mathbf{1.21 \times 10^{-8}}$ ($1.09 \times 10^{-8}, 1.33 \times 10^{-8}$) | $1.23 \times 10^{-8}$ |

**Table D2.** Part 1. Evaporation rates (units given in $s^{-1}$) computed from a posterior distribution of the thermodynamic parameters (cluster formation enthalpies and entropies) which had previously been determined from the steady-state concentration measurements at temperatures 278 and 292 K. Here the mode of distribution (bold face) is given together with the range of possible values in the parenthesis. The last column presents the quantum-chemistry-based evaporation rates used for creating the synthetic data. In reactions "A" stands for $H_2SO_4$ and "N" for $NH_3$.

| Symbol | Steady-state data for 278 K and 292 K ($s^{-1}$) | QC ($s^{-1}$) |
|---|---|---|
| 21: 4A3N → 2A1N | $\mathbf{1.65 \times 10^{-8}}$ ($1.30 \times 10^{-8}$, $1.99 \times 10^{-8}$) | $1.66 \times 10^{-8}$ |
| 22: 5A3N → 1A | $\mathbf{7.98 \times 10^{-1}}$ ($7.63 \times 10^{-1}$, $8.43 \times 10^{-1}$) | $7.83 \times 10^{-1}$ |
| 23: 5A3N → 2A | $\mathbf{6.40 \times 10^{-7}}$ ($5.76 \times 10^{-7}$, $7.24 \times 10^{-7}$) | $6.37 \times 10^{-7}$ |
| 24: 5A3N → 1A1N | $\mathbf{1.71 \times 10^{-9}}$ ($1.54 \times 10^{-9}$, $1.88 \times 10^{-9}$) | $1.70 \times 10^{-9}$ |
| 25: 5A3N → 2A1N | $\mathbf{1.87 \times 10^{-5}}$ ($1.66 \times 10^{-5}$, $2.07 \times 10^{-5}$) | $1.85 \times 10^{-5}$ |
| 26: 5A3N → 2A2N | $\mathbf{3.56 \times 10^{-10}}$ ($2.83 \times 10^{-10}$, $4.30 \times 10^{-10}$) | $3.52 \times 10^{-10}$ |
| 27: 4A4N → 1N | $\mathbf{3.82 \times 10^{1}}$ ($3.69 \times 10^{1}$, $3.95 \times 10^{1}$) | $3.75 \times 10^{1}$ |
| 28: 4A4N → 1A1N | $\mathbf{8.97 \times 10^{-6}}$ ($8.13 \times 10^{-6}$, $1.01 \times 10^{-5}$) | $9.06 \times 10^{-6}$ |
| 29: 4A4N → 2A2N | $\mathbf{1.34 \times 10^{-9}}$ ($1.07 \times 10^{-9}$, $1.62 \times 10^{-9}$) | $1.33 \times 10^{-9}$ |
| 30: 5A4N → 1A | $\mathbf{1.76 \times 10^{-3}}$ ($1.56 \times 10^{-3}$, $1.96 \times 10^{-3}$) | $1.77 \times 10^{-3}$ |
| 31: 5A4N → 1N | $\mathbf{8.70 \times 10^{-2}}$ ($7.68 \times 10^{-2}$, $1.00 \times 10^{-1}$) | $8.87 \times 10^{-2}$ |
| 32: 5A4N → 1A1N | $\mathbf{7.42 \times 10^{-6}}$ ($6.59 \times 10^{-6}$, $8.24 \times 10^{-6}$) | $7.33 \times 10^{-6}$ |
| 33: 5A4N → 2A1N | $\mathbf{2.92 \times 10^{-5}}$ ($2.45 \times 10^{-5}$, $3.40 \times 10^{-5}$) | $2.97 \times 10^{-5}$ |
| 34: 5A4N → 2A2N | $\mathbf{6.40 \times 10^{-9}}$ ($5.40 \times 10^{-9}$, $7.40 \times 10^{-9}$) | $6.42 \times 10^{-9}$ |
| 35: 4A5N → 1N | $\mathbf{8.85 \times 10^{2}}$ ($8.58 \times 10^{2}$, $9.12 \times 10^{2}$) | $8.89 \times 10^{2}$ |
| 36: 5A5N → 1A | $\mathbf{5.38 \times 10^{-10}}$ ($2.01 \times 10^{-11}$, $2.24 \times 10^{-9}$) | $2.23 \times 10^{-10}$ |
| 37: 5A5N → 1N | $\mathbf{2.77 \times 10^{-4}}$ ($1.09 \times 10^{-5}$, $1.15 \times 10^{-3}$) | $1.17 \times 10^{-4}$ |
| 38: 5A5N → 1A1N | $\mathbf{5.05 \times 10^{-11}}$ ($1.87 \times 10^{-12}$, $2.10 \times 10^{-10}$) | $2.11 \times 10^{-11}$ |
| 39: 5A5N → 2A2N | $\mathbf{3.07 \times 10^{-11}}$ ($1.16 \times 10^{-12}$, $1.28 \times 10^{-10}$) | $1.31 \times 10^{-11}$ |

**Table D3.** Part 2. Evaporation rates (units given in $s^{-1}$) computed from a posterior distribution of the thermodynamic parameters (cluster formation enthalpies and entropies) which had previously been determined from the steady-state concentration measurements at temperatures 278 and 292 K. Here the mode of distribution (bold face) is given together with the range of possible values in the parenthesis. The last column presents the quantum-chemistry-based evaporation rates used for creating the synthetic data. In reactions "A" stands for $H_2SO_4$ and "N" for $NH_3$.

*Author contributions.* Author Shcherbacheva A. produced the codes and conducted all the computational experiments for generation of the synthetic data and the MCMC parameter estimation, prepared all the plots presented in the manuscripts. Authors Balehowsky T. and Shcherbacheva A. are responsible for writing the Abstract, Methods and Results sections, and partly the Conclusion section. Author Olenius

T. assisted with generation of the synthetic data, preformed sanity check of the results, gave valuable comments regarding the manuscript. Authors Helin T. and Balehowsky T. actively participated in development of the methodological approach. Author Laine M. provided technical assistance with the 'mcmcstat' toolbox which was used for MCMC simulations. Author Kubečka J. assisted with the code compilation and debug. Author Haario H. assisted with interpretation of the MCMC results and proper usage of the DRAM computational method. Authors Kurtén T. and Vehkamäki H. wrote the Introduction and partly the conclusion, verified the text of the manuscript and helped to interpret the results. The latter two authors verified and edited the manuscript and helped to interpret the outcomes of the study.

*Competing interests.* The authors declare that they have no conflict of interest

*Acknowledgements.* We thank the European Research Council project 692891-DAMOCLES, Academy of Finland (project number 307331), and University of Helsinki: Faculty of Science ATMATH project, for funding, and the CSC-IT Centre for Science in Espoo, Finland, for computational resources. We also thank Olli Pakarinen (Institute for Atmospheric and Earth System Research, University of Helsinki, Helsinki, Finland) for advise in plotting the synthetic data used in the present study.

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
