# Peer review of "Identification of molecular cluster evaporation rates, cluster formation enthalpies and entropies by Monte Carlo method"

_Atmospheric Chemistry and Physics, 2019_

## Referee Comment (RC1) · Anonymous Referee #2 · 26 May 2020

1. The author proposes to use the Markov chain Monte Carlo (MCMC) algorithm to solve the problem of cluster evaporation rate based on cluster distribution, and this is a novel idea for us to evaluate the thermal stability of clusters. But I have a question about the cluster distribution. The author uses ACDC to simulate the cluster distribution (from 1SA.1NH3 to 5SA.5NH3 box) instead of experimental data. Is this simulation result good enough to replace the experimental data? Simulation results are affected by accurate structure, calculation method and basis set. So I suggest that first the author expand the SA.NH3 system to a larger size (1.7 nm). Before using MCMC, simulate the SA.NH3 formation rate and compare it with the experiment data (Nature 502, 359-363, 2013) to illustrate the reliability of the simulation cluster distribution.

2. "time-independent steady-state" in abstract could be revised to be "steady-state";

3. The motivation and test results about the case of single temperature steady-state cluster distributions should be mentioned in the abstract;

4. The best result in this study is the case for steady-state concentration with two temperatures. Is this conclusion general or very specific? How sensitive towards the number of ammonia concentrations and the box size (referring to the cluster types here) is this conclusion?

5. VODE mentioned in L107 may be different from the solver used in McGrath et al. (2012) (ode15s). If so, "A detailed description of this program was published in McGrath et al. (2012)." should be deleted and a simple benchmark should be made to compare different solvers.

6. For table 3, why the minimal values of H and S are set to be -400?

7. L156, "ACDC plus VODE" should be revised to be "ACDC based on VODE";

8. L233, "upper limit" needs to be explained further.

9. L244, "well-defined" need to be defined.

---

## Referee Comment (RC2) · Anonymous Referee #1 · 26 May 2020

This manuscript applies Markov Chain Monte Carlo method to estimate cluster evaporation rates and cluster thermodynamic parameters such as formation enthalpies and entropies while taking collision rates from kinetic gas theory. Cluster evaporation rates were estimated from two data sets: steady-state and transient data. While the transient data can improve the estimates of the evaporation rates compared to the steady state data, neither of them can be satisfied from both magnitude and the marginal posterior distributions of the rates. Cluster formation enthalpies and entropies were then estimated from steady-state cluster concentrations at two temperatures (278 and 292 K) and the cluster evaporation rates were inversed from the cluster Gibbs free energies (determined by enthalpies and entropies). It turns out that the evaporation rates were

greatly improved in terms of variation and the probability distributions except for clusters containing both 5 sulfuric acid and 5 ammonia. Since cluster evaporation rate is an essential parameter that controls cluster growth, this parameter ought to be accurately determined in order to understand atmospheric nucleation. The scientific questions are worthy exploring and are important topics in atmospheric research. However, several major issues need to be fully resolved before the manuscript is considered for publication in this journal.

1. Section 2: the way the authors describe simulation methods is hard to understand. It seems that the authors wrote paragraphs in casual ways, in particular, when describing MCMC simulations, it is very hard to follow the logic. It is suggested that the authors use more plain languages and better logic to rearrange section 2 in order for readers to understand the methods and data sets the authors used or generated.

2. It is quite confused that throughout the paper, the authors use identification of the rates and thermodynamic enthalpies/entropies. Is it better to use for example estimate or similar words?

3. For pairwise marginal posterior distributions, either for evaporation rates or enthalpies/entropies, what criteria the authors used to create these correlations? For example, it seems that evaporation of different monomers from different clusters might be irrelevant.

4. Section 3.4: can the authors present more details of the comparison instead of just some dry descriptions? For example, the authors can add a table to summarize the knowledge up-to-date regarding the evaporation rates from both measurements and modeling so that the readers can be benefit from reading this paper.

5. Can the authors give some plausible explanation why evaporation rates estimated from transient data seem better than those from steady-state data?

6. The authors claimed that the 5A5N has low variance in free energies. However,

an order of magnitude is not small for free energies and it is substantial if this value is applied to the evaporation rates (Line 319 on p18).

7. There are several rather minor comments below:

1) P11, lines 233, do the authors mean that the lower limits of evaporation of a monomer from those clusters are far above the 10ˆ-10 as defined for complete growth?

2) P11, line 240, Figures 3-4 can actually be combined to one figure since they basically represent different parts of the same thing. There are some figures that have similar issues.

3) P15, Figure 5, no label for a, b, c, d.

4) P15, line 284, how the evaporation rates of monomers for clusters 2A display inverse linear correlations in Figures C4-C8?

5) P18, the claim that the estimated formation enthalpies vary at most by 1 kcal mol$-1$, while the variance for the formation entropies is less than 1 cal K-1 mol-1 is not right.

6) P18, line 313 and line 321, Figure 9 should not appear before figure 8.

7) There are lot of typos of molecular sulfuric acid formula throughout the manuscript and a thorough check should be made before submitting the revision. For example, $H_2SO_2$.

8) The references cited in the text are not followed the journal guidelines.

9) Line 34 on p2, subscript; line 37, miss a comma? Line 39, "," is surplus.

10) Line 54 on p3, "-" superscript? line 59, miss a comma between experiment and these? It is apparent an ill-sentence (line 65).

11) Line 104 on p4, into instead of in to?

12) Table 1, it is suggested to add a third column to indicate the number of clusters in each row.

13) Line 123 on p5, kinetic model?

14) Line 369 on p23, what is question mark for?

15) Figure D2, kkal/mol?

---

## Author Comment (AC1) · 27 Jun 2020

[12pt,oneside,a4paper]article

authblk algpseudocode hyperref

amssymb,amsmath,amsthm,mathrsfs

color [normalem]ulem

**Response to discussion-stage referee comments for the paper "Identification of molecular cluster evaporation rates, enthalpies and entropies by Monte Carlo method"**

June 27, 2020

**1   Overview**

In this document we respond to the referee comments for the paper "Identification of molecular cluster evaporation rates, enthalpies and entropies by Monte Carlo method". These comments were provided at the public discussion stage of the review process for publication in Atmospheric Chemistry and Physics.

In Section 2 we list each of Referee's comments. We also include our comment-by-comment responses. Each of the referee's comments are denoted with "**C**" and our responses to the referee's comments are denoted with "**R**".

We thank the referee for his/her time, thoughtfulness, and feedback. All the remarks and suggestions for our paper have been very helpful.

**2  Referee 2 comments and our responses**

**Referee 2's summary:** The author proposes to use the Markov chain Monte Carlo (MCMC) algorithm to solve the problem of cluster evaporation rate based on cluster distribution, and this is a novel idea for us to evaluate the thermal stability of clusters. But I have a question about the cluster distribution. The author uses ACDC to simulate the cluster distribution (from 1SA.1NH3 to 5SA.5NH3 box) instead of experimental data. Is this simulation result good enough to replace the experimental data? Simulation results are affected by accurate structure, calculation method and basis set. So I suggest that first the author expand the SA.NH3 system to a larger size (1.7 nm). Before using MCMC, simulate the SA.NH3 formation rate and compare it with the experiment data (Nature 502, 359-363, 2013) to illustrate the reliability of the simulation cluster distribution.

**R: The answer to reviewer's summary:**

1. "The objective of the present study is to investigate if we can extract evaporation rates from the type of data generated by experiments. Here we search to identify the combination of estimated parameters and experimental data which enables to obtain the estimates for evaporation rates with fair accuracy (i.e., the estimates with the variances comprising less then one order of magnitude).

   In Besel et all, 2020 (J. Phys. Chem. A.) is was shown that the 5x5 simulation box (which is used for generation of the synthetic data is the present study)

produces results in a good agreement with the measurements obtained from the CLOUD chamber experiment. Howevere, the quality of data is not a major issue for our parameter estimation procedure, since the main point is not here to reproduce CLOUD data with the quantum chemical calculations, but to find the settings which will give fair estimates of the evaporation rates in case if the data are available.

The MCMC results are not specific for the simulation box considered in the present study, but rather general. This is supported by the fact that although the size of the system (the number of clusters included into simulations) has impact on the particle formation rates at high temperatures ($>$ 278 K), the particle formation rates and cluster concentrations produced using different simulation boxes are qualitatively similar. Thus the changes of the ACDC outputs due to the difference in the simulation box does not change for MCMC parameter estimation results.

The experimental data can differ from the synthetic data in the sense that they contain noise which originate from measurement instruments and uncertainies associated with experimental conditions (e.g., in CLOUD chamber experiments). Treating the noise inherent for experimental data will be the topic of our future studies. "

2. **C:** "time-independent steady-state" in abstract could be revised to be "steady-state"

   **R:** We have made this change of wording.

3. **C:** The motivation and test results about the case of single temperature steady-state cluster distributions should be mentioned in the abstract;

   **R:** At the end of line 12, we have added:

   "We also estimated the evaporation rates using synthetic steady-state cluster concentration data at one temperature (which has appeared in previous literature) and compared our two study cases to this setting. Both the transient con-
centration data and two-temperature steady-state concentration data estimated
the evaporation rates with less variance than the steady-state one temperature
case. "

4. **C:** The best result in this study is the case for steady-state concentration with two
temperatures. Is this conclusion general or very specific? How sensitive towards
the number of ammonia concentrations and the box size (referring to the cluster
types here) is this conclusion?

**R:** The MCMC results are not specific for the simulation box considered in the
present study, but rather general. This is supported by the fact that although
the size of the system (the number of clusters included into simulations) has im-
pact on the particle formation rates at high temperatures ($> 278$ K), the particle
formation rates and cluster concentrations produced using different simulation
boxes are qualitatively similar. Thus the changes of the ACDC outputs due to
the difference in the simulation box does not change for MCMC parameter esti-
mation results. In Besel et all, 2020 (J. Phys. Chem. A.) is was shown that the
5x5 simulation box (which is used for generation of the synthetic data) produces
reasonable results with a good agreement with the measurements obtained from
the CLOUD chamber experiment. Additionally, the boundary conditions for the
outgrowing clusters (the choice of the clusters that are considered as formed
particles) has only minor influence on the simulation results, given that the simu-
lated system of clusters is defined in a reasonable way (see Besel at al., 2020, J.
Phys. Chem. A).

In general, the accuracy of the MCMC results increases when we include addi-
tional data. In particular, including more concentration data measured at different
ammonia concentrations will yield better estimates for the evaporation rates. The
sensitivity of the estimates to the number of ammonia concentrations will be con-
sidered in the future work. In the present study we rather focus on the question

which combination of estimated parameters and concentration data will produce an accurate estimates for the evaporation rate.

The data of steady-state concentration with two temperatures allowed us to apply two general principles of inverse problems/Bayesian estimation to the problem of estimating evaporation rates. First, the two temperature data set enabled us to reformulate the problem in a numerically effective way (in terms of enthalpy and entropy) that reduced the number of unknown parameters we sought to estimate. Second, the reformulated differential equation describing the time evolution of the concentrations was more numerically stable than the original expression (the stiffness of the equation was reduced in the reformulated form). This made our estimates for the rates less sensitive to small perturbations/errors.

However, the reformulation we used was to parametrize the evaporation rates in terms of enthalpy and entropy. The fact that the entropies and enthalpies were strongly correlated made them an effective parametrization. The strong inverse correlations have a physical explanation. Firstly, both enthalpy and entropy follow from the partition function of the molecular complex, and their functional forms are partly similar. Practically, if a cluster has really strong bonds between the molecules, then that means the formation enthalpy is very negative, and also the intermolecular vibrational frequencies corresponding in a broad sense to vibrations involving those bonds (note that these frequencies dominate the "variable part" of the formation entropy, as the entropy effect from the loss of translational and rotational degrees of freedom is almost a constant factor) are fairly high, meaning that the entropy loss in forming the cluster is large. So if the formation enthalpy is very negative so is also the formation entropy. Conversely, if the cluster is only quite weakly bound, the formation enthalpy is only slightly negative, and the intermolecular frequencies can be very low, leading to a less negative (though still negative of course) formation entropy.

In line 343 we add the Section 3.5."Discussion and future work", where we place

the above-written answer to the reviewer's question.

5. **C:** VODE mentioned in L107 may be different from the solver used in McGrath et al. (2012) (ode15s). If so, "A detailed description of this program was published in McGrath et al. (2012)." should be deleted and a simple benchmark should be made to compare different solvers.

   **R:** We compared the ode15s with those for the vode when creating synthetic data, and they were producing practically identical results.

6. **C:** For table 3, why the minimal values of H and S are set to be -400?

   **R:**

   (a) A narrower range could have been used for the formation enthalpies, since the upper limit correspond to evaporation which in practice almost always happens before growth. The lower limit formally corresponds to zero evaporation. Physically, an upper limit of 0 can be justified by the fact that $> 0$ formation enthalpies would mean no attractive interactions at all, which is obviously physically wrong for polar, H-bonding molecules such as $H_2SO_4$ and $NH_3$. For the lower limit (-400) we mean that on average each $H_2SO_4$ cluster is bound more strongly than in the (extremely strongly bound) $HSO_4^- * H_2SO_4$ cluster, for which the best available computational studies indicate a binding enthalpy roughly around -40 kcal/mol. So it seems unlikely that the average binding per H2SO4 could be tens of kcal/mol stronger than that in the larger clusters where the effect of charge should be much smaller. In any case, a formation enthalpy below -400 kcal/mol means practically zero evaporation so it makes no difference if this is set to a lower value. On the other hand, the largest cluster included into the system has 5 $H_2SO_4$ and 5 $NH_3$, so 10 molecules, and -400 kcal/mol would mean -40 kcal/mol per molecule, which 1) corresponds to the strongest known cluster in the system and 2) means evaporation of practically zero.

(b) For the formation entropies, the 0 cal/Kmol upper limit can be justified as follows: clustering has to have a negative $\Delta H$, as we are reducing the number of gas molecules (and converting translational and rotational degrees of freedom into much more constrained vibrational degrees of freedom). Probably a much lower upper limit could have been used, but certainly the $\Delta S$ values can never be $> 0$. For the lower limit, we state that the typical per-molecule $\Delta S$ for clustering is around -30 cal/Kmol, with a typical variation of up to +-10 cal/mol K, see Kürten, 2019. So for the largest clusters the upper limit corresponds to a per-molecule $\Delta S$ of -40 cal/Kmol. In this case, all the new vibrational degrees of freedom formed in the product clusters are quite rigid, i.e. have very low entropy.

(c) After the line 153 we edit an explanation on the sampling limits selected for the thermodynamic parameters: "Next we justify the limits selected for data setting 2, where we sample thermodynamic parameters. For the formation enthalpies an upper limit of 0 kcal/mol is chosen by the fact that a positive $\Delta H$ would mean an absence of attractive interactions in the molecular cluster, which is physically incorrect for polar, H-bonding molecules such as $H_2SO_4$ and $NH_3$. For the lower limit (-400 kcal/mol) we mean that on average each $H_2SO_4$ is bound substantially stronger than in the $HSO_4^- * H_2SO_4$ cluster, for which the most recent computational studies indicate a binding enthalpy roughly around -40 kcal/mol. Another motivation for the prior distribution selected for the cluster formation enthalpies comes from the fact that the largest cluster included into the system has 5 $H_2SO_4$ and 5 $NH_3$, so 10 molecules, and -400 kcal/mol would give an enthalpy of -40 kcal/mol per molecule, which 1) corresponds to the strongest known cluster in the system and 2) which implies that the evaporation rate is zero for all purposes of measurement.

Next, we set the upper limit for the formation entropies to 0 cal/K/mol, since molecule clustering must have a negative $\Delta H$,as the number of gas

molecules is reduced (and translational and rotational degrees of freedom are converted into much more constrained vibrational degrees of freedom). For the lower limit of -400 cal/K/mol, we state that the typical per-molecule $\Delta S$ for clustering is around -30 cal/K/mol, with a typical variation of up to +-10 cal/mol K, see Kürten, 2019. So for the largest clusters the upper limit corresponds to a per-molecule $\Delta S$ of -40 cal/Kmol. In this situation,all the new vibrational degrees of freedom formed in the product clusters are quite rigid, i.e. have very low entropy."

7. **C:** L156, "ACDC plus VODE" should be revised to be "ACDC based on VODE"

   **R:** We have rewritten this paragraph for clarity, and this emphasis for ACDC has been redirected to Section 2.1. The new paragraph which includes the old line 156 is as follows:

   "We make our initial guess $\boldsymbol{\theta} = \boldsymbol{\theta}_{old}$, where $\boldsymbol{\theta}_{old}$ is the flat distribution which obeys the estimates in Table 4. We also assume that the conditional probability distributions for the parameters given the concentration data are of Gaussian type. Once initialized, the following iterative steps take place. From the likelihood probability distribution for $\boldsymbol{\theta}_{old}$, a new candidate for the unknown parameter values, $\boldsymbol{\theta}_{new}$, is sampled using the proposed Gaussian likelihood distribution. We then use the algorithm in Section 2.1 to obtain concentration outputs from the evaporation rates $\boldsymbol{\theta}_{new}$. In the first stage of DRAM, we chose to accept the new proposed values $\boldsymbol{\theta}_{new}$ with probability ... "

8. **C:** L233, "upper limit" needs to be explained further.

   **R:** We have edited the sentence to read "... all the parameter chains for the evaporation rates have values bounded above by an upper limit which differs for different evaporation rates.'

9. **C:** L244, "well-defined" need to be defined.

**R:** We have rewritten the sentence to state:

"All the evaporation rates larger than $10^{-3}s^{-1}$ are well-identified (see subfigures labelled 1, 2, 4, 5, 7, 10, 12, 16, 18, 22, 27, 31 and 35 in Figures 3- 4), in the sense that the variances for these cluster types are well within our accepted error range of less then one order of magnitude."

**Supplement:**

**Response to discussion-stage referee comments for the paper "Identification of molecular cluster evaporation rates, enthalpies and entropies by Monte Carlo method"**

immediate

June 27, 2020

**1 Overview**

In this document we respond to the referee comments for the paper "Identification of molecular cluster evaporation rates, enthalpies and entropies by Monte Carlo method". These comments were provided at the public discussion stage of the review process for publication in Atmospheric Chemistry and Physics.

In Section 2 we list each of Referee's comments. We also include our commentby-comment responses. Each of the referee's comments are denoted with " $\mathbf{C}$ " and our responses to the referee's comments are denoted with " $\mathbf{R}$ ".

We thank the referee for his/her time, thoughtfulness, and feedback. All the remarks and suggestions for our paper have been very helpful.

**2 Referee 1 comments and our responses**

**Referee 1's summary:** This manuscript applies Markov Chain Monte Carlo method to estimate cluster evaporation rates and cluster thermodynamic parameters such as formation enthalpies and entropies while taking collision rates from kinetic gas theory. Cluster evaporation rates were estimated from two data sets: steady-state and transient data. While the transient data can improve the estimates of the evaporation rates compared to the steady state data, neither of them can be satisfied from both magnitude and the marginal posterior distributions of the rates. Cluster formation enthalpies and entropies were then estimated from steady-state cluster concentrations at two temperatures (278 and 292 K) and the cluster evaporation rates were inversed from the cluster Gibbs free energies (determined by enthalpies and entropies). It turns out that the evaporation rates were greatly improved in terms of variation and the probability distributions except for clusters containing both 5 sulfuric acid and 5 ammonia. Since cluster evaporation rate is an essential parameter that controls cluster growth, this parameter ought to be accurately determined in order to understand atmospheric nucleation. The scientific questions are worthy exploring and are important topics in atmospheric research. However, several major issues need to be fully resolved before the manuscript is considered for publication in this journal.

1. C: Section 2: the way the authors describe simulation methods is hard to understand. It seems that the authors wrote paragraphs in casual ways, in particular, when describing MCMC simulations, it is very hard to follow the logic. It is suggested that the authors use more plain languages and better logic to rearrange section 2 in order for readers to understand the methods and data sets the authors used or generated.

**R**: We have cleaned up the wording in several places in Section 2. Below are the changes we have made.

- In section 2 just before subsection 2.1, we added "In this section we describe the methods used to create synthetic cluster concentration data sets. We also explain the Monte Carlo type algorithms used to estimate the cluster evaporation rates from the data sets."
- In line 93, added "particle" before the word cluster.
- In line 102 we replace "(see the Table 2)" with the sentence "See Table 2 for the summary of ammonia mixing ratio and the source of sulphuric acid monomer used for the ACDC simulations".
- Starting from line 103, rewrote the paragraph to read: "First, we computed the collision rates using the Eq. A3 from kinetic gas theory. Then, we were using these values for the collision rates along with Eq. A4 and the Gibbs free energies computed from Eq. A5 to obtain the evaporation rates. Note that to compute

the Gibbs free energies, we substituted the values for cluster formation enthalpies and entropies given by Olenius et al. (2013b) into Eq. A5. Additionally, we consider the losses on the CLOUD chamber walls which depend on the cluster size computed with Eq A5 (see Kürten (2015)) and a dilution loss of  $S = 9.6 \times 10^{-5}$  $s^{-1}$ . These values for the rates and losses were substituted into the ACDC algorithm (see McGrath et al. (2012)), which simulates the time evolution of molecular cluster concentrations. The ACDC code computes the first-order non-linear, ordinary differential system of cluster concentrations as given by Eq. A1. We then integrate the system produced by ACDC using the Fortran ordinary differential equation solver VODE (N. Brown et al. (1989)). A detailed description of this strategy for solving the forwardproblem of finding the cluster concentration rates from Eq. A1 was published in McGrath et al. (2012). To reproduce the experimental conditions as realistically as possible, each simulation was initialized with non-zero concentration of ammonia monomer and no sulphuric acid. The source of sulphuric acid monomer was supplied at a constant rate.

The above method we used for producing synthetic concentration rates is similar to the one described in Kupiainen-Määttä (2016). We note that unlike Kupiainen-Määttä (2016), in this paper, our particle system is considered at various temperatures."

- In line 110, we changed the first sentence to "Using the above algorithm, model configuration and parameters, we generated two data sets."
- In line 111, we changed the sentence "The maximum time we run is 60 minutes in the above model configurations" to "The maximum time we run is 60 minutes from beginning of the simulation, in the above model configurations"
- In line 112, we reformulated the sentence to clarify how the timedependent synthetic data were generated: "In this case, it is assumed that the concentrations for all the clusters are measured under constant temperature with time resolution comprising 1.5 minutes, which comprises overall 41 time-dependent concentration data for each of the cluster types *i* measured from beginning to the end of each ACDC simulation, before the system has attained a steady state."
- In line 114, we added at the end of the sentence

- In line 127, we added the sentence "Now we describe how we estimate the evaporation rates from the noisy synthetic data sets obtained by the method described in Section 2.1. We first give a general overview of the basic Metropolis algorithm (Metropolis (1953)), then describe a modification of the algorithm we implemented in this study, and finally, in Section 2.2.3 we apply this general framework to each of our study cases."
- We added section 'The Metropolis algorithm' restructured the Section 2.2 into three sub-sections,
- We changed the sentences starting from line 129 to read The objective of MCMC in parameter estimation is to identify all the possible parameter values which yield the best fit with the experimental data. Unlike optimization algorithms that produce one best combination of parameter values, the in the MCMC procedure all the most-probable combinations of parameter values are estimated given the data. To obtain these combinations, the values of parameters are generated and stored into the MCMC "chain". The MCMC chain will converges to the distribution containing all the most-likely combinations of parameter values as a number of sampled parameter sets (i.e., the chain length) increases. The distribution formed from the chain approximates a posterior probability density function which gives the likelihood of observing each of the parameters given the concentration data.
- To make the MCMC workflow more logical, we rearranged the remaining content of Section 2.2 into 3 subsections: "The Metropolis algorithm" (Section 2.2.1), "The DRAM algorithm" (Section 2.2.2) and "The overview of the MCMC runs" (Section 2.2.3). The fist section explains the basic Metropolis algorithm, the second section gives a detailed description of the Delayed Rejection Adaptive Metropolis algorithm used in the present study, the last subsection explains the domain restrictions for sampled parameters and parameter representation of the evaporation rates.
- After the line 132 We added subsection with the caption 'The Metropolis algorithm'.
- Starting with line 133, we wrote the subsection describing the basic Metropolis algorithm in application to our simulation: "First, a prior distribution for the parameter values  $\boldsymbol{\theta}$  (represented in array form) is chosen and set to be the proposed "true" distribution from which possible parameters are sampled. The prior is typically selected based on the previous knowledge for the parameter values. Then an initial guess for parameter values (denoted as  $\theta_0$  or  $\theta_{old}$ ) is selected from the prior distribution.

Starting from the initial guess, the algorithm samples candidate parameter values (denoted as  $\theta_{\text{new}}$ ) from a proposal distribution centred at the previous point (denoted as  $q(\theta_{\text{old}}, \theta_{\text{new}})$ ). The proposal density  $q(\theta_{\text{old}}, \theta_{\text{new}})$  is symmetric, which means that the probability of step taken from the 'old'  $\theta_{\text{old}}$  to the 'new' point  $\theta_{\text{new}}$  is same as the probability of the reverse step ( $q(\theta_{\text{old}}, \theta_{\text{new}}) = q(\theta_{\text{new}}, \theta_{\text{old}})$ ).

Then the candidate point  $\boldsymbol{\theta}_{new}$  is either accepted or rejected, according to the least-squares fit of the output to the data, which measures the difference between the modelled  $\mathbf{Y}_{mod}$  and measured  $\mathbf{Y}_{exp}$  cluster concentrations:

$$F(\boldsymbol{\theta}_{\text{new}}) = \sum_{i=1}^{N} \frac{(Y_{\text{exp},i} - Y_{\text{mod},i}(\boldsymbol{\theta}_{\text{new}}))^2}{\sigma_i^2},$$
 (1)

where N stands for the number of measurements in synthetic data. We consider two sets of synthetic cluster concentrations: timedependent, measured at T = 278 K and steady-state, measured for two temperatures (at T = 278 K and T = 292 K), as explained in Section 2.1. For the time-dependent synthetic data  $N = N_C \times N_t$ , where  $N_C = 16$  stands for the number of cluster types included into simulations, while  $N_t = 41$  stands for the number of time-step measurements available for each of the cluster types. For the second data set,  $N = N_C \times N_T$ , where  $N_T = 2$ denotes the number of experiments conducted at different temperatures. In the formula above we scale the squared residuals by the measurement error variance  $\sigma_i^2$  to avoid overfitting to the larger concentration values. The error variance  $\sigma_i^2$  is matched depending on cluster type, time instance and temperature. See A2 for more details.

At each iteration of the Metropolis algorithm, the value  $F(\boldsymbol{\theta}_{\text{new}})$  is compared to the least-square sum from the previous step  $F(\boldsymbol{\theta}_{\text{old}})$ . If the new value is lower (i.e., the candidate parameters fit the data at least as good as the the old values), then the step is accepted. In the opposite case, when  $F(\boldsymbol{\theta}_{\text{new}}) > F(\boldsymbol{\theta}_{\text{old}})$ , the point will be accepted with the probability

$$\alpha_{\rm acc} = \exp\left[-\frac{1}{2}(F(\boldsymbol{\theta}_{\rm new}) - F(\boldsymbol{\theta}_{\rm old}))\right].$$
 (2)

If the candidate point is accepted, the parameter combination  $\boldsymbol{\theta}_{\text{new}}$  is added to the chain, in the opposite case the old value is replicated in the chain. Finally, the value  $F(\boldsymbol{\theta}_{\text{old}})$  is replaced with  $F(\boldsymbol{\theta}_{\text{new}})$  and saved for the next iteration."

In this paper we employ a variant of the Metropolis algorithm which is more efficient at parameter sampling when the parameter space is large (Haario (2006)). This variant is called the Delayed Rejection Adaptive Metropolis (DRAM), introduced in Haario (2006). We briefly explain our approach below.

- We move the text starting from the line 134 ("We remark that to create a reliable sample from the underlying parameter distribution..") and ending at the end of the paragraph to Section 2.2.3 ("The overview of the MCMC runs").
- We move the lines 142-143 to the end of the Section 2.1.
- In line 142 we insert the Section 2.2.2 "The DRAM algorithm".
- In line 144 we add the sentence to "Similar to the basic Metroplois algorithm, the DRAM is initialized with the prior distribution and the initial guess for parameter values."
- In line 150, we cut the word "predefined".
- We move the Tables 3 and 4 to Section 2.2.3, titled as "The overview of the MCMC runs".
- We move the lines 143-144 to the end of the Section 2.2.2. We insert them after the description of the DRAM algorithm (after the line 188).
- We move the explanations of prior limits used for sampling the evaporation rates and thermodynamic data (lines 147-154) to Section 2.2.3.
- Starting from line 154, we changed the paragraph to "We make our initial guess  $\boldsymbol{\theta} = \boldsymbol{\theta}_{old}$ , where the prior distribution is flat; i.e., all the values within the upper and lower limits that were chosen for the sampled parameters are equally probable. The limits are summarized in Table 4. We also assume that the conditional probability distributions for the parameters given the concentration data are of Gaussian type.

Once initialized, the following iterative steps take place. From the previous point in the MCMC chain  $\theta_{old}$ , a new candidate for the unknown parameter values,  $\theta_{new}$ , is sampled using the Gaussian proposal distribution. We then use the algorithm in Section 2.1

to obtain concentration outputs from the evaporation rates  $\boldsymbol{\theta}_{new}$ . In the first stage of DRAM, we chose to accept the new proposed values  $\boldsymbol{\theta}_{new}$  with probability ... "

R:

- Changed in line 162 "... the concentrations obtained from the ACDC and VODE simulations with parameters  $\theta_{old}$  and  $\theta_{new}$ , respectively."
- After the paragraph 186-189 we insert the Section 2.2.3 with the caption "The overview of the MCMC runs".
- At the beginning of the Section 2.2.3 we insert the paragraph "In our implementation of the DRAM algorithm, we impose upper and lower limits for the parameter values. We add such domain restrictions to exclude unphysical estimates for our parameters. These restrictions are encoded in our prior distribution, which we set to be a combination of so-called "flat priors", which are distributions that are proportional to a constant, (see Tables 3-4)."
- Next, we include an explanation of the prior distribution and physical limitations for the sampled parameters, which starts as follows: "We emphasize that there are currently no theoretical principles or experimental results which indicate possible restrictions for even the order of magnitude of the evaporation rates."
- After the domain restrictions, we explain the parameterization that we use for the evaporation rates and illustrate the sampling procedure (with Figure 1), i.e., we insert the lines 191-218.
- Next we insert the lines 134-138, starting from the sentence "We remark that to create a reliable sample from the underlying 135 parameter distribution...".
- We conclude the Section 2.2.3 with the lines 132-134, where we rephrase the sentences: "In all simulations of the algorithm given in the previous section, the sets of parameters which produce cluster concentrations within the allotted noise level of the data are kept in the chain. More specifically, the sampled parameters 270 of the posterior distribution represent the model evaluations which produce values within the noise level of 0.001% of the data concentrations for each of the respective cluster types".
- 2. C: It is quite confused that throughout the paper, the authors use identification of the rates and thermodynamic enthalpies/entropies. Is it

better to use for example estimate or similar words?

**R**: It is common language to use the words "identification/identify/determine/etc." in the inverse problems literature. We have changed some instances of these words to "estimate/estimation" to suit the atmospheric audience.

3. C: For pairwise marginal posterior distributions, either for evaporation rates or enthalpies/entropies, what criteria the authors used to create these correlations? For example, it seems that evaporation of different monomers from different clusters might be irrelevant.

**R**: We created pairwise marginal posterior distributions from the history of the sampled chains for both cases: in case of evaporation rates and thermodynamic parameters. We observe that the evaporations of different monomers are correlated for some of the cluster types. For example, see Figure C4 and the monomer evaporations from  $(H_2SO_4)_2(NH_3)_1$ ; and Figure C7 and the monomer evaporations from  $(H_2SO_4)_5(NH_3)_4$  which display non-linear correlations. Also the evaporation rates for different non-monomers from different clusters can be correlated. For example, see Figure C7, where the evaporation rates  $(H_2SO_4)_4(NH_3)_4 \rightarrow (H_2SO_4)(NH_3)$  and  $(H_2SO_4)_5(NH_3)_3 \rightarrow (H_2SO_4)_2(NH_3)$  that display inverse linear correlation. However, as the reviewer had mentioned, the evaporation of different monomers from different clusters is irrelevant.

4. C: Section 3.4: can the authors present more details of the comparison instead of just some dry descriptions? For example, the authors can add a table to summarize the knowledge up-to-date regarding the evaporation rates from both measurements and modeling so that the readers can be benefit from reading this paper.

C: We add a short summary paragraph regarding the evaporation rates and how they can be obtained: "The evaporation rates can be obtained either experimentally or computationally, when applying the Quantum Chemical (QC) methods, see Kürten, 2019. Experimental detection was conducted from the measurements in a flow tube (Hanson and Eisele, 2002; Jen et al., 2014, 2016; Hanson et al., 2017) and in the CLOUD chamber (Kurtén et al., 2007; Nadykto and Yu, 2007; Ortega et al., 2012; Elm et al., 2013; Elm and Kristensen, 2017; Yu et al., 2018). However, experimental detection is only available for the charged clusters. The summary of thermodynamic parameters obtained from different methods has previously been published in Kürten, 2019. These parameters can be employed to calculated the evaporation rates at different temperatures."

5. C: Can the authors give some plausible explanation why evaporation rates estimated from transient data seem better than those from steady-state data?

**R**: The transient data is a larger data set than that of just the steadystate data at one temperature. The extra information contained in the transient data reduces the size of the space of allowable evaporation rates, as it there are more restrictions on the possible values the evaporation rates make take. Also the transient data contain information about the slope of the concentrations changing with time, which contributes to quantification of the associated processes (such as collisions and evaporations). We have added the following sentences to emphasize this point:

• Starting in line 262, we change the paragraph to "First, we extend the synthetic measurement data from steady state concentrations to transient concentrations. The data set for transient cluster concentrations at one temperature is larger than the data set for steady-state cluster concentrations at one temperature, as the transient data contains the concentration values at multiple times instances. Also the transient data contain information about the slope of the concentrations changing with time (see Figure C1), which contributes to quantification of the molecular-scale processes (such as collisions and evaporations). We thus expect that this larger data set will reduce the dimension of the solution space for the evaporation rates. Indeed, we will show that this is the case. We generate a synthetic transient cluster concentration data set using the method in Section 2.1. The time resolution of our new synthetic data set is 1.5 minutes, which results in  $\frac{2624}{2}$ 656 total concentration measurements for all the cluster type measured for four different ammonia concentrations. These data sets are illustrated in Figure C1."

Then in line 267, we added: "From this transient cluster concentration data set, we then conduct analogous MCMC runs (as described in Section 2.2). As in the steady-state ..."

• Here we summarize the main differences between the steady-state and transient data as follows: "In the case of the steady-state cluster concentrations we include only one value for each of the 16 cluster types considered in the study, which were taken when the system has attained a steady state (at the end of the ACDC simulation). The transient data contain the steady-state data as subset. Specifically, in this case we consider the concentrations measured when the system has attained the steady state together with the time-step concentration data measured from the starting point to the end of the ACDC simulation."

6. C: The authors claimed that the 5A5N has low variance in free energies. However, an order of magnitude is not small for free energies and it is substantial if this value is applied to the evaporation rates (Line 319 on p18).

**R**: We change the sentence in line 319 to: "Although the posterior distributions of sampled thermodynamic parameters for  $(H_2SO_4)_5(NH_3)_5$  feature higher uncertainties in comparison to the corresponding posterior distributions identified for the smaller clusters, the evaporation rates for evaporations from  $(H_2SO_4)_5(NH_3)_5$ , as calculated from the aforementioned posterior distributions, have low variances, see Table D3."

**R**: Note to TB: We will rather point out that the evaporation rates for the biggest cluster calculated from a posterior distributions of thermodynamic parameters feature low variances. Do you agree?

- 7. C: There are several rather minor comments below:
  - (a) P11, lines 233, do the authors mean that the lower limits of evaporation of a monomer from those clusters are far above the 10-10 as defined for complete growth?
  - (b) P11, line 240, Figures 3-4 can actually be combined to one figure since they basically represent different parts of the same thing. There are some figures that have similar issues.
  - (c) P15, Figure 5, no label for a, b, c, d.
  - (d) P15, line 284, how the evaporation rates of monomers for clusters 2A display inverse linear correlations in Figures C4-C8?
  - (e) P18, the claim that the estimated formation enthalpies vary at most by 1 kcal mol-1, while the variance for the formation entropies is less than 1 calK-1mol-1 is not right.

**R**: We calculated the variances of estimated parameters and the claim will be corrected by replacing the sentence in P18 with "It can be seen that for all the clusters except  $(H_2SO_4)_5(NH_3)_5$  the

variance for the estimated formation enthalpies are less than 0.46 kcal mol-1, while the estimated formation entropies vary at most by 5.4 cal  $K^{-1}mol^{-1}$ ."

- (f) P18, line 313 and line 321, Figure 9 should not appear before figure 8.
- (g) There are lot of typos of molecular sulfuric acid formula throughout the manuscript and a thorough check should be made before submitting the revision. For example, H2SO2.
- (h) The references cited in the text are not followed the journal guidelines.
- (i) Line 34 on p2, subscript; line 37, miss a comma? Line 39, "," is surplus.
- (j) Line 54 on p3, "-" superscript? line 59, miss a comma between experiment and these? It is apparent an ill-sentence (line 65).
- (k) Line 104 on p4, into instead of in to?
- (l) Table 1, it is suggested to add a third column to indicate the number of clusters in each row.
- (m) Line 123 on p5, kinetic model?
- (n) Line 369 on p23, what is question mark for?
- (o) Figure D2, kkal/mol?

**R**: We have made changes to the document to correct for these typos. We are very grateful to the the referee for their careful eye!

---

## Author Response (AR2)

**Response to major revision comments for the paper "Identification of molecular cluster evaporation rates, enthalpies and entropies by Monte Carlo method"**

September 9, 2020

**1 Overview**

In this document we respond to the referee comments for the paper "Identification of molecular cluster evaporation rates, enthalpies and entropies by Monte Carlo method". These comments were provided at the major revision stage of the review process for publication in Atmospheric Chemistry and Physics journal.

We wish to thank the Referee for their insightful comments which we feel substantially increased the quality of the manuscript. We believe that we have addressed all of the major and minor comments made by the reviewer and, in so doing, have produced a paper that is more rigorous in structure and more clear in presentation.

Next, in Section 2 we list the Referee's comments. We also include our comment-by-comment responses. Each of the referee's comments are denoted with "**C**" and our responses to the referee's comments are denoted with "**R**". At the end of the document we supply a marked-up version of the paper which contains a detailed comparison of the previous and revised versions of the manuscript.

**2 Referee comments and our responses**

**Recommendation to the editor**

1. Scientific significance
   Does the manuscript represent a substantial contribution to scientific progress within the scope of this journal (substantial new concepts, ideas, methods, or data)?
   Outstanding **Excellent** Good Fair Low

2. Scientific quality
   Are the scientific approach and applied methods valid? Are the results discussed in an appropriate and balanced way (consideration of related work, including appropriate references)?
   Outstanding Excellent **Good** Fair Low

3. Presentation quality
   Are the scientific results and conclusions presented in a clear, concise, and well structured way (number and quality of figures/tables, appropriate use of English language)?
   Outstanding Excellent Good **Fair** Low

For final publication, the manuscript should be reconsidered after major revisions. I would be willing to review the revised paper, if the editor considers it necessary.

**Suggestions for revision or reasons for rejection (will be published if the paper is accepted for final publication)**
**Comment:** Like the way the authors wrote the paper in a casual way, they seem to respond to the referees' comments in a similar way. It is hard to follow the response letter. There are so many errors, especially what were written in the letter are not the same as those in the revision. Although the authors addressed most of the comments and improvement has indeed seen in the revision, the authors will still need more efforts to improve the quality and the readability of the manuscript. There are still lot of errors/typos and those are really surprising. Below are some issues that need to be resolved before the paper can be publishable in ACP.

**Response:** Thank you for pointing out the typos, errors and inconsistencies that appear in the previous response letter. We strongly apologize for the inconvenience of reading and proof-checking our previous author responses and

tracking related manuscript changes. We further take your recommendations into account and thereby avoid confusions and inaccuracies here.

1. **Comment:** Section 2 is still hard to understand, although it is greatly improved after the revision. In addition, it is very lengthy and redundant. Would it be shortened to make it concise? Some of technical descriptions in my opinion can be moved to the Appendix or supplementary. In addition, the authors use a lot of very short paragraphs and the paper looks like a boring novel. Also, some languages used here are really awkward, given below are some examples:

   **Response:** We have restructured and rewritten Section 2 to improve its quality and readability. We merged and reformulated the paragraphs to make the workflow more logical.
   We first explain generation of synthetic data. Next, we place the section dedicated to Markov chain Monte-Carlo simulations which is subdivided into two parts: selection of minimum and maximum limits for unknown parameters, and overview of the MCMC runs. Both subsections have been made conciser. We have moved the technical details of Metropolis algorithm and its extended version (the DRAM method) to Appendix. These methods are given in A2 and A3, respectively. Additionally we reformulated the language in many of the sentences (see examples below). Here the lines from revised version of the manuscript are given in bold.
   We have moved the first paragraph from "Discussions and future work" to Section 2.1 ("Generation of synthetic data"). Here we explain the sensitivity of MCMC parameter estimation to the quality and limitations of the synthetic data.

   (a) **Comment:** Line 108, by the following method. You really mean by the following procedures or steps, right?
   **Response:** Section 2.1 has been substantially rewritten. We removed this sentence from the text. Instead, we explain the origin of synthetic data as follows: "We generated the birth-death equations using the ACDC code (McGrath et al., 2012), and then solved for the cluster concentrations using the Fortran ordinary differential equation solver VODE (N. Brown et al., 1989). " See **Lines 111-114.**

   (b) **Comment:** Line 118-119, what do you mean "each simulation was initialized with ... and no sulphuric acid"? You mean "without

sulphuric acid"?

**Response:** We reformulated this part into "...the initial sulfuric acid was set to zero in each simulation." See **Line 109.**

(c) **Comment:** Line 122, what is "our particle system"? In particular, what is "particle" here?

**Response:** Thank you for highlighting this important issue which we believe makes our notions more consistent. Throughout the text, we have replaced the 'particle system' with 'the simulated system of clusters' or 'the set of molecular clusters considered here'. See, e.g., **Lines 115, 122.**

(d) **Comment:** Line 124, for time values less than the time at which...., do you mean "for time values before the system has attained the steady state"?

**Response:** Indeed, we intended to say "for time values before the system has attained the steady state". We reformulated this part as follows: "... measured at 1.5 min time intervals before the system reaches a steady state. This corresponded to a total of 41 time steps." See **Lines 123-124.**

(e) **Comment:** Line 126-127, "with time resolution comprising 1.5 minutes", do you mean "with a time interval of 1.5 minutes"?

**Response:** We reformulated this part, as mentioned above.

(f) **Comment:** Line 129, would it be "first...second"?

**Response:** Thank you for recommendation. We have changed the language accordingly. We explain two data sets generated for synthetic data (in **Line 123**). These are referred as "..the fist set..." and "In the second case..." (**Line 125**).

(g) **Comment:** Line 132, reached not reached to and the sentence "The measure of how close ..."is so complicated and awkward. Would it be modified for the sake of readers' benefit?

**Response:** We modified this sentence as follows: "Additionally, we include a convergence parameter for assessing the closeness of cluster concentrations to the steady state for every individual ACDC simulation." See **Lines 127-128.**

(h) **Comment:** Line 228-229, please rearrange the sentence.

**Response:** As it was mentioned above, the discussion related to Metropolis algorithm was moved to Appendix. We changed the sentences which describe the initial assumptions for parameter values as follows: "We first select the flat prior distribution from which we will initially sample unknown parameters, as we wish

to generate physically reasonable parameter estimates. Therefore, we generate unknown parameters within the chosen minimum and maximum bounds where all the points are equally likely to be sampled" (**Lines 367-369**).

**Comment:** It is strongly recommended that this section should be completely rewritten.
**Response:** The section had been restructured, rewritten and shortened in accordance with the advice of the Referee, as mentioned above.

2. **Comment:** The Results and Discussion section looks better than section 2. However, there are still some improvements need to be made. A lot of sentences are quite redundant and need to be modified for conciseness. For example, Line 293, adding "(Table 2) after concentrations will serve the purpose; you don't need to say listed earlier in Table 1, "(Table 1) will be the same. Line 295, "the steady-state" steady-state here I believe is adjective. There are several throughout the manuscript. Line 300, An example of one of, is it necessary to include "one of" here?
**Response:** Following recommendations from the Referee, we restructured and rewrote Section 3. Below we summarize the main structural changes as well as some minor edits.
Initially, we reformulated Sections 3.1-3.3 and thus removed the redundancies. According to the Referee's advise, we omit the references to Tabs. 1 and 2.
Next, we deleted Sections 3.4 and 3.5. The last section (i.e. "Discussion and Future Work") has been redistributed over the manuscript as follows. First, we have moved the text in Lines 436-449 (related to quality of the computer-simulated cluster concentrations) to Section 2.1 ("Generation of synthetic data"). See **Lines 115-122.**
Second, we inserted Lines 450-455 into Conclusion (**Lines 311-317**). This part explains two general principles of inverse problems/Bayesian parameter estimation applied in our study.
The final part of Section 3.5 (Lines 456-465), which explains the correlations between formation enthalpies and entropies is moved to Section 3.3 (**Lines 288-297**). Here we explain the results of parameter estimation after we expressed evaporation rates as parametrized functions of the temperature, with the cluster formation enthalpies and entropies as the unknown parameters. Naturally, we discuss why re-parametrization has improved the results, and the reasons why thermodynamic data display correlations.

Following advise from the Referee, we distinguish between the adjective "steady-state" and the noun "steady state" in the revised version of the manuscript.

The authors decided to keep the sentence in Line 300 as in the previous version ( "An example of one of the sampled chains ...".) We believe that the sentence is sufficiently comprehensive for the reader.

3. **Comment:** The ideas and the results are very interesting and the paper can benefit the community but the way the authors represent do really discouragement and the manuscript needs to be substantially improved in the next revision.

    **Response:** Thank you for acknowledging the advantages and benefits of modelling the type accomplished in this study. We made a considerable effort which we hope has helped to improve the quality of the presentation at the major revision stage. The changes are reflected in updated version of the manuscript.

[revised manuscript text omitted]

$$\gamma_{i+j \rightarrow i,j} = \mathrm{f}(\mathrm{T}, \{\Delta \mathrm{H_k}, \Delta \mathrm{S_k}\}_{k \in \{i+j,i,j\}}). \tag{1}$$

210 In Eq. 1, we set $T = 278$ K or $T = 292$ K. We emphasize that the rates $\gamma_{i+j \rightarrow i,j}$ now depend on temperature and six other parameters: the [..[182] ]formation enthalpy $\Delta \mathrm{H_{i+j}}$ and entropy $\Delta \mathrm{S_{i+j}}$ of the evaporating/fragmenting cluster $i + j$, and the formation enthalpies $\Delta \mathrm{H_i}, \Delta \mathrm{H_j}$ and entropies $\Delta \mathrm{S_i}, \Delta \mathrm{S_j}$ of the product clusters $i$ and $j$ respectively. In this setting $\boldsymbol{\theta}$ represents the array of quantities $\Delta \mathrm{H_{i+j}}, \Delta \mathrm{S_{i+j}}, \Delta \mathrm{H_i}, \Delta \mathrm{H_j}, \Delta \mathrm{S_i}, \Delta \mathrm{S_j}$ with $i + j \in \{1, 2, \ldots, 16\}$. Similar approaches were applied for the inverse problem of chemical kinetics modelled by the Arrhenius equation, where chemical reaction rates are
215 temperature-dependent (?).

[revised manuscript text omitted]

[..[292] ]
* * *
[286] removed: thermodynamic data

[287] removed: In this section we describe another method for regularizing our problem of estimating evaporation rates from steady-state concentration data. We will determine the

[288] removed: from

[289] removed: synthetic,

[290] removed: now measured at

[291] removed: This data set is

[292] removed: We will demonstrate that reparameterization (in terms of thermodynamic data) plus the extended data set transforms our parameter estimation problem from an ill-posed problem to a well-posed one. We use synthetic steady-state cluster concentrations generated for two temperatures to recover the

[revised manuscript text omitted]

**A1.1 The Metropolis algorithm**

We first select the flat prior distribution from which we will initially sample unknown parameters, as we wish to generate physically reasonable parameter estimates. Therefore, we generate unknown parameters within the chosen minimum and maximum bounds where all the points are equally likely to be sampled. Please see Section 2.2.3 and Tabs. 3-4 for more details. From the prior distribution, a starting guess for the parameters $\boldsymbol{\theta}_{old} \in \mathbf{R}^{n_{coef}}$ is chosen (here $n_{coef}$ is the total number of parameters).

**A2 [..[348] ]**

[..[349] ]The Metropolis algorithm then requires us to specify how to sample new parameter values $\boldsymbol{\theta}_{new}$. This is done by choosing a proposal distribution. We chose a multivariate Gaussian proposal density $q$, defined by:

$$[..^{350}]q([..^{351}]\boldsymbol{\theta}_{old}, \boldsymbol{\theta}_{new})[..^{352}] \simeq \exp[..^{353}]\left(-\frac{1}{2}[..^{354}][..^{355}]\left(\boldsymbol{\theta}_{new} - \boldsymbol{\theta}_{old}\right)^{\mathsf{T}}\boldsymbol{\Sigma}^{-1}\left(\boldsymbol{\theta}_{new} - \boldsymbol{\theta}_{old}\right)\right), \tag{A6}$$

where [..[356] ]$\boldsymbol{\Sigma}$ is a covariance matrix (of dimensions $n_{coefs} \times n_{coefs}$) which specifies the scaling and spatial orientation of the Gaussian proposal distribution. As the normalization constants are cancelled out in Eq. A9, we do not take them into consideration.

Next, we run the ACDC and Fortran simulations with the parameter values $\boldsymbol{\theta}_{new}$. We collect the cluster concentration outputs in the column-vector $\mathbf{y}_{mod}(\boldsymbol{\theta}_{new}) \in \mathbb{R}^{n_{out}}$, where $n_{out}$ is the number of [..[357] ]elements. The candidate vector of parameters $\boldsymbol{\theta}_{new}$ is either accepted or rejected according to the least-squares fit of $\mathbf{y}_{mod}(\boldsymbol{\theta}_{new})$ to the synthetic cluster concentrations $\mathbf{y}_{exp}$:

$$SS(\boldsymbol{\theta}_{new}) = \sum_{i=1}^{n_{out}} \frac{\left(y_{exp,i} - y_{mod,i}(\boldsymbol{\theta}_{
[revised manuscript text omitted]

 $(8.03 \times 10^2, 8.36 \times 10^2)$ | $8.23 \times 10^2$ |
| 2: $1A1N \to 1N$ | $\mathbf{4.76 \times 10^3}$
 $(4.66 \times 10^3, 4.87 \times 10^3)$ | $4.74 \times 10^3$ |
| 3: $2A1N \to 1A$ | $\mathbf{3.64 \times 10^{-4}}$
 $(3.48 \times 10^{-4}, 3.84 \times 10^{-4})$ | $3.64 \times 10^{-4}$ |
| 4: $2A1N \to 1N$ | $\mathbf{1.23 \times 10^{-3}}$
 $(1.16 \times 10^{-3}, 1.29 \times 10^{-3})$ | $1.21 \times 10^{-3}$ |
| 5: $3A1N \to 1A$ | $\mathbf{3.01 \times 10^1}$
 $(2.93 \times 10^1, 3.09 \times 10^1)$ | $3.02 \times 10^1$ |
| 6: $3A1N \to 2A$ | $\mathbf{6.12 \times 10^{-6}}$
 $(5.77 \times 10^{-6}, 6.47 \times 10^{-6})$ | $6.09 \times 10^{-6}$ |
| 7: $2A2N \to 1N$ | $\mathbf{1.77 \times 10^2}$
 $(1.71 \times 10^2, 1.82 \times 10^2)$ | $1.76 \times 10^2$ |
| 8: $2A2N \to 1A1N$ | $\mathbf{5.33 \times 10^{-6}}$
 $(5.02 \times 10^{-6}, 5.64 \times 10^{-6})$ | $5.33 \times 10^{-6}$ |
| 9: $3A2N \to 1A$ | $\mathbf{6.09 \times 10^{-4}}$
 $(5.14 \times 10^{-4}, 7.05 \times 10^{-4})$ | $6.07 \times 10^{-4}$ |
| 10: $3A2N \to 1N$ | $\mathbf{3.89 \times 10^{-3}}$
 $(3.27 \times 10^{-3}, 4.50 \times 10^{-3})$ | $3.84 \times 10^{-3}$ |
| 11: $3A2N \to 1A1N$ | $\mathbf{1.65 \times 10^{-5}}$
 $(1.40 \times 10^{-5}, 1.90 \times 10^{-5})$ | $1.64 \times 10^{-5}$ |
| 12: $4A2N \to 1A$ | $\mathbf{5.45 \times 10^0}$
 $(5.25 \times 10^0, 5.65 \times 10^0)$ | $5.43 \times 10^0$ |
| 13: $4A2N \to 2A$ | $\mathbf{1.49 \times 10^{-6}}$
 $(1.27 \times 10^{-6}, 1.72 \times 10^{-6})$ | $1.48 \times 10^{-6}$ |
| 14: $4A2N \to 1A1N$ | $\mathbf{2.82 \times 10^{-6}}$
 $(2.37 \times 10^{-6}, 3.26 \times 10^{-6})$ | $2.80 \times 10^{-6}$ |
| 15: $4A2N \to 2A1N$ | $\mathbf{1.01 \times 10^{-1}}$
 $(8.35 \times 10^{-2}, 1.18 \times 10^{-1})$ | $9.94 \times 10^{-2}$ |
| 16: $3A3N \to 1N$ | $\mathbf{4.64 \times 10^{-2}}$
 $(4.47 \times 10^{-2}, 4.81 \times 10^{-2})$ | $4.60 \times 10^{-2}$ |
| 17: $3A3N \to 1A1N$ | $\mathbf{3.77 \times 10^{-9}}$
 $(3.19 \times 10^{-9}, 4.36 \times 10^{-9})$ | $3.74 \times 10^{-9}$ |
| 18: $4A3N \to 1A$ | $\mathbf{2.08 \times 10^{-3}}$
 $(1.86 \times 10^{-3}, 2.29 \times 10^{-3})$ | $2.10 \times 10^{-3}$ |
| 19: $4A3N \to 1N$ | $\mathbf{1.87 \times 10^{-5}}$
 $(1.69 \times 10^{-5}, 2.05 \times 10^{-5})$ | $1.88 \times 10^{-5}$ |
| 20: $4A3N \to 1A1N$ | $\mathbf{1.21 \times 10^{-8}}$
 $(1.09 \times 10^{-8}, 1.33 \times 10^{-8})$ | $1.23 \times 10^{-8}$ |

**Table D2.** Part 1. Evaporation rates (units given in $s^{-1}$) computed from a posterior distribution of the thermodynamic parameters (cluster formation enthalpies and entropies) which had previously been determined from the steady-state concentration measurements at temperatures 278 and 292 K. Here the mode of distribution (bold face) is given together with the range of possible values in the parenthesis. The last column presents the quantum-chemistry-based evaporation rates used for creating the synthetic data (borrowed from **?**). The notation $xAyN$ corresponds to a cluster with x sulfuric acid and y ammonia molecules.[..[460]]

| Symbol | Steady-state data for 278 K and 292 K $(s^{-1})$ | QC $(s^{-1})$ |
|---|---|---|
| 21: 4A3N → 2A1N | **$1.65 \times 10^{-8}$** ($1.30 \times 10^{-8}$,$1.99 \times 10^{-8}$) | $1.66 \times 10^{-8}$ |
| 22: 5A3N → 1A | **$7.98 \times 10^{-1}$** ($7.63 \times 10^{-1}$,$8.43 \times 10^{-1}$) | $7.83 \times 10^{-1}$ |
| 23: 5A3N → 2A | **$6.40 \times 10^{-7}$** ($5.76 \times 10^{-7}$,$7.24 \times 10^{-7}$) | $6.37 \times 10^{-7}$ |
| 24: 5A3N → 1A1N | **$1.71 \times 10^{-9}$** ($1.54 \times 10^{-9}$,$1.88 \times 10^{-9}$) | $1.70 \times 10^{-9}$ |
| 25: 5A3N → 2A1N | **$1.87 \times 10^{-5}$** ($1.66 \times 10^{-5}$,$2.07 \times 10^{-5}$) | $1.85 \times 10^{-5}$ |
| 26: 5A3N → 2A2N | **$3.56 \times 10^{-10}$** ($2.83 \times 10^{-10}$,$4.30 \times 10^{-10}$) | $3.52 \times 10^{-10}$ |
| 27: 4A4N → 1N | **$3.82 \times 10^{1}$** ($3.69 \times 10^{1}$,$3.95 \times 10^{1}$) | $3.75 \times 10^{1}$ |
| 28: 4A4N → 1A1N | **$8.97 \times 10^{-6}$** ($8.13 \times 10^{-6}$,$1.01 \times 10^{-5}$) | $9.06 \times 10^{-6}$ |
| 29: 4A4N → 2A2N | **$1.34 \times 10^{-9}$** ($1.07 \times 10^{-9}$,$1.62 \times 10^{-9}$) | $1.33 \times 10^{-9}$ |
| 30: 5A4N → 1A | **$1.76 \times 10^{-3}$** ($1.56 \times 10^{-3}$,$1.96 \times 10^{-3}$) | $1.77 \times 10^{-3}$ |
| 31: 5A4N → 1N | **$8.70 \times 10^{-2}$** ($7.68 \times 10^{-2}$,$1.00 \times 10^{-1}$) | $8.87 \times 10^{-2}$ |
| 32: 5A4N → 1A1N | **$7.42 \times 10^{-6}$** ($6.59 \times 10^{-6}$,$8.24 \times 10^{-6}$) | $7.33 \times 10^{-6}$ |
| 33: 5A4N → 2A1N | **$2.92 \times 10^{-5}$** ($2.45 \times 10^{-5}$,$3.40 \times 10^{-5}$) | $2.97 \times 10^{-5}$ |
| 34: 5A4N → 2A2N | **$6.40 \times 10^{-9}$** ($5.40 \times 10^{-9}$,$7.40 \times 10^{-9}$) | $6.42 \times 10^{-9}$ |
| 35: 4A5N → 1N | **$8.85 \times 10^{2}$** ($8.58 \times 10^{2}$,$9.12 \times 10^{2}$) | $8.89 \times 10^{2}$ |
| 36: 5A5N → 1A | **$5.38 \times 10^{-10}$** ($2.01 \times 10^{-11}$,$2.24 \times 10^{-9}$) | $2.23 \times 10^{-10}$ |
| 37: 5A5N → 1N | **$2.77 \times 10^{-4}$** ($1.09 \times 10^{-5}$,$1.15 \times 10^{-3}$) | $1.17 \times 10^{-4}$ |
| 38: 5A5N → 1A1N | **$5.05 \times 10^{-11}$** ($1.87 \times 10^{-12}$,$2.10 \times 10^{-10}$) | $2.11 \times 10^{-11}$ |
| 39: 5A5N → 2A2N | **$3.07 \times 10^{-11}$** ($1.16 \times 10^{-12}$,$1.28 \times 10^{-10}$) | $1.31 \times 10^{-11}$ |

**Table D3.** Part 2. Evaporation rates (units given in $s^{-1}$) computed from a posterior distribution of the thermodynamic parameters (cluster formation enthalpies and entropies) which had previously been determined from the steady-state concentration measurements at temperatures 278 and 292 K. Here the mode of distribution (bold face) is given together with the range of possible values in the parenthesis. The last column presents the quantum-chemistry-based evaporation rates used for creating the synthetic data (borrowed from **?**). The notation $xAyN$ corresponds to a cluster with x sulfuric acid and y ammonia molecules.[..[461] ]

*Author contributions.* Author Shcherbacheva A. produced the codes and conducted all the computational experiments for generation of the synthetic data and the MCMC parameter estimation, prepared all the plots presented in the manuscripts. Authors Balehowsky T. and Shcherbacheva A., Kurtén T. and Vehkamäki H. and Haario H. are responsible for writing the manuscript. Author Olenius T. assisted with

generation of the synthetic data, preformed sanity check of the results, gave valuable comments regarding the manuscript. Authors Helin T. and Balehowsky T. actively participated in development of the methodological approach. Author Laine M. provided technical assistance with the 'mcmcstat' toolbox which was used for MCMC simulations. Author Kubečka J. assisted with the code compilation and debug. Author Haario H. assisted with interpretation of the MCMC results and proper usage of the DRAM computational method. Authors Kurtén T. and Vehkamäki H. helped to interpret the outcomes of the study.

*Competing interests.*  The authors declare that they have no conflict of interest

*Acknowledgements.*  We thank the European Research Council project 692891-DAMOCLES, Academy of Finland (project number 307331), and University of Helsinki: Faculty of Science ATMATH project, for funding, and the CSC-IT Centre for Science in Espoo, Finland, for computational resources. We also thank Olli Pakarinen (Institute for Atmospheric and Earth System Research, University of Helsinki, Helsinki, Finland) for advise in plotting the synthetic data used in the present study.

---

## Author Response (AR3)

**Response to major revision comments for the paper "Identification of molecular cluster evaporation rates, enthalpies and entropies by Monte Carlo method"**

October 14, 2020

**1 Overview**

In this document we respond to the referee comments for the paper "Identification of molecular cluster evaporation rates, enthalpies and entropies by Monte Carlo method". These comments were provided at the final minor revision stage of the review process for publication in Atmospheric Chemistry and Physics journal.

We wish to thank the Referee for their helpful comments and attentive proof-checking of the manuscript which we believe helped in formulating a more solid conclusion part and correcting the remaining typos. We feel that we have addressed all the issues mentioned by the reviewer and, in so doing, polished the final version of our manuscript.

Next, in Section 2 we list the Referee's comments. We also include our comment-by-comment responses. At the end of the document we supply a marked-up version of the paper which contains a detailed comparison of the previous and revised versions of the manuscript.

**2 Referee comments and our responses**

**Recommendation to the editor**

1. Scientific significance
   Does the manuscript represent a substantial contribution to scientific progress within the scope of this journal (substantial new concepts, ideas, methods, or data)?
   Outstanding **Excellent** Good Fair Low

2. Scientific quality
   Are the scientific approach and applied methods valid? Are the results discussed in an appropriate and balanced way (consideration of related work, including appropriate references)?
   Outstanding Excellent **Good** Fair Low

3. Presentation quality
   Are the scientific results and conclusions presented in a clear, concise, and well structured way (number and quality of figures/tables, appropriate use of English language)?
   Outstanding Excellent **Good** Fair Low

For final publication, the manuscript should be accepted subject to minor revisions

**Suggestions for revision or reasons for rejection (will be published if the paper is accepted for final publication)**
**Comment:** This revision has been substantially improved. The authors addressed all the concerns raised in the last review. The text has been shortened and much concise in this version. There are several typos needed to be corrected and it is in a much better shape. It is recommended to be published after some minor changes.

**Response:** Thank you for acknowledging the improvements made in the revised version of our manuscript. We further take your recommendations into account and thereby improve the conclusion and correct the typos.

**Comment:** 1. The current conclusions of the paper seem just too general. Although it is indeed admitted that estimates of evaporation rates are quite

challenging and intriguing, it might still be more encouraging if some values or ranges of values for the evaporation rates can be concluded from the paper.

**Response:** Since the paper does not actually treat any real experimental data, we can unfortunately not make any direct conclusions about values or ranges for the actual real evaporation rates (if this is what the reviewer is asking for). However, we agree that the conclusions can be made more concrete, with numerical examples of the values or ranges corresponding to our synthetic data. We have accordingly added a section on this to the conclusions.

**Addition to conclusions** In Line 319, we add: "...to within acceptable accuracy In practice, the most important evaporation rates for modelling new particle formation are those which are roughly of the same order of magnitude as the rates at which the clusters collide with the vapor molecules. If we assume that the mixing ratios for the clustering vapours are in the ppt...ppb range and use kinetic gas theory collision rates for small molecules and nanometer-sized clusters, we approximately should obtain evaporation rates in the range of $10^{-3}$ to $10^3$ $s^{-1}$. Fortunately, our approach is able to constrain these evaporations rates to within a factor of 10 or less. Evaporation rates below $10^{-4}$ $s^{-1}$ are not as well constrained. However, the corresponding processes are usually not relevant for determining overall new-particle formation rates. While the high accuracy of estimated evaporation rates originates from the assumptions of small-noise synthetic data and the concentrations measured for all the cluster types, similar accuracy can be expected if high-quality experimental steady-state data at two temperatures is used instead. " **Comment:** 2. Some typos

1. Line 40 on p.2, "methods, (", ","here is redundant?

2. Line 103 on p.4, two parentheses? Line 105, a list of the 16 considered clusters? Line 106, between.1.3??? Line 113, N. Brown et al., please keep the citation consistent;

3. Line 125 on p.5, data sets?

4. Line 164 on p.6, +-10, can one character used for "+-"?

5. Line 194 on p.8, Figure 1 is not below, below is not necessary here;

6. Line 203 on p.9, are you sure you can write the source rate like this?

7. Line 264 on p.13, a "," is needed after C1-C2.

[revised manuscript text omitted]

 $(1.30 \times 10^{-8}, 1.99 \times 10^{-8})$ | $1.66 \times 10^{-8}$ |
| 22: 5A3N → 1A | **$7.98 \times 10^{-1}$**
 $(7.63 \times 10^{-1}, 8.43 \times 10^{-1})$ | $7.83 \times 10^{-1}$ |
| 23: 5A3N → 2A | **$6.40 \times 10^{-7}$**
 $(5.76 \times 10^{-7}, 7.24 \times 10^{-7})$ | $6.37 \times 10^{-7}$ |
| 24: 5A3N → 1A1N | **$1.71 \times 10^{-9}$**
 $(1.54 \times 10^{-9}, 1.88 \times 10^{-9})$ | $1.70 \times 10^{-9}$ |
| 25: 5A3N → 2A1N | **$1.87 \times 10^{-5}$**
 $(1.66 \times 10^{-5}, 2.07 \times 10^{-5})$ | $1.85 \times 10^{-5}$ |
| 26: 5A3N → 2A2N | **$3.56 \times 10^{-10}$**
 $(2.83 \times 10^{-10}, 4.30 \times 10^{-10})$ | $3.52 \times 10^{-10}$ |
| 27: 4A4N → 1N | **$3.82 \times 10^{1}$**
 $(3.69 \times 10^{1}, 3.95 \times 10^{1})$ | $3.75 \times 10^{1}$ |
| 28: 4A4N → 1A1N | **$8.97 \times 10^{-6}$**
 $(8.13 \times 10^{-6}, 1.01 \times 10^{-5})$ | $9.06 \times 10^{-6}$ |
| 29: 4A4N → 2A2N | **$1.34 \times 10^{-9}$**
 $(1.07 \times 10^{-9}, 1.62 \times 10^{-9})$ | $1.33 \times 10^{-9}$ |
| 30: 5A4N → 1A | **$1.76 \times 10^{-3}$**
 $(1.56 \times 10^{-3}, 1.96 \times 10^{-3})$ | $1.77 \times 10^{-3}$ |
| 31: 5A4N → 1N | **$8.70 \times 10^{-2}$**
 $(7.68 \times 10^{-2}, 1.00 \times 10^{-1})$ | $8.87 \times 10^{-2}$ |
| 32: 5A4N → 1A1N | **$7.42 \times 10^{-6}$**
 $(6.59 \times 10^{-6}, 8.24 \times 10^{-6})$ | $7.33 \times 10^{-6}$ |
| 33: 5A4N → 2A1N | **$2.92 \times 10^{-5}$**
 $(2.45 \times 10^{-5}, 3.40 \times 10^{-5})$ | $2.97 \times 10^{-5}$ |
| 34: 5A4N → 2A2N | **$6.40 \times 10^{-9}$**
 $(5.40 \times 10^{-9}, 7.40 \times 10^{-9})$ | $6.42 \times 10^{-9}$ |
| 35: 4A5N → 1N | **$8.85 \times 10^{2}$**
 $(8.58 \times 10^{2}, 9.12 \times 10^{2})$ | $8.89 \times 10^{2}$ |
| 36: 5A5N → 1A | **$5.38 \times 10^{-10}$**
 $(2.01 \times 10^{-11}, 2.24 \times 10^{-9})$ | $2.23 \times 10^{-10}$ |
| 37: 5A5N → 1N | **$2.77 \times 10^{-4}$**
 $(1.09 \times 10^{-5}, 1.15 \times 10^{-3})$ | $1.17 \times 10^{-4}$ |
| 38: 5A5N → 1A1N | **$5.05 \times 10^{-11}$**
 $(1.87 \times 10^{-12}, 2.10 \times 10^{-10})$ | $2.11 \times 10^{-11}$ |
| 39: 5A5N → 2A2N | **$3.07 \times 10^{-11}$**
 $(1.16 \times 10^{-12}, 1.28 \times 10^{-10})$ | $1.31 \times 10^{-11}$ |

**Table D3.** Part 2. Evaporation rates at temperature 278 K (units given in $s^{-1}$) computed from a posterior distribution of the thermodynamic parameters (cluster formation enthalpies and entropies) which had previously been determined from the steady-state concentration measurements at temperatures 278 and 292 K. Here the mode of distribution (bold face) is given together with the range of possible values in the parenthesis. The last column presents the quantum-chemistry-based evaporation rates used for creating the synthetic data (borrowed from **?**). The notation $xAyN$ corresponds to a cluster with x sulfuric acid and y ammonia molecules.

*Author contributions.* Author Shcherbacheva A. produced the codes and conducted all the computational experiments for generation of the synthetic data and the MCMC parameter estimation, prepared all the plots presented in the manuscripts. Authors Balehowsky T. and Shcherbacheva A., Kurtén T. and Vehkamäki H. and Haario H. are responsible for writing the manuscript. Author Olenius T. assisted with

generation of the synthetic data, preformed sanity check of the results, gave valuable comments regarding the manuscript. Authors Helin T. and Balehowsky T. actively participated in development of the methodological approach. Author Laine M. provided technical assistance with the 'mcmcstat' toolbox which was used for MCMC simulations. Author Kubečka J. assisted with the code compilation and debug. Author Haario H. assisted with interpretation of the MCMC results and proper usage of the DRAM computational method. Authors Kurtén T. and Vehkamäki H. helped to interpret the outcomes of the study.

*Competing interests.* The authors declare that they have no conflict of interest

*Acknowledgements.* We thank the European Research Council project 692891-DAMOCLES, Academy of Finland (project number 307331), and University of Helsinki: Faculty of Science ATMATH project, for funding, and the CSC-IT Centre for Science in Espoo, Finland, for computational resources. We also thank Olli Pakarinen (Institute for Atmospheric and Earth System Research, University of Helsinki, Helsinki, Finland) for advise in plotting the synthetic data used in the present study.